Description and basic evaluation of simulated mean state, internal variability, and climate sensitivity
in MIROC6
Hiroaki Tatebe[1], Tomoo Ogura[2], Tomoko Nitta[3], Yoshiki Komuro[1], Koji Ogochi[1], Toshihiko
Takemura[4], Kengo Sudo[5], Miho Sekiguchi[6], Manabu Abe[1], Fuyuki Saito[1], Minoru Chikira[3], Shingo
Watanabe[1], Masato Mori[7], Nagio Hirota[2], Yoshio Kawatani[1], Takashi Mochizuki[1], Kei Yoshimura[3],
Kumiko Takata[2], Ryouta O'ishi[3], Dai Yamazaki[8], Tatsuo Suzuki[1], Masao Kurogi[1], Takahito Kataoka[1],
Masahiro Watanabe[3], and Masahide Kimoto[3]
1: Japan Agency for Marine-Earth Science and Technology, Yokohama, Japan
2: National Institute for Environmental Studies, Tsukuba, Japan
3: Atmosphere and Ocean Research Institute, University of Tokyo, Kashiwa, Japan
4: Research Institute for Applied Mechanics, Kyushu University, Kasuga, Japan
5: Graduate School of Environmental Studies, Nagoya University, Nagoya, Japan
6: Tokyo University of Marine Science and Technology, Tokyo, Japan
7: Research Center for Advanced Science and Technology, University of Tokyo, Tokyo, Japan
8: Institute of Industrial Sciences, University of Tokyo, Tokyo, Japan
Corresponding author: Hiroaki Tatebe (E-mail: tatebe@jamstec.go.jp)
Research Center for Environmental Modeling and Application, Japan Agency for Marine-Earth
Science and Technology
3173-25 Showamachi, Kanazawaku, Yokohama, Kanagawa 236-0001, Japan
**Abstract**

24         The sixth version of the Model for Interdisciplinary Research on Climate (MIROC), called

MIROC6, was cooperatively developed by a Japanese modeling community. In the present manuscript,
simulated mean climate, internal climate variability, and climate sensitivity in MIROC6 are evaluated
and briefly summarized in comparison with the previous version of our climate model (MIROC5) and
observations. The results show that overall reproducibility of mean climate and internal climate
variability in MIROC6 is better than that in MIROC5. The tropical climate systems (e.g., summertime
precipitation in the western Pacific and the eastward propagating Madden-Julian Oscillation) and the
mid-latitude atmospheric circulations (e.g., the westerlies, the polar night jet, and troposphere-
stratosphere interactions) are significantly improved in MIROC6. These improvements can be
attributed to the newly implemented parameterization for shallow convective processes and to the
inclusion of the stratosphere . While there are significant differences in climates and variabilities
between the two models, the effective climate sensitivity of 2.5 K remains the same because the
differences in radiative forcing and climate feedback tend to offset each other. With an aim towards
contributing to the sixth phase of the Coupled Model Intercomparison Project, designated simulations
tackling a wide range of climate science issues, as well as seasonal-to-decadal climate predictions and
future climate projections, are currently ongoing using MIROC6.

## 1 Introduction

As the global warming due to increasing emissions of the anthropogenic greenhouse gases progresses, global and regional patterns of atmospheric circulations and precipitation as well as temperature are projected to be drastically changed until the end of the twentieth-first century (e.g., Neelin et al., 2006; Zhang et al., 2007; Bengtsson et al., 2009; Andrews et al., 2010; Scaife et al., 2012) and that occurrence frequency of extreme weather events such as heatwaves, droughts will be increased and extratropical cyclones will be stronger than in the present (e.g., Mizuta et al., 2012; Sillmann et al., 2013; Zappa et al., 2013). Corresponding to the atmospheric changes under the global warming, the sea levels will rise due to the thermal expansion of sea water and ice-sheet melting in the polar continental regions (e.g., Church and White, 2011; Bamber and Aspinall, 2013). Additionally, ocean acidification due to absorption of atmospheric carbon dioxide ($CO_2$) and changes in carbon-nitrogen cycles are expected to lead to the loss of Earth biodiversity (e.g., Riebesell et al., 2009; Rockström, et al. 2009; Taucher and Oschlies, 2011; Watanabe et al., 2017). Societal demands for information on the global and regional climate changes have increased significantly worldwide in order to meet information requirements for political decision making related to mitigation and adaptation to the global warming.

The Intergovernmental Panel on Climate Change (IPCC) has continuously published the assessment reports (ARs) in which a comprehensive view of past, present, and future climate changes on various timescales, including the centennial global warming, are synthesized. Together with observations, climate models have been contributing to the IPCC-ARs through a broad range of numerical simulations, especially, future climate projections after the twenty-first century. However, there are many uncertainties in future climate projections and the range of uncertainties has not been narrowed by an update of the IPCC reports. The uncertainties are arising from imperfections of climate models in representing micro- to global-scale physical and dynamical processes in sub-systems of the

Earth's climate and their interactions. To reduce the uncertainties and errors in climate projections and
predictions, utilizing observations, extracting essences of physical processes in the real climate, and
investigating the response of the climate system to various external forcings based on a set of climate
model simulations are necessary. In particular, a state-of-the-art climate model which can represent
various processes in the Earth's climate system is a powerful tool for deeper understanding the Earth's
climate system.
One of Japanese climate models, which is called MIROC (Model for Interdisciplinary
Research on Climate), has been cooperatively developed at the Center for Climate System Research
(CCSR; the precursor of a part of the Atmosphere and Ocean Research Institute), the University of
Tokyo, the Japan Agency for Marine-Earth Science and Technology (JAMSTEC), and the National
Institute for Environmental Studies (NIES). Utilizing MIROC, our Japanese climate modelling group
has been tackling a wide range of climate science issues and seasonal-to-decadal climate predictions
and future climate projections. At the same time, by providing simulation data, we have been
participating to the third and fifth phases of the Coupled Model Intercomparison Projects (CMIP3 and
CMIP5; Meehl et al. 2007; Taylor et al. 2011) which have been contributing to the IPCC-ARs by
synthesizing multi-model ensemble datasets.
In the years up to the IPCC fifth assessment report (IPCC-AR5; IPCC 2013), we have
developed four versions of MIROC, three of which (MIROC3m, MIROC3h, and MIROC4h) have
almost the same dynamical and physical packages, but different resolutions. MIROC3m (K-1 model
developers, 2004) is composed of T42L20 atmosphere and 1.4°L43 ocean. Resolutions of MIROC3h
(K-1 model developers, 2004) are higher than MIROC3m and are T106L56 for the atmosphere and
eddy-permitting for the ocean (1/4° × 1/6°). Only the horizontal resolution of the atmosphere of
MIROC3h is changed to T213 in MIROC4h (Sakamoto et al., 2012). MIROC5 is composed of T85L40
atmosphere and 1.4°L50 ocean, but with considerably updated physical and dynamical packages
(Watanabe et al., 2010). These models have been used to study various scientific issues such as the
detection of natural influences on climate changes (e.g., Nozawa et al., 2005; Mori et al, 2014;
Watanabe et al., 2014), uncertainty quantification of climate sensitivity (e.g., Shiogama et al., 2012;
Kamae et al., 2016), future projections of regional sea-level rises (e.g., Suzuki et al., 2005; Suzuki and
Ishii, 2011), and mechanism studies on tropical decadal variability (e.g., Tatebe et al., 2013; Mochizuki
et al., 2016).

During the last decade, our efforts have been preferentially devoted to providing science-

oriented risk information on climate changes that is beneficial to international, domestic, and
municipal communities. For example, so-called event attribution (EA) studies with large ensemble
simulations initiated from slightly different conditions have been conducted in order to statistically
evaluate influences of the global warming on the occurrence frequencies of observed individual
extremes (e.g., Imada et al., 2013; Watanabe et al., 2013; Shiogama et al., 2014). Seasonal-to-decadal
climate predictions are also of significant concerns. By initializing prognostic variables in our climate
models using observation-based data (Tatebe et al., 2012), significant prediction skills in several
specific phenomena, such as the El Niño/Southern Oscillation (ENSO) and the Arctic sea-ice extent
on seasonal timescales, the Pacific Decadal Oscillations (PDO; Mantua et al., 1997), the Atlantic
Multi-decadal Oscillations (AMO; Schlesinger and Ramankutty, 2004), and the tropical trans-basin
interactions between the Pacific and the Atlantic on decadal timescales, are detected (e.g., Mochizuki
et al., 2010; Chikamoto et al. 2015; Imada et al., 2015; Ono et al., 2018).

However, while the applicability of MIROC has been extended to a wide range of climate

science issues, almost all of the above-mentioned approaches were based on our medium-resolution
versions of MIROC (MIROC3m and MIROC5), and it is well known that higher-resolution models
are capable of better representing the model mean climate and internal climate variability, such as
regional extremes, orographic winds, and oceanic western boundary currents/eddies than lower-
resolution models (e.g., Shaffrey et al., 2009; Roberts et al., 2009; Sakamoto et al., 2012). Nevertheless,
there remain persistent biases associated with, for example, cloud-aerosol-radiative feedback and
turbulent vertical mixing of the air in the planetary boundary layer (e.g., Bony and Dufresne, 2005;
Bodas-Salcedo et al., 2012; Williams et al., 2013), which are tightly linked with dominant uncertainties
in climate projections. Therefore, improvement of physical parameterizations for sub-grid scale
processes is essential for better representing observed climatic-mean states and internal climate
variability. As well as physical parameterizations, enhanced vertical resolution in both of atmosphere
and ocean components, along with a highly accurate tracer advection scheme, have been suggested to
have impacts on reproducibility of model-climate and internal climate variations (e.g., Tatebe and
Hasumi, 2010; Ineson and Scaife, 2009; Scaife et al., 2012).
Recently, we have developed the sixth version of MIROC, called MIROC6. This newly
developed climate model has updated physical parameterizations in all sub-modules. In order to
suppress an increase of computational cost, the horizontal resolutions of MIROC6 are not significantly
higher than those of MIROC5. The reason is that a larger number of ensemble members are required
to realize significant seasonal predictions of, for example, the wintertime Eurasian climate (Murphy
et al., 1990; Scaife et al., 2014). Indeed, climate predictions by the older versions of MIROC having
at most 10 ensemble members are skillful only in the tropical climate and the mid-latitude ocean not
in the mid-latitude atmosphere. Large ensemble predictions are also required in decadal-scale
predictions in order to evaluate the human influences on the near-term climate changes. The model
top in MIROC6 is placed at the 0.004 hPa pressure level which is higher than that of MIROC5 (3 hPa),
and the stratospheric vertical resolution has been enhanced in comparison to MIROC5 in order to
represent the stratospheric circulations. Overall, the reproducibility of the mean climate and internal
variability of MIROC6 is better than those of MIROC5, but the model's computational cost is about
3.6 times as large as that of MIROC5. Considering that the computational costs of large ensemble
predictions based on climate models with horizontal resolutions of, for example, 50 km atmosphere
and eddy-resolving ocean are still huge on recent computer systems, the use of relatively low
resolution models such as MIROC6 with further elaborated parameterizations can still be actively
useful in science-oriented climate studies and climate predictions produced for societal needs.
The rest of the present paper is organized as follows. We describe the model configuration,
tuning and spin-up procedures in Section 2, while simulated mean-state, internal variability, and
climate sensitivity are evaluated in Section 3. Simulation performance of MIROC6 and remaining
issues are briefly summarized and discussed in Section 4. Currently, MIROC6 is being used for various
simulations designed by the sixth phase of the CMIP (CMIP6; Eyring et al., 2016), which aims to
strengthen the scientific basis of the IPCC-AR6. Large ensemble simulations and climate predictions
using MIROC6 are also on-going for science-oriented studies in our modeling group and for societal
benefits. In addition, the latest earth system model version of MIROC with the global carbon cycle,
whose physical core will be MIROC6, has been developed for the CMIP6 towards further wide range
issues of climate and societal applications (Hajima et al., in preparation).

**2 Model configurations and spinup procedures**
MIROC6 is composed of three sub-models: atmosphere, land, and sea ice-ocean. The
atmospheric model is based on the CCSR-NIES atmospheric general circulation model (AGCM;
Numaguti et al., 1997). The land surface model is based on Minimal Advanced Treatments of Surface
Interaction and Runoff (MATSIRO; Takata et al. 2003), which includes a river routing model of Oki
and Sud (2003) based on a kinematic wave flow equation (Ngo-Duc et al., 2007) and a lake module
where one-dimensional thermal diffusion and mass conservation are considered. The sea ice-ocean
model is based on the CCSR Ocean Component model (COCO; Hasumi, 2006). A coupler system
calculates heat and freshwater fluxes between the sub-models in order to ensure that all fluxes are
conserved within machine precision and then exchanges the fluxes among the sub-models (Suzuki et
al., 2009). No flux adjustments are used in MIROC6. In the remaining part of this section, we will
provide details of MIROC6 configurations, focusing on updates from MIROC5. Readers may also
refer to Table A in Appendix where the updates are briefly summarized.

**2.1 Atmospheric component**

MIROC6 employs a spectral dynamical core in its AGCM component as in MIROC5. The

horizontal resolution is a T85 spectral truncation that is an approximately 1.4° grid interval for both
latitude and longitude. The vertical grid coordinate is a hybrid $\sigma$-$p$ coordinate (Arakawa and Konor,
1996). The model top is placed at 0.004 hPa, and there are 81 vertical levels (Fig. 1a). The vertical
grid arrangement in MIROC6 is considerably enhanced in comparison to that in MIROC5 (40 levels;
3 hPa) in order that the stratospheric circulations can be represented. A sponge layer that damps wave
motions is set at the model top level by increasing Rayleigh friction to prevent extra wave reflection
near the model top. The atmospheric component of MIROC6 has standard physical parameterizations
for cumulus convections, radiation transfer, cloud microphysics, turbulence, and gravity wave drag. It
also has an aerosol module. These are basically the same as those used in MIROC5, but several updates
have been made, as will be detailed below. The parameterizations for cloud micro-physics and
planetary boundary layer processes in MIROC6 are the same as in MIROC5. The standard timestep
for MIROC6 is 6 minutes which is shorter than that of MIROC5 (12 minutes) because stratospheric
winds whose speed sometimes exceeds 150 ms$^{-1}$ must be resolved in time integration. The timestep
for radiative transfer models is set separately and is 3 hours in both of MIROC6 and MIROC5.

A cumulus parameterization proposed by Chikira and Sugiyama (2010), which uses an

entrainment formulation of Gregory (2001), is adopted in MIROC6 as in MIROC5. This
parameterization deals with multiple cloud types including shallow cumulus and deep convective
clouds. MIROC5, however, tends to overestimate the low-level cloud amounts over the low-latitude
oceans and has a dry bias in the free troposphere. These biases appear to be the result of insufficient
vertical mixing of the humid air in the planetary boundary layer and the dry air in the free troposphere.
To alleviate these biases, an additional parameterization for shallow cumulus convection based on
Park and Bretherton (2009) is implemented in MIROC6. Shallow convections associated with the
atmospheric instability are calculated by the Chikira and Sugiyama (2010) scheme, and those
associated with turbulence in the planetary boundary layer are represented by the Park and Bretherton
(2009) scheme. The shallow convective parameterization is a mass flux scheme based on a buoyancy-
sorting, entrainment-detrainment single plume model that calculates the vertical transport of liquid
water, potential temperature, total water mixing ratio, and horizontal winds in the lower troposphere.
The cloud-base mass flux is controlled by turbulent kinetic energy within the sub-cloud layer and
convective inhibition. The cloud-base height for shallow cumulus is set between the lifting
condensation level and the boundary layer top, which is diagnosed based on the vertical profile of
relative humidity. When implementing the parameterization in MIROC6, the following conditions for
triggering the shallow convection are specified: 1) The estimated inversion strength (Wood and
Bretherton, 2006) is smaller than a tuning parameter, and 2) the convection depth diagnosed by a
separate cumulus convection scheme (Chikira and Sugiyama, 2010) is smaller than a tuning parameter.

The Spectral Radiation-Transport Model for Aerosol Species (SPRINTARS; Takemura et

al., 2000, 2005, 2009) is used as an aerosol module for MIROC6 to predict the mass mixing ratios of
the main tropospheric aerosols which are black carbon, organic matter, sulfate, soil dust, and sea salt,
and the precursor gases of sulfate (sulfur dioxide, $SO_2$, and dimethylsulfide). By coupling the radiation
and cloud-precipitation schemes in MIROC, SPRINTARS calculates not only the aerosol transport
processes of emission, advection, diffusion, sulfur chemistry, wet deposition, dry deposition, and
gravitational settling, but also the aerosol-radiation and aerosol-cloud interactions. There are two
primary updates in SPRINTARS of MIROC6 that were not included in MIROC5. One is the treatment
of precursor gases of organic matters as prognostic variables. In the previous version, the conversion
rates from the precursor gases (e.g., terpene and isoprene) to organic matters are prescribed (Takemura
et al., 2000), while an explicit simplified scheme for secondary organic matters was introduced from
a global chemical climate model (Sudo et al., 2002). The other is a treatment of oceanic primary and
secondary organic matters. Emissions of primary organic matters are calculated with wind at a 10-m
height, the particle diameter of sea salt aerosols, and chlorophyll-*a* concentration at the ocean surface
(Gantt et al., 2011). The oceanic isoprene and monoterpene, which are precursor gases of organic
matters, are emitted depending on the photosynthetically active radiation, diffuse attenuation
coefficient at 490 nm, and the ocean surface chlorophyll-*a* concentration (Gantt et al., 2009).
The radiative transfer in MIROC6 is calculated by an updated version of the *k*-distribution
scheme used in MIROC5 (Sekiguchi and Nakajima 2008). The single scattering parameters have been
calculated and tabulated in advance, and liquid, ice, and five aerosol species can be treated in this
updated version. Given the significant effect of crystal habit on a particle's optical characteristics
(Baran, 2012), the assumption of ice particles habit has been updated from our previous simple
assumption of sphere used in MIROC5 to a hexagonal solid column (Yang et al., 2013) in MIROC6.
The upper limits of the mode radius of cloud particles have been extended from 32 μm to 0.2 mm for
liquids and from 80 μm to 0.5 mm for ice. Therefore, the scheme can now handle the large-sized water
particles (e.g., drizzle and rain) that have been shown to have significant radiative impacts (Waliser et
al., 2011).
Following Hines (1997) and Watanabe et al. (2011), a non-orographic gravity wave
parameterization is newly implemented into MIROC6 in order to represent realistic large-scale
circulations and thermal structures in the stratosphere and mesosphere. Following Watanabe (2008), a
present-day climatological source of non-orographic gravity waves, which is estimated using results
of a gravity wave-resolving version of MIROC-AGCM (Watanabe et al., 2008), is launched at the 70
hPa level in the extratropics, while an isotropic source of non-orographic gravity waves is launched at
the 650 hPa level in the tropics. Together with this parameterization, an orographic gravity wave
parameterization of McFarlane (1987) is also adopted as in MIROC5. In both the orographic and non-
orographic gravity wave parametrizations, wave source parameters at launch levels are tuned so that
the realistic seasonal progress of the middle atmosphere circulations, frequency of sudden
stratospheric warmings, and period and amplitude of the equatorial quasi-biennial oscillations (QBOs)
can be represented.

**2.2 Land surface component**
The land surface model is also basically the same as in MIROC5. Energy and water
exchanges between land and atmosphere are calculated, considering the physical and physiological
effects of vegetation with a single layer canopy, and the thermal and hydrological effects of snow and
soil respectively with a three-layers snow and a six-layers soil down to a 14 m depth. Sub-grid fractions
of land use and snow cover have also been considered. The timestep for the land surface model
integration is 1 hour in MIROC6 which is the same as in MIROC5. In addition to the standard package
in MIROC5, a few other physical parameterizations are implemented as described below.
A physically-based parameterization of sub-grid snow distribution (SSNOWD; Liston,
2004; Nitta et al., 2014) replaces the simple functional approach of snow water equivalent in
calculating sub-grid snow fractions in MIROC5 in order to improve seasonal cycle of snow cover. In
SSNOWD, the snow cover fraction is formulated for accumulation and ablation seasons separately.
For the ablation season, the snow cover fraction decreases based on the sub-grid distribution of the
snow water equivalent. A lognormal distribution function is assumed and the coefficient of variation
category is diagnosed from the standard deviation of the sub-grid topography, coldness index, and
vegetation type that is a proxy of surface winds. While the cold degree month was adopted for coldness
in the original SSNOWD, we decided instead to introduce the annually averaged temperature over the
latest 30 years using the time-relaxation method of Krinner et al. (2005), in which the timescale
parameter is set to 16 years. The temperature threshold for a category diagnosis is set to 0°C and 10°C.
In addition, a scheme representing a snow-fed wetland that takes into consideration sub-grid terrain
complexity (Nitta et al., 2017) is incorporated. The river routing model and lake module are the same
as those used in MIROC5, but the river network map is updated to keep the consistency to the new
land-sea mask (Yamazaki et al., 2009).

**2.3 Ocean and sea-ice component**

The ocean component of MIROC6 is basically the same as that used in MIROC5, but

several updates are implemented as described below. The warped bipolar horizontal coordinate system
in MIROC5 has been replaced by the tripolar coordinate system proposed by Murray (1996). Two
singular points in the bipolar region to the north of about 63°N are placed at (63°N, 60°E) in Canada
and (63°N, 120°W) in Siberia (Fig. 2). In the spherical coordinate portion to the south of 63°N, the
longitudinal grid spacing is 1° and the meridional grid spacing varies from about 0.5° near the equator
to 1° in the mid-latitudes. In the central Arctic Ocean where the bipole coordinate system is applied,
the grid spacings are about 60 km in zonal and 33 km in meridional, respectively. By introducing the
horizontal tripolar coordinate system, it is expected that theoretical westward propagation of the
oceanic baroclinic Rossby can be represented with less numerical dispersions because of agreement
of the coordinate system and the geographical coordinate system and that the horizontal resolutions in
the Arctic Ocean where the Rossby radius of deformation is relatively small are higher than in the case
where the bipolar warped coordinate system in MIROC5 is adopted. There are 62 vertical levels in a
hybrid $\sigma$-z coordinate system. The horizontal grid spacing in MIROC5 is nominally 1.4°, except for
the equatorial region and there are 49 vertical levels. The resolutions in MIROC6 are higher than in
MIROC5. In particular, 31 (23) of the 62 (49) vertical layers in MIROC6 (MIROC5) are within the
upper 500 m depth (Fig. 1b). The increased vertical layers in MIROC6 have been adopted in order to
better represent the equatorial thermocline and observed complex hydrography in the Arctic Ocean.
An increase in computational costs of the ocean component due to higher resolutions in MIROC6 is
suppressed by implementing a time-staggered scheme for the tracer and baroclinic momentum
equations (Griffies et al., 2005). Owing to the time-staggered scheme, the timestep for the ocean and
sea-ice components of MIROC6 is 20 minutes which is longer than that in MIROC5 (15 minutes).

The tracer advection scheme (Prather, 1986), the surface mixed layer parameterization

(Noh and Kim, 1999), and the parameterization for eddy isopycnal diffusion (Gent et al., 1995) used
in MIROC6 are the same as those used in MIROC5. Also as in MIROC5, the bottom boundary layer
parameterization of Nakano and Suginohara (2002) is introduced south (north) of 54°S (49°N) for
representing the down-sloping flow of dense waters. The constant parameters used in the above-
mentioned parameterizations are determined in the same manner as that of MIROC5, except for the
Arctic region. An empirical profile of background vertical diffusivity, which is proposed in Tsujino et
al. (2000), is modified above the 50 m depth to the north of 65°N. It is $1.0 \times 10^{-6} \, \text{m}^2 \, \text{s}^{-1}$ in the uppermost
29 m and gradually increases to $1.0 \times 10^{-5} \, \text{m}^2 \, \text{s}^{-1}$ at the 50 m depth. Additionally, the turbulent mixing
process in the surface mixed layer is changed so that there is no surface wave breaking and no resultant
near-surface mixing in regions covered by sea ice. The combination of the weak background vertical
diffusivity and suppression of turbulent mixing under the sea-ice contributes to better representations
of the surface stratification in the Arctic Ocean with little impact on the rest of the global oceans
(Komuro, 2014).

The sea-ice component in MIROC6 is almost the same as in MIROC5. A brief description,

along with some major parameters, is given here. Readers may refer to Komuro et al. (2012) and
Komuro and Suzuki (2013) for further details. A subgrid-scale sea-ice thickness distribution is
incorporated by following Bitz et al. (2001). There are five ice categories (plus one additional category
for open water), and the lower bounds of the ice thickness for these categories are set to 0.3, 0.6, 1,
2.5, and 5 m. The momentum equation for sea-ice dynamics is solved using elastic-viscous-plastic
rheology (Hunke and Dukowicz, 1997). The strength of the ice per unit thickness and concentration is
set at $2.0 \times 10^4$ N m$^{-2}$, and the ice–ocean drag coefficient is set to 0.02. The surface albedo for bare ice
surface is 0.85 (0.65) for the visible (infrared) radiation. The surface albedo in snow-covered areas is
0.95 (0.80) when the surface temperature is lower than -5ºC for the visible (infrared) radiation, and it
is 0.85 (0.65) when the temperature is 0ºC. Note that the albedo changes linearly between -5ºC and
0ºC. These parameter values listed here are the same as those listed in MIROC5.

**2.4 Boundary conditions**

A set of external forcing data recommended by the CMIP6 protocol are used. The historical

solar irradiance spectra, greenhouse gas concentrations, anthropogenic aerosol emissions, and biomass
burning emissions are given by Matthes et al. (2017), Meinshausen et al. (2017), Hoesly et al. (2018),
and van Marle et al. (2017), respectively. The concentrations of greenhouse gases averaged globally
and annually are given to MIROC6. Radiative forcing of stratospheric aerosols due to volcanic
eruptions are computed by vertically integrating extinction coefficients for each radiation band, which
are provided by Thomason et al. (2016), in the model layers above the tropopause. Three-dimensional
atmospheric concentrations of historical ozone ($O_3$) are produced by the Chemistry-Climate Model
Initiative    (Hegglin    et    al.,    in    preparation;    the    data    are    available    at
http://blogs.reading.ac.uk/ccmi/forcing-databases-in-support-of-cmip6/).         Three        dimensional
concentrations of the OH radical, hydrogen peroxide ($H_2O_2$) and Nitrate ($NO_3$) are precalculated by a
chemical atmospheric model of Sudo et al. (2002). As precursors of secondary organic aerosol,
emission data of terpenes and isoprene provided by the Global Emissions Inventory Activity (Guenther
et al., 1995) are normally used, although simulated emissions from the land ecosystem model of Ito
and Inatmoni (2012) are also used alternatively.

For specifying the soil types and area fractions of natural vegetation and crop-land on grids

of the land-surface component, the harmonized land-use dataset (Hurtt et al., in prep.), Center for
Sustainability and the Global Environment global potential vegetation dataset (Ramankutty and Foley,
1999), and the dataset provided by the International Satellite Land Surface Climatology Project
Initiative I (Sellers et al., 1996) are used. These datasets are also used in prescribing background
reflectance at the land surface. Leaf-area index data are prepared based on the Moderate Resolution
Imaging Spectroradiometer Leaf-area index products of Myneni et al. (2002).

The forcing dataset used for the preindustrial control simulation is basically composed of

the data for the year 1850, which are included in the above-mentioned historical dataset. The
stratospheric aerosols and solar irradiance in the preindustrial simulation are given as monthly
climatology averaged in 1850 − 2014 and in 1850 − 1873, respectively. The total solar irradiance is
about 1361 $Wm^{-2}$, and the global-mean concentrations of $CO_2$, methan ($CH_4$), and nitrous oxide ($N_2O$)
are 284.32 ppm, 808.25 ppb, and 273.02 ppb, respectively.

**2.5 Spin-up and tuning procedures**

Firstly, the stand-alone ocean component of MIROC6, which includes the sea-ice

processes, is integrated from the initial motionless state with the observed temperature and salinity
distribution of the Polar Science Center hydrographic climatology (Steele et al., 2001). Ocean model
coastline geometry and bottom bathymetry are specified based on horizontal interpolation of the land
and sea-floor dataset of ETOPO5 (National Geophysical Data Center, 1993). The ocean component is
spun-up for 1000 years by the monthly climatological surface fluxes of Röske (2006). An acceleration
method of Bryan (1984) is used in the spin-up stage in order to obtain a thermally and dynamically
quasi-steady state. After the spin-up, additional integration for 200 years is performed without the
acceleration method. By analyzing the last 50-yr-long data from the stand-alone ocean component, the
monthly climatology of typical variables (e.g., zonal-mean temperature and salinity in several basins,
volume transports across major straits and archipelagos, meridional overturning circulations, and sea-
ice distributions) are compared with observations. Once the configuration of the ocean component is
frozen, the land-sea distribution and land-sea area ratios on the atmospheric and land surface model
grids are determined according the coastline geometry of the ocean component, after which the
atmospheric and the land surface components are coupled with the ocean component. Surface
topography in the atmospheric and land surface component are also made using the ETOPO5 dataset.
Note that horizontal grid arrangement of the land surface model is exactly same as the atmospheric
component. The coupling interval among the sub-models is 1 hour. An initial condition of the ocean
component in MIROC6 is given by the stand-alone ocean experiment, and those of the atmosphere
and land are taken from an arbitrary year of the pre-industrial control run of MIROC5.

After coupling the sub-models, climate model tuning is done under the pre-industrial

boundary conditions. Conventionally, the climate models of our modeling community are retuned in
coupled modes after stand-alone sub-model tuning. This is because reproducibility of climatic-mean
state and internal climate variations is not necessarily guaranteed in climate models with the same
parameters determined in stand-alone sub-model tuning, which is particularly the case in the tropical
climate. In our tuning procedures described below, many of the 10-yr-long climate model runs are
conducted with different parameter values. There are numerous parameters associated with physical
parameterizations, whose upper/lower bounds are constrained by empirical or physical reasoning. The
main parameters used in our tuning procedures are chosen referring to a perturbed parameter ensemble
set made by Shiogama et al. (2012) in which parameter sensitivity to cloud-radiative processes is
examined. The impact of parameter tuning on the present climate is also discussed by Ogura et al
(2017), focusing on the top-of-the-atmosphere (TOA) radiation and clouds. Any objective and optimal
methods for parameter tuning are not used in our modeling group and the tuning procedures are like
those in other climate modeling groups as summarized in Hourdin et al. (2017).

In the first model tuning step, climatology, seasonal progression, and internal climate

variability in the tropical coupled system are tuned in order that departures from observations or
reanalysis datasets are reduced. Here, it should be noted that representation of the tropical system in
MIROC6 is sensitive to the parameters for convections and planetary boundary layer processes.
Specifically, parameters of reference height for cumulus precipitation, efficiency of the cumulus
entrainment of surrounding environment and maximum cumulus updraft velocity at the cumulus base
are used to tune strength of the equatorial trade wind, climatological position and intensity of the Inter-
Tropical Convergence Zone (ITCZ) and South Pacific Convergence Zone (SPCZ), and interannual
variability of El-Niño/Southern Oscillation (ENSO). In particular, the parameter for the cumulus
entrainment is known as a controlling factor of ENSO in MIROC5 (Watanabe et al., 2011).
Summertime precipitation in the western tropical Pacific and characteristic of tropical intraseasonal
oscillations are tuned by using the parameter for shallow convection describing the partitioning of
turbulent kinetic energy between horizontal and vertical motions at the sub-cloud layer inversion. Next,
the wintertime mid-latitude westerly jets and the stationary waves in the troposphere are tuned using
the parameters of the orographic gravity wave drag and the hyper diffusion of momentum. The
parameters of the hyper diffusion and the non-orographic gravity wave drag are also used when tuning
stratospheric circulations of the polar vortex and QBO. Finally, the radiation budget at the TOA is
tuned, primarily using the parameters for the auto-conversion process so that excess downward
radiation can be minimized and maintained closer to 0.0 $\text{Wm}^{-2}$. The surface albedos for bare sea-ice
and snow-covered sea-ice are set to higher values than in observations (see Section 2.3) in order to
avoid underestimating of the summertime sea-ice extent in the Arctic Ocean due to excess downward
shortwave radiation in this region. In addition, parameter tuning for the total radiative forcing
associated with aerosol-radiation and aerosol-cloud interactions is done. In order that the total radiative
forcing can be closer to the estimate of -0.9 $Wm^{-2}$ (IPCC, 2013; negative value indicates cooling) with
an uncertainty range of -1.9 to -0.1 $Wm^{-2}$, parameters of cloud microphysics and the aerosol transport
module, such as timescale for cloud droplet nucleation, in-cloud properties of aerosol removal by
precipitation, and minimum threshold of number concentration of cloud droplets, are perturbed. To
determine a suitable parameter set, several pairs of a present-day run under the anthropogenic aerosol
emissions at the year 2000 and a pre-industrial run are conducted. A pair of the present and
preindustrial runs has exactly the same parameters, and differences of tropospheric radiations between
two runs are considered as anthropogenic radiative forcing. Note that MIROC6 in a coupled mode is
used in this tuning procedure, and thus the sea surface temperature (SST) is not fixed. The estimated
radiative forcing here is not strictly the same as the effective radiative forcing estimated in IPCC
(2013). However, by the present tuning procedure, the global-mean surface air temperature (SAT)
change after the mid-19th century is well reproduced in the historical runs by MIROC6 (details are
discussed in Section 4). As above-mentioned, reproducibility of the global-mean SAT is not a tuning
goal but is a typical metric which reflects results of the parameter tunings for individual processes of
convections, dynamics, and radiative forcing.
After fixing the model parameters, the climate model is spun-up for 2000 years. During
the first several hundred years, waters contained in the land surface are drained to the ocean via river
runoff, which leads to a temporal weakening of the meridional overturning circulations in the ocean
and a rising of the global-mean sea level. After the global hydrological cycle reaches to an equilibrium
state, the strengths of the meridional overturning circulations recover and keep quasi steady state. The
above-mentioned processes spend about 1000 years, after which an additional 1000-yr-long
integration is performed in order to obtain a thermally and dynamically quasi-steady ocean state.

Figure 3 shows the time series of the global-mean quantities after the spin-up. The labeled

year in Fig. 3 indicates the elapsed year after the spin-up duration of 2000 years. Linear trend of the
global-mean SAT is $9.5 \times 10^{-3}$ K/100 yr and is much smaller than the observed value of about 0.62
K/100 yr in the twentieth century, indicating that there is no significant drift and the global-mean SAT
is in a quasi-steady state. While the global-mean SST is in a quasi-steady state (linear trend of $7.0 \times$
$10^{-3}$ K/100 yr), the global-mean ocean temperature shows a larger trend of $6.8 \times 10^{-3}$ K/100 yr in the
first 500 years than that of $1.3 \times 10^{-3}$ K/100 yr in the later period. In the later sections, the 200-yr-long
data between the 500-th and 699-th years are analyzed.

The trend of the global-mean ocean temperature in the later period suggests slight but

continuous warming of the deep ocean. The radiation budget at the TOA is 1.1 $Wm^{-2}$ downward on
average (linear trend of $9.5 \times 10^{-3}$ K/100 yr) and the net heat input at the sea surface is 0.32 $Wm^{-2}$. The
deep ocean warming is explained by the net heat input. Note that there is about 0.78 $Wm^{-2}$
inconsistency between the TOA radiation budget and the ocean heat uptake. This heat energy
inconsistency is due to that internal energy associated with precipitation, water vapor and river runoff
is not taken account in the atmospheric and land surface component in MIROC6 and that these waters
with no temperature information implicitly set their temperature to the SST when they flow or fall into
the ocean. Perpetual melting of the prescribed Antarctic ice-sheet with invariant ice thickness, which
is occurred due to the warm SAT bias in the Antarctic region (details will be discussed in Section
3.1.3), is also a cause of the heat energy inconsistency.

**3 Results of pre-industrial simulation**

Representations of climatic-mean field and internal climate variability in MIROC6 are

evaluated in comparison with MIROC5 and observations. The 200-yr-long data of the preindustrial
control simulation by MIROC5 are used. The observations and reanalysis datasets used in the
comparison are listed in Table 1.
Here, the model climatology in the pre-industrial simulations is compared with
observations in the recent decades. Because observations are obtained concurrently with the progress
of the global-warming due to increasing anthropogenic radiative forcing, the model climate under the
pre-industrial conditions may not be adequate for use when making comparisons with recent
observations. However, the root-mean-squared (RMS) errors of typical variables (e.g., the global-
mean SAT) in the climate models with respect to observations are much larger than the RMS
differences between the model climatology in the pre-industrial simulation and those in the last 30-yr-
long period in the historical simulations. Therefore, the era differences where climatology is defined
are not significant concern in comparisons among the climate models and observations.

**3.1 Climatology**
**3.1.1 Atmosphere and Land-surface**
First, model systematic biases in radiations at the TOA are evaluated because they reflect
model deficiencies in cloud-radiative processes that contribute to a large degree of uncertainty in
climate modelling. Figure 4 shows annual-mean biases in radiative fluxes at the TOA in MIROC6 and
MIROC5 with respect to the recent Clouds and the Earth's Radiant Energy System (CERES) estimate
(Loeb et al., 2009; the data are available at https://ceres.larc.nasa.gov/). At the top-right of each panel,
a global-mean (GM) value and a root-mean-squared error (RMSE) with respect to observations are
written. In the present manuscript, RMSE is computed without model and observed global-mean
quantities unless otherwise noted.
Persistent overestimates of net shortwave radiative flux and the sum of net shortwave and
net longwave fluxes over low-latitude oceans in MIROC5 are significantly reduced in MIROC6.
Hereafter, net shortwave, longwave, and the sum of them are denoted as OSR, OLR and NET,
respectively, for simplicity. As described in Ogura et al. (2017), since parameter tuning cannot
eliminate the above-mentioned excess upward radiations, it is suggested that implementing a shallow
convective parameterization is required in order to reduce the biases. Figure 5 shows annual-mean
moistening rates associated with deep and shallow convections at the 850 hPa pressure level in
MIROC6. Moistening due to shallow convections occurs mainly over the low-latitude oceans,
especially the eastern subtropical Pacific and the western Atlantic and Indian oceans. These active
regions of shallow convections occur separately from regions with active deep convections in the
western tropical Pacific and the ITCZ. The clear separation of the two convection types is consistent
with satellite-based observations (Williams and Tselioudis, 2007). Owing to the shallow convective
process that mixes the humid air in the planetary boundary layer with the dry air in the free troposphere,
low-level cloud cover over the low-latitude oceans is better represented in MIROC6 than in MIROC5.
Figure 6 shows annual-mean biases in cloud covers with respect to the International Satellite Cloud
Climatology Project (ISCCP; Rosso et al., 1996; Zhang et al., 2004; the data are available at
https://isccp.giss.nasa.gov/). Overestimate of low-level cloud cover over the low-latitude oceans in
MIROC5 (Fig. 6b) is apparently reduced in MIROC6 (Fig. 6a), which results in the smaller biases in
NET and OSR biases (Fig. 4). RMS error in low-level cloud cover in MIROC6 is 9% lower than that
in MIROC5.

OSR in the mid-latitudes are also better represented in MIROC6 than in MIROC5. Zonally

distributed downward OSR bias in MIROC5 is reduced or becomes a relatively small upward bias in
MIROC6 (Figs. 4cd). This difference in the OSR bias is commonly found in both hemispheres. Cloud
covers at middle and high levels are larger in MIROC6 over the subarctic North Pacific, North Atlantic,
and the Southern Ocean (Figs. 6c-f), while low-level cloud cover over the same regions is smaller in
MIROC6 than in MIROC5 over the same regions (Figs. 6ab). The smaller low-level cloud cover in
MIROC6 is inconsistent with the larger upward OSR bias in MIROC6. The wintertime mid-latitude
westerlies are stronger and are located more poleward in MIROC6 than in MIROC5. Correspondingly,
activity of sub-weekly disturbances in the mid-latitudes is strengthened in MIROC6 (details are
described later). These differences in the mid-latitude atmospheric circulations between MIROC6 and
MIROC5 lead to an enhanced poleward moist air transport from the subtropics to the subarctic region,
which could result in an increase in the mid- and high-level cloud covers in MIROC6, as reported in
previous modeling studies (e.g., Bodas-Salcedo et al., 2012; Williams et al., 2013). Consequently, the
downward OSR bias in the mid-latitudes is smaller in MIROC6 than in MIROC5. In polar regions,
both biases in OSR and NET remain the same as in MIROC5.

Systematic bias in the outgoing longwave radiative flux (hereafter, OLR) is worse in

MIROC6 than in MIROC5 because MIROC6 tends to underestimate OLR over almost the entire
global domain, except for Antarctica (Figs. 4ef). The global-mean of the high-level cloud cover in
MIROC6 is larger than in MIROC5 by 0.04 (Figs. 6ef), which is consistent with the smaller OLR in
MIROC6. The increased moisture transport due to the strengthening of the westerlies and sub-weekly
disturbances can partly explain the increase in the mid-latitude high-level clouds in MIROC6, but
high-level cloud cover is also larger in the low-latitudes. Hirota et al. (2018) reported that moistening
of the free troposphere due to shallow convections creates favorable conditions for atmospheric
instabilities that leads to the resultant activation of deep convections in the low-latitudes. Such
processes may contribute to the inferior representation of OLR in MIROC6.

Next, we will discuss on the global budget of the radiative fluxes and the RMS errors

between models and observations. Note that only deviations from the global means are considered
when calculating RMS errors. As written on the upper right of panels in Fig. 4ab, the global-mean
(RMS errors) NETs are -1.11 (12.7) $Wm^{-2}$ in MIROC6 and -0.98 (15.9) $Wm^{-2}$ in MIROC5, respectively,
and these values are consistent with the observed value of -0.81 $Wm^{-2}$ (CERES; Loeb et al, 2009).
However, the observed value is estimated in the present-day condition. Ideally, the model value in the
preindustrial condition should be 0 $Wm^{-2}$ and is in the marginally acceptable range. If NET is divided
into OSR and OLR, so-called error compensation becomes apparent. The global means of OSR (OLR)
are -231.3 (230.2) $Wm^{-2}$ in MIROC6 and -237.6 (236.6) $Wm^{-2}$ in MIROC5, respectively (Figs. 4c-f).
The observed global-means of OSR and OLR are -240.5 $Wm^{-2}$ and 239.7 $Wm^{-2}$. Biases in the global-
mean OSR (OLR) with respect to observations are 9.2 (-9.5) $Wm^{-2}$ in MIROC6 and 2.9 (-3.1) $Wm^{-2}$
in MIROC5, respectively. Thus, the global-mean OSR and OLR in MIROC6 are worse than those in
MIROC5. Further division of OSR and OLR into cloud-radiative forcing and clear-sky shortwave
(longwave) radiative components shows that shortwave cloud-radiative forcing is dominant on the
biases in radiative fluxes. The biases in the global-mean shortwave (longwave) cloud-radiative forcing
with respect to observations are 12.0 (6.7) $Wm^{-2}$ in MIROC6 and -4.0 (-0.2) $Wm^{-2}$ in MIROC5,
respectively.

The global radiation budget in MIROC6 is inferior to that in MIROC5, while

reproducibility of climatic means of typical model variables, other than radiative fluxes, and internal
variations are better simulated in MIROC6 (details are shown later). As described in Section 2.5, the
intensive tuning by perturbing model parameters is done focusing on reproducibility of climatic means,
internal variations, and radiative forcing due to anthropogenic aerosols. During this procedure, the
global radiation budget is traded-off. On the other hand, RMS errors in NET, OSR, and OLR are 12.7,
16.2, and 6.3 $Wm^{-2}$ in MIROC6 and 15.9, 18.9, and 6.8 $Wm^{-2}$ in MIROC5, respectively, thereby
indicating that the errors in MIROC6 have been reduced by 7% to 20 %. This is also the case for
shortwave and longwave cloud radiative forcings, where the corresponding errors have been reduced
by 17% and 13 %, respectively. Taken togthher, these results show that the spatial patterns of the
radiative fluxes are better simulated in MIROC6 than in MIROC5.

The improvement in spatial radiation patterns, especially in low-latitude OSR, is explained

primarily by the implementation of shallow convective processes, which results in a moister free
troposphere in MIROC6 than in MIROC5. Figures 7ab show zonal-mean biases in annual-mean
specific humidity with respect to the European Centre for Medium-Range Weather Forecast interim
reanalysis (ERA-I; Dee et al., 2011; the data are available at
https://www.ecmwf.int/en/forecasts/datasets/archive-datasets/reanalysis-datasets/era-interim). Dry
bias in 30ºS–30ºN, which occurs persistently in MIROC5, are largely reduced in MIROC6 owing to
vertical mixing at the interface of the planetary boundary layer and the free troposphere. On the other
hand, moist bias below the 600 hPa pressure level in the mid-latitudes is somewhat worse in MIROC6
than in MIROC5. Shallow convections also contribute to the improvement of precipitations in the low
latitudes. Figure 8 shows global maps for climatological precipitation in boreal winter (December–
February) and summer (June–August). The second version of the Global Precipitation Climatology
Project (GPCP; the data are available at https://precip.gsfc.nasa.gov/) Monthly Precipitation Analysis
(Adler et al., 2003) is used for the observations. While MIROC5 suffers from underestimate of
summertime precipitation over the western tropical Pacific, the underestimate is largely reduced in
MIROC6 (Figs. 8df). The increase of precipitations is associated with deep convections because the
moister free troposphere in MIROC6 is more favorable for the occurrence of deep convections (Hirota
et al., 2018). On the other hand, model representation of the precipitation in MIROC6 is not necessarily
alleviated other than the western tropical Pacific. For example, the overestimate of wintertime
precipitation over the Indian Ocean and the mid-latitude North Pacific is worse in MIROC6 than in
MIROC5.

Zonal-mean biases in annual-mean air temperature and zonal wind velocity are also better

represented in MIROC6 than in MIROC5 (Figs. 7c-f). The upper stratospheric warm bias in 50ºS–
50ºN in MIROC5 is significantly reduced in MIROC6. The model top of MIROC6 is located at the
0.004 hPa pressure level and there are 42 vertical layers above the 50 hPa pressure level, while the
model top of MIROC5 is placed at the 3 hPa pressure level. As a result, there are significant differences
in stratospheric circulations between the models. As shown in the annual-mean mass stream function
calculated using zonal-mean meridional winds (Fig. 9), an upward wind continuing from the low-
latitude troposphere to the stratosphere is stronger in MIROC6 than in MIROC5. It is considered that
an increased upward advection of the temperature minimum around the tropopause in 30ºS–30ºN may
lead to reduction of warm temperature bias in the stratosphere which is significant in MIROC5.
Correspondingly, the stratospheric westerly bias in low latitudes of MIROC5 is also considerably
alleviated in MIROC6. Note that the atmospheric $O_3$ concentration data used in MIROC5 is different
from those in MIROC6, and the concentration in the stratosphere is higher than the data used in
MIROC6. About 25% of the above-mentioned reduction in the stratospheric warm biases is explained
by the smaller absorption of shortwave radiation by $O_3$. Note that the zonal-mean temperature bias in
Fig. 7c is smaller when the climatological-mean temperature from 1980 to 2009 in a historical
simulation are evaluated against observations because of the known stratospheric cooling with
increased greenhouse gases and reduced $O_3$ concentrations.

The zonal-means of the air temperature and zonal wind in MIROC6 are also better

simulated in the mid- and high latitudes. A pair of easterly and westerly biases in MIROC5, which is
in the troposphere of the Northern Hemisphere, is associated with a weaker mid-latitude westerly jet
and its southward shift with respect to observations. The pair of the biases is reduced in MIROC6,
thereby suggesting that a strengthening and northward shift of the westerly jet occurs in MIROC6.
Indeed, as shown in the upper panels of Fig. 10, the meridional contrast of high and low biases in the
500 hPa pressure level (Z500) along the wintertime westerly jet is weaker in MIROC6 than in
MIROC5. The latitudes with the maximal meridional gradient of Z500 are located further northward
in MIROC6 than in MIROC5, especially over the North Atlantic. Correspondingly, wintertime storm
track activity (STA), which is defined as an 8-day-high-pass-filtered eddy meridional temperature flux
at the 850 hPa pressure level, is stronger over the North Pacific and Atlantic in MIROC6 than in
MIROC5 (see upper panels of Fig. 11) and is accompanied by an associated increase in precipitation,
especially in the North Pacific (Figs. 8ce). In the stratosphere above the 10 hPa pressure level, the
polar night jet is reasonably captured in MIROC6, although the westerly is somewhat overestimated
in 30ºN–60ºN. Also, in the Southern Hemisphere, representation of the tropospheric westerly and the
polar night jets are better in MIROC6 than in MIROC5, and the easterly bias centered at 60ºS in the
troposphere is clearly reduced in MIROC6. Although causality is unclear, the warm air temperature
bias above the tropopause to the south of 60ºS is smaller in MIROC6 than in MIROC5.

The enhanced wintertime STA in MIROC6 leads to a strengthening of the Ferrel circulation

in the Northern Hemisphere and a broadening of its meridional width. As shown in Fig. 9, the northern
edge of the Ferrel cell is located further northward in MIROC6 than in MIROC5. Because the Ferrel
cell is a thermally indirect circulation driven primarily by eddy temperature and momentum fluxes,
the stronger STA in MIROC6 possibly causes the Ferrel cell differences between the two models.
Associated with the northward extension of the Ferrel cell, the upward wind between the Ferrel cell
and the polar cell centered at 65ºN is stronger in MIROC6 than in MIROC5 and the meridional width
of the polar cell is smaller. Also, in the Southern Hemisphere, the upward wind around 60ºS at the
southern edge of the Ferrel cell is stronger in MIROC6 than in MIROC5. Correspondingly, high sea
level pressure (SLP) biases in polar region in MIROC5 are significantly reduced in MIROC6 (figures
are omitted) and RMS errors with respect to observations (ERA-I) are decreased by 30 %. Meanwhile,
in the stratosphere, anti-clockwise (clockwise) circulations to the north (south) of 50ºN (S) are stronger
and extend further upward in MIROC6 than in MIROC5. These circulations seem to continue from
the troposphere into the stratosphere, thereby implying that more active troposphere-stratosphere
interactions associated with wave-coupling exist in MIROC6. Further details will be described later,
focusing on the occurrence of the sudden stratospheric warmings.

Parameterizations of SSNOWD (Liston, 2004; Nitta et al., 2014) and a wetland due to

snow-melting water have been newly implemented into MIROC6 (Nitta et al., 2017). In comparison
of MIROC6 with MIROC5, it can be seen that the former parameterization brings about significant
improvement in the Northern Hemisphere snow cover fractions from the early to the late winter (Fig.
12). Compared with observations of the Northern Hemisphere EASE-Grid 2.0 (Brodzik and
Armstrong, 2013; the data are available at https://nsidc.org/data/ease/), the distribution of the snow
cover fractions is more realistic in MIROC6 than MIROC5, especially where and when the snow water
equivalent is relatively small (e.g., mid- and high latitudes in November, over Siberia in February).
Note that no clear improvement is found in May. This is because the newly implemented SSNOWD
represents hysteresis in the snow water equivalent-snow cover fraction relationship in both the
accumulation and ablation seasons. MIROC6 underestimates the snow cover fraction in the partially
snow-covered regions and overestimates it on the Tibetan plateau and in some parts of China. We note
that meteorological (e.g., precipitation or temperature) phenomena might affect these biases, but
further investigation will be necessary to identify their causes. Nevertheless, in spite of those
discrepancies, it can be said that the seasonal changes of the snow cover fraction are better simulated
in MIROC6 than in MIROC5 (Fig. 12j).

**3.1.2 Ocean**

Next, we evaluate the climatological fields of the ocean hydrographic structure, meridional

overturning circulations (MOCs), and sea-ice distribution. The zonal-mean potential temperature and
salinity are displayed in Figs. 13 and 14, respectively. Both MIROC6 and MIROC5 capture the general
features of the observed climatological hydrography (ProjD; Ishii et al., 2003). However, the potential
temperatures in the deep and bottom layers to the south of 60ºS in the two models are warmer than
observations because of insufficient formation and sinking of cold and dense water due to intense
surface cooling around Antarctica (Figs. 13a-c and 14a-c). Such warm temperature bias associated
with deep water formation is also found in northern high latitudes of the Atlantic sector (Figs. 13a-c).
By horizontal advection of the warm temperature biases associated with the Pacific and Atlantic
MOCs, the model temperatures in deep layers apart from polar regions are also warmer than in
observations. The warm potential temperature bias in the deep layer is worse in MIROC6 than in
MIROC5 in both of the Atlantic and Pacific sectors and the warm bias influences the subsurface and
the intermediate layers above the 3000 m depth, which might be attributed to the excess ocean heat
uptake and longer integration time in MIROC6 than in MIROC5 (the spinup duration of MIROC6 is
2000 years and that of MIROC5 is about 1000 years, repectively). Also the low salinity bias below
the 2000 m depth is worse in MIROC6 than in MIROC5, especially in the Pacific sector (Figs. 14ef).
This worsening can be explained the excess supply of the freshwater in the Southern Ocean and weaker
northward intrusion of the less saline water in MIROC6.
In the Arctic Ocean, the halocline above the upper 500 m depth is sharper and more realistic
in MIROC6 than in MIROC5 and the high salinity bias below the 500 m depth in MIROC5 is alleviated
in MIROC6 (Figs. 13ef) because, as described in Section 2.3, there are many more vertical levels in
the surface and subsurface layers of MIROC6. In addition, vertical diffusivity in the Arctic Ocean is
set to smaller values in MIROC6 than in MIROC5, and the turbulent kinetic energy input induced by
surface wave breaking, as a function of the sea-ice concentration in each grid cell, is reduced in
MIROC6, as shown in Komuro (2014). In the North Pacific, the southward intrusion of North Pacific
Intermediate Water (NPIW) around the 1000 m depth retreats northward in MIROC6. Strong tide-
induced vertical mixing of sea water is observed along the Kuril Islands (e.g., Katsumata et al., 2004).
The locally enhanced tide-induced mixing is known to reinforce the southward intrusion of the
Oyashio and associated water mass transport from the subarctic to subtropical North Pacific, and to
feed the salinity minimum of NPIW (Nakamura et al., 2004; Tatebe and Yasuda, 2004). Hence,
NPIW reproducibility is better in MIROC5 where enhanced tidal mixing is considered than in
MIROC6. Because we encountered significant uncertainty in implementing the tidal mixing, we
decided to quit implementing it in developing phase of MIROC6, at the expense of NPIW
reproducibility.

The annual-mean potential temperature and zonal currents along the equator in MIROC6

are better simulated in MIROC6 than in MIROC5 (Fig. 15). Relatively cold water below the equatorial
thermocline is upwelled in MIROC6, especially in the eastern tropical Pacific, which leads to a
strengthening of the vertical temperature gradient across the thermocline. The eastward speed of the
Equatorial Undercurrent in MIROC6 is over 80 cm s$^{-1}$, and is closer to the products of Simple Ocean
Data Assimilation (SODA; Carton and Giese, 2008; the data are available at
http://www.atmos.umd.edu/~lchen/SODA3.3_Description.html) than in MIROC5. These
improvements are mainly attributed to the higher vertical resolution of MIROC6 in the surface and
subsurface layers. However, the thermocline depths in the western tropical Pacific are still larger in
the models than in observations and are attributed to the stronger trade winds in the models. When
both of MIROC6 and MIROC5 are executed as stand-alone AGCMs with the prescribed SST obtained
from observations, the overestimate of the equatorial trade winds also appears due to overestimate of
the upward winds over the maritime continent associated with deep cumulus convection and the
resultant strengthening of the Walker circulation over the equatorial Pacific. Better parameterizing
deep cumulus convection in the models could be required for better representation of the equatorial
trade winds and thus oceanic states.

Figure 16 displays annual-mean Atlantic and Pacific MOCs. In the Atlantic, two deep

circulation cells associated with North Atlantic Deep Water (NADW; upper cell) and Antarctic Bottom
Water (AABW, lower cell) are found in both of the models. NADW transport across 26.5ºN is 17.2
(17.6) Sv (1 Sv = 10$^6$ m$^3$ s$^{-1}$) in MIROC6 (MIROC5). These values are consistent with the
observational estimate of 17.2 Sv (McCarthy et al., 2015). RMS amplitudes of NADW transport are
about 0.9 Sv in MIROC6 and 1.1 Sv in MIROC5 on longer-than-interannual timescales, respectively.
These are smaller than the observed amplitude of 1.6 Sv in 2005–2014. Because observations include
the weakening trend of the Atlantic MOC due to the global warming, they can be larger than the model
variability under the preindustrial conditions. In the Pacific Ocean, both the models have the deep
circulation associated with Circumpolar Deep Water (CDW), but the northward transport of CDW
across 10ºS is 8.6 Sv in MIROC6, which is slightly larger than 7.5 Sv of MIROC5. Although these
models values are somewhat smaller than observations, they are within the uncertainty range of
observations (Talley et al., 2003; Kawabe and Fujio, 2010).

Northern Hemisphere sea-ice concentrations are shown in Fig. 17. Here, it can be seen that

both the March and September sea-ice distributions in MIROC6 resemble the satellite-based
observation (SSM/I; Cavarieli et al., 1991; the data are available at https://nsidc.org/). In general, the
spatial patterns of the models resemble the observations. Sea-ice areas in March (September) are 12.4
(6.1), 13.0 (6.9), and 14.9 (5.7) Million $km^2$ in MIROC6, MIROC5, and observations, respectively.
The model estimates are smaller (larger) in March (September) than in observations. The
underestimate in March is still found in MIROC6 and is attributed to the underestimate of sea-ice area
in the Sea of Okhotsk and the Gulf of St. Lawrence, even though the sea-ice area in the former region
is better simulated in MIROC6 than in MIROC5. Meanwhile, the eastward retreat of the sea-ice in the
Barents Sea is better represented in MIROC6 than in MIROC5. The overestimates in September in the
models are due to that the model climatology is defined under the pre-industrial conditions while
observations are taken in present-day conditions of 1980–2009, where a rapid decreasing trend of
summertime sea-ice area (including a few events of drastic decreases) is on-going (e.g., Comiso et al.,
2008). Note that the model September sea-ice area in 1980–2009 of historical simulations is smaller
than the observations and the sea-ice area does not show year-to-year drastic sea-ice decrease with
comparable amplitude with observations. The underestimate of the mean September sea-ice area in
MIROC6 might be attributed to slightly rapid warming of the Arctic climate in MIROC6 than in
observations. On the other hand, the modeled sea-ice areas in the Southern Ocean are unrealistically
smaller than in observations. Southern Hemisphere sea-ice areas in March (September) are 0.1 (3.4),
0.2 (5.2), and 5.0 (18.4) Million $km^2$ in MIROC6, MIROC5, and observations, respectively. Since
there are no significant differences between the two models, the spatial maps for the sea-ice area in
the southern hemisphere are omitted.

Figure 18 shows the global maps of annual-mean sea level height relative to the geoid. The

absolute dynamic height data provided by Archiving, Validation, and Interpretation of Satellite
Oceanographic (AVISO; Rio et al. 2014) data are used as observed sea level height (the data are
available at https://www.aviso.altimetry.fr/en/home.html). Overall oceanic gyre structures in the two
models are consistent with observations. Although representation of the gyres in MIROC6 remain
generally the same as in MIROC5, there are a few improvements in the North Pacific and the North
Atlantic. The mid-latitude westerly in MIROC6 is stronger and is shifted further northward than in
MIROC5 (Fig. 10), which results in the strengthening of the subtropical gyres, northward shifts of the
western boundary currents, and their extensions. In particular, the current speed of the Gulf Stream
and the North Atlantic Current are faster in MIROC6 than in MIROC5, and the contours emanating
from the North Atlantic reach the Barents Sea in MIROC6. A corresponding increase in warm water
transport from the North Atlantic to the Barents Sea leads to sea-ice melting and an eastward retreat
of the wintertime sea-ice there in MIROC6 (Figs. 17a-c). An improvement in MIROC6 is also found
in the Subtropical Countercurrent (STCC) in the North Pacific along 20ºN. As reported in Kubokawa
and Inui (1999), the low potential vorticity water associated with a wintertime mixed layer deepening
in the western boundary current region is transported southward in the subsurface layer and it pushes
up isopycnal surfaces around 25ºN. Thus, the eastward-flowing STCC is induced around 25ºN.
Although both of the models show the wintertime mixed layer deepening, the ocean stratification along
160ºE is weaker in MIROC6 than in MIROC5 (not shown). This suggests that the isopycnal advection
of low potential vorticity water in MIROC6 is more realistic than in MIROC5.

**3.1.3 Discussions on model climatological biases**
We have evaluated the simulated climatology in MIROC6 in comparison with MIROC5
and observations. The model climatology in MIROC6 shows certain improvements in simulating
radiations, atmospheric and oceanic circulations, and the snow cover fractions in the Northern
Hemisphere. In Fig. 19, we display the model biases in annual-mean SAT and SST (Fig. 19) because
these are typical variables that reflect errors in individual processes in the climate system. The global-
mean of SAT (SST) is 15.2 (18.1) ºC in MIROC6, 14.6 (18.0) ºC in MIROC5, and 14.4 (18.1) ºC in
observations. The modeled global-mean SATs and SSTs are generally consistent with observations.
However, since the observed (model) value is estimated in the present-day (preindustrial) condition,
the model global-mean SATs and SSTs are overestimated. Here, it should be noted that while the
spatial patterns of the SAT and SST biases in MIROC6 resemble those in MIROC5, there are several
improvements. For example, cold SAT bias in MIROC5 extending from the Barents Sea to Eurasia is
significantly smaller in MIROC6, possibly owing to the increase in warm water transport by the North
Atlantic Current and the resultant eastward retreat of the sea ice in the Barents Sea (Figs. 17 and 18).
Warm SAT and SST biases along the west coast of the North America are smaller in MIROC6 than in
MIROC5. The reason is that an increase of southeastward Ekman transport in the eastern subarctic
North Pacific due to the strengthening of the mid-latitude westerly jet (Fig. 10) and the Aleutian low
tend to cancel out the relatively warm water supply from the subtropics to the subarctic region by the
surface geostrophic current. Although it is not clear from Fig. 19, the SAT and SST in the subtropical
North Pacific around 20ºN are warmer by 2 K in MIROC6 than in MIROC5. Also in the Atlantic, the
SAT in the western tropics is warmer in MIROC6. These warmer surface temperatures in MIROC6
indicates a reduction of the cold SAT and SST biases that can be alleviated by an increase in the
downward OSR in MIROC6 due to the implementation of a shallow convective parameterization (Fig.
4), and by an increase in eastward transport of the warm pool temperature associated with the stronger
STCC in MIROC6 (Fig. 18).

On the other hand, the warm SAT and SST biases in the Southern Ocean and the warm

SAT bias in Middle East and the Mediterranean are worse in MIROC6 than in MIROC5. Consequently,
the RMS error in SAT is larger in MIROC6 (2.4 K) than in MIROC5 (2.2 K). The former is due
essentially to the underestimate of mid-level cloud covers, excess downward OSR, and the resultant
underestimate of the sea ice in the Southern Ocean. Such bias commonly occurs in many of climate
models and is normally attributed to errors in cloud radiative processes (e.g., Bodas-Salcedo et al.,
2012; Williams et al., 2013). In addition, poor representations of mixed layer depths and open ocean
deep convections due to the lack of mesoscale processes in the Antarctic Circumpolar Current are
causes of the warm bias (Olbers et al., 2004; Downes and Hogg, 2013). The latter warm bias, seen in
Middle East around the Mediterranean, can be explained by a tendency to underestimate the radiative
forcing of aerosol-radiation interactions due to underestimate of dust emissions from the Sahara Desert
in MIROC6 (not shown).

**3.2 Internal climate variations**
**3.2.1 Madden-Julian oscillation and East Asian Monsoon**

In this section, we will evaluate the reproducibility of internal climate variations in

MIROC6 in comparison with MIROC5 and observations, beginning with an examination of the
equatorial waves in the atmosphere. Zonal wavenumber–frequency power spectra normalized by
background spectra for the symmetric and antisymmetric components of OLR are calculated following
Wheeler and Kiladis (1999) and are shown in Fig. 20. The daily-mean OLR data derived from the
Advanced Very High-Resolution Radiometer (AVHRR) of the National Oceanic and Atmospheric
Administration (NOAA) satellites (Liebmann and Smith, 1996; the data are available at
https://www.esrl.noaa.gov/psd/data/gridded/data.interp_OLR.html) are used for observational
references. The signals corresponding to the Madden-Julian oscillation (MJO), equatorial Kelvin (EK),
equatorial Rossby (ER), eastward inertia-gravity (n=1 EIG), and westward inertia-gravity (WIG)
waves in the symmetric component and mixed Rossby-gravity (MRG) and eastward inertia-gravity
(n=0 EIG) waves in the antisymmetric component stand out from the background spectra in
observations. MIROC5 qualitatively reproduces these spectral maxima of the symmetric MJO, EK,
and ER qualitatively, while the amplitudes of the MJO and the EK are underestimated. These
underestimates are partially mitigated in MIROC6. The power summed over the eastward
wavenumber 1–3 and periods of 30–60 days corresponding to the MJO are 20% larger in MIROC6
than in MIROC5. Furthermore, some additional analyses indicate that many aspects of the MJO,
including its eastward propagation over the western tropical Pacific, are improved in MIROC6. Those
improvements are primarily associated with the implementation of the shallow convective scheme that
moistens the lower troposphere. The results of these additional analyses, along with some sensitivity
experiments, are described in a separate paper (Hirota et al., 2018). The EIG and WIG in the symmetric
component and the MRG and the EIG in the antisymmetric component are missing in both MIROC6
and MIROC5.

Figure 21 shows the June–August (JJA) climatology of precipitation and circulations in

the East Asia. As shown in observations (ERA-I; Fig. 21a), the East Asian summer monsoon (EASM)
is characterized by the monsoon low over the warmer Eurasian continent and the subtropical high over
the colder Pacific Ocean (e.g., Ninomiya and Akiyama, 1992). The southwesterly between these
pressure systems transports moist air to the mid-latitudes forming a rainband called *Baiu* in Japanese.
The general circulation pattern of the EASM and the rainband are well simulated in both MIROC6
and MIROC5. It should be noted that one of major deficiencies in MIROC5, the underestimate of the
precipitation around the Philippines, has been largely alleviated in MIROC6. This improvement is,
again, associated with the moistening of the lower troposphere by shallow convective processes.
Interannual EASM variabilities are examined using an empirical orthogonal function (EOF) analysis
of vorticity at the 850 hPa pressure level over [100°E–150°E, 0°N–60°N] following Kosaka and
Nakamura (2010). The regressions of precipitation and 850hPa vorticity with respect to the time series
of the first mode (EOF1) are shown in the lower panels of Fig. 21. In observations, precipitation and
vorticity anomalies show a tripolar pattern with centers located around the Philippines, Japan, and the
Sea of Okhotsk (Hirota and Takahashi, 2012). The anomalies around the Philippines and Japan
correspond to the so-called Pacific-Japan pattern (Nitta et al., 1987). The southwest-northeast
orientation of the wave-like anomalies is better simulated in MIROC6 than in MIROC5.

Figure 22 shows the wintertime (December–February) climatology of circulations and the

STA in the East Asia. The East Asian winter monsoon (EAWM) is characterized by northwesterly
between the Siberian high and the Aleutian low in observations (ERA-I; e.g. Zhang et al., 1997). The
monsoon northwesterly advects cold air to East Asia, enhancing the meridional temperature gradients
and strengthening the subtropical jet around Japan. The jet's strength influences synoptic wave
activities in the storm track. MIROC5 captures the circulation pattern, but significantly underestimates
the STA. The STA in MIROC6 is better simulated than in MIROC5, but it is still smaller than in
observations. Interannual variability of the EAWM is also better represented in MIROC6 than in
MIROC5. The dominant variability of the monsoon northwesterly is extracted as the EOF1 of the
meridional wind at the 850 hPa pressure level over the region [30°N–60°N, 120°E–150°E]. In
observations, the regressions with respect to the time series of the EOF1 show stronger northwesterly
accompanied with suppressed STA, which is consistent with previous studies (Fig. 22d; e.g.,
Nakamura, 1992). This relationship between the circulations and the STA can be found in MIROC6
but not in MIROC5 (Figs. 22e, f). The explained variance of the EOF1 is 46.0% in observations, 37.1%
in MIROC5, and 47.1% in MIROC6, suggesting that the amplitude of this variability in MIROC6 is
consistent with observations.

**3.2.2 Stratospheric circulations**
A few of the major changes in the model setting from MIROC5 to MIROC6 are higher
vertical resolution and higher model top altitude in MIROC6, namely, representation of the
stratospheric circulations. Here, we examine representation of the Quasi-Biennial Oscillations (QBOs)
in MIROC6. Figure 23 shows the time-height cross-sections of the monthly mean, zonal-mean zonal
wind over the equator for observations (ERA-I) and MIROC6. In this figure, an obvious QBO with
mean period of approximately 22 months can be seen in MIROC6. The mean period is slightly shorter
than that of ~28 months in observations, and the simulated QBO period varies slightly from cycle to
cycle. The maximum speed of the easterly at the 20 hPa pressure level is approximately -25 m s$^{-1}$ in
MIROC6 and that of the westerly is 15 m s$^{-1}$. On the other hand, the observed maximum wind speeds
are -35 m s$^{-1}$ for the easterly and 20 m s$^{-1}$ westerly, respectively. The simulated QBO has somewhat
weaker amplitude in MIROC6 than observations, but the same east-west phase asymmetry. The QBO
in the MIROC6 shifts upward compared with that in observations, and the simulated amplitude is
larger above the 5 hPa pressure level and smaller in the lower stratosphere. The simulated downward
propagation of the westerly shear zones of zonal wind ($\partial \bar{u}/\partial z > 0$, where $z$ is the altitude) is faster
than the downward propagation of easterly shear zones ($\partial \bar{u}/\partial z$) < 0, which agrees with observations.
The QBOs in MIROC6 are qualitatively similar to that represented in the MIROC-ESM, which is an
Earth system model with a similar vertical resolution that participated in the CMIP5 (Watanabe et al.,
2011). Note that nothing resembling a realistic QBO was simulated in the previous low-top version
MIROC5, which only has a few vertical layers in the stratosphere.

Recently, Yoo and Son (2016) found that the observed MJO amplitude in the boreal winter

is stronger than normal during the QBO easterly phase at the 50 hPa pressure level. They also showed
that the QBO exerted greater influence on the MJO than did ENSO. Marshall et al. (2016) pointed out
the improvement in forecast skill during the easterly phase of the QBO and indicated that the QBO
could be a potential source of the MJO predictability. MIROC6 successfully simulates both the MJO
and QBO in a way consistent with observations, as mentioned above, but correlations between the
QBO and MJO are insignificant. One possible reason is smaller amplitude of the simulated QBO in
the lowermost stratosphere. The QBO contribution to tropical temperature variation at the 100 hPa
pressure level is ~0.1 K in the MIROC6, which is much smaller than the observed value of ~0.5 K
(Randel et al., 2000). The simulated QBO has little effects on static stability and vertical wind shear
in the tropical upper troposphere.

MIROC6 can also simulate Sudden Stratospheric Warming (SSW), which is a typical intra-

seasonal variability of the mid-latitude stratosphere in the Northern Hemisphere. Standard deviation
of monthly and zonal-mean zonal wind (colors) superimposed on monthly climatology of zonal-mean
zonal wind (black contours) in February are shown in Fig. 24 (a)-(c). There are two maxima of the
standard deviations over the equatorial stratosphere and the mid-to-high latitude upper stratosphere in
the Northern Hemisphere in observations (Fig. 24a), which correspond to QBO and polar vortex
variability. This feature is well captured in MIROC6 (Fig. 24b), while there are too small variations in
MIROC5 where the stratosphere cannot be well resolved (Fig. 24c). The better representation of the
polar vortex variability in MIROC6 is closely associated with that of the SSW. As shown in the bottom
panels of Fig. 24, abrupt and short-lived warming events associated with SSW are detected in
MIROC6, which are reproduced comparably to observations in terms of magnitude, but are not
detected in MIROC5. This is consistent with previous modeling studies that reported the importance
of the well-resolved stratosphere for better simulation of stratospheric variability (e.g., Cagnazzo and
Manzini, 2009; Charlton-Perez et al., 2013; Osprey et al., 2013). In December–January, however,
MIROC6 still underestimates the frequency of SSW events, which is a common bias in other high-top
climate models (e.g., Inatsu et al., 2007; Charlton-Perez et al., 2013; Osprey et al., 2013). It is
conjectured that the less frequent SSW in December–January could be attributed to less frequent
stationary wave breakings due to overestimate of climatological zonal wind speed of the polar night
jet in MIROC6 (Figs. 24d and e).

The inclusion of a well-resolved stratosphere in MIROC6 is also considered to be

important for improvement in representation of stratosphere-troposphere coupling. In order to evaluate
this, we examine the time-development of the Northern Annular Modes (NAM) associated with
strongly weakened polar vortex events in the stratosphere. The NAM indices are defined by the first
EOF mode of the zonal-mean year-round daily geopotential height anomalies over the Northern
Hemisphere and are computed separately at each pressure level (Baldwin and Thompson, 2009). The
height anomalies are first filtered by a 10-day low-pass filter to remove transient eddies. Figure 25
shows the composite of time development of the NAM index for weak polar vortex events. The events
are determined by the dates on which the 10 hPa NAM index exceeded -3.0 standard deviations
(Baldwin and Dunkerton, 2001). Note that the NAM index is multiplied by the square root of the
eigenvalue in each level before the composite, that is, the composite having the geopotential height
dimension. The weak polar vortex signal in the stratosphere propagates downward to the surface and
persists approximately 60 days in the lower stratosphere and upper troposphere. These observational
features are well represented in MIROC6 (Figs. 25ab). Although MIROC5 has also captured
downward propagating signals, its magnitude is approximately half in the stratosphere, and its
persistency is weak in the lower stratosphere and upper troposphere. Therefore, these results strongly
indicate that the inclusion of a well-resolved stratosphere in a model is important for representing not
only stratospheric variability, but also stratosphere-troposphere coupling.

### 3.2.3 El Niño/Southern Oscillation and Indian Ocean Dipole mode

Among the various internal climate variabilities on interannual timescales, ENSO is of
great importance because it can influence climate not only in tropics but also mid- and high latitudes
of both hemispheres through atmospheric teleconnections associated with wave propagations (e.g.,
Hoskins and Karoly, 1981; Alexander et al., 2002). Here, we describe representation of ENSO and
related teleconnection pattern. Figure 26 shows anomalies of SST, precipitation, the 500 hPa pressure
height, and the equatorial ocean temperature regressed onto the NINO3 index which is defined as the
area average of the SST in [5°S–5°N, 150°W–90°W]. ProjD and ERA-I in 1980–2009 are used as
observations. Although the maximum of the SST anomalies in the tropical Pacific is shifted more
westward than in observations, the ENSO-related SST anomalies simulated in both of MIROC6 and
MIROC5 are globally consistent with observations (Figs. 26a-c). Simulated positive precipitation
anomalies in MIROC6 still overextend to the western Pacific (Figs. 26d-f). Meanwhile, dry anomalies
over the maritime continent, the eastern equatorial Indian Ocean, and the SPCZ are better simulated
in MIROC6 than in MIROC5. ENSO teleconnection patterns in Z500 (Figs. 26g-i) are also realistically
simulated as seen in, for example, the Pacific-North American pattern (Wallace and Gutzler, 1981).
Equatorial subsurface ocean temperature anomalies in MIROC6 are more confined within the
thermocline than in MIROC5 (Figs. 26j-l), and the signals in MIROC6 are closer to observations.
However, the subsurface signals in MIROC6 reside deeper than in observations. This is due to the
difference in the climatological structure of the equatorial thermocline, which is attributed to the
overestimate of the trade winds over the equatorial Pacific, as mentioned in Section 3.1.2.
As well as ENSO, the Indian Ocean Dipole (IOD) mode is recognized as a prominent
interannual variability (Saji et al., 1999; Webster et al., 1999). Figure 27 shows anomalies of SST, 10
m wind, and precipitation regressed onto the autumn (September–November) dipole mode index
(DMI) which is defined as the zonal difference of the anomalous SST averaged over [10°S–10°N,
50°E–70°E] and that averaged in [10°S–10°N, 90°E–110°E]. ProjD and ERA-I in 1980–2009 are used
as observations. The observed positive IOD phase is characterized by a basin-wide zonal mode with
positive (negative) SST anomalies in the western (eastern) Indian Ocean, and precipitation is increased
(decreased) over the positive (negative) SST anomalies (Figs. 27ad). The dipole SST pattern is better
simulated in MIROC6 than in MIROC5 where the eastern SST anomalies are located more southward
than in observations (Figs. 27a-c). Correspondingly, a meridional dipole pattern in the precipitation of
MIROC5 is alleviated, and MIROC6 shows a zonal dipole precipitation pattern, as in observations
(Figs. 27d-f). Seasonal IOD phase locking to boreal autumn, which is assessed based on RMS
amplitude of the DMI, is also better simulated in MIROC6 than in MIROC5 (not shown). Seasonal
shoaling of the eastern equatorial thermocline in the Indian Ocean is realistically simulated in
MIROC6 during boreal summer to autumn. The shallower thermocline leads the stronger thermocline
feedback which is evaluated based on the SST anomalies regressed onto the 20°C isotherm depth
anomalies averaged over the eastern part of the IOD region. As displayed in the top of the upper panels
of Fig. 27, the thermocline feedback in MIROC6 is comparable to observations. This larger
thermocline feedback in MIROC6 possibly leads to the above-mentioned improvements in the IOD
pattern. Note that the simulated surface wind anomalies are more realistic in MIROC6 than in
MIROC5, although the magnitude of SST anomalies is overestimated in MIROC6. The overestimate
of the SST anomalies may have arisen from an excessive response of the equatorial and coastal Ekman
up- and down-welling to the wind changes, which are favorable in coarse-resolution ocean models.

**3.2.4 Decadal-scale variations in the Pacific and Atlantic Oceans**

On longer-than-interannual timescales, the PDO (Mantua et al., 1997) or the Inderdecadal

Pacific Oscillations (IPO; Power et al., 1999) is known to be a dominant climate mode that is detected
in the SST and the SLP over the North Pacific. To examine simulated PDO patterns, monthly SST and
wintertime (December–February) SLP anomalies are regressed onto the PDO index defined as the 1st
EOF mode of the North Pacific SST to the north of 20°N and are shown in Fig. 28. In order to detect
the decadal-scale variation, the COBE-SST2/SLP2 data (Hirahara et al., 2014) from 1900 to 2013 are
used as observations. Negative SST anomalies in the western and central North Pacific and positive
SST anomalies in the eastern North Pacific are found in observations. These signals are also
represented in both of MIROC6 and MIROC5. The regression of SLP anomalies corresponding to the
deepening of the Aleutian low are well simulated in the models over the subarctic North Pacific, and
it can be seen that the amplitudes of the SLP anomalies are larger in MIROC6 than in MIROC5, which
is closer to the observation. In the tropical Pacific, positive SST anomalies, which are among the more
important driving processes of the PDO (e.g. Alexander et al., 2002), are seen in both the models and
the observations. In MIROC5, the 5-yr running means of the wintertime (November–March) North
Pacific Index (NPI), defined as the SLP averaged over [30°N–65°N, 160°E–140°W], are less sensitive
to the NINO3 index (correlation coefficient $r = -0.37$) than to the NINO4 index ($r = -0.64$). Note that
the NINO4 index is defined as the area average of the SST in [5°S–5°N, 160°E–150°W]. The distorted
response of the extratropical atmosphere to the tropical SST variations works to unsuitably modify the
extratropical ocean and plays a major role in limiting the decadal predictability of the PDO index in
MIROC5 (Mochizuki et al., 2014). In contrast, those in MIROC6 are well correlated with the NINO3
index ($r = -0.61$) in addition to the NINO4 index ($r = -0.62$). Overestimate of the tropical signals of
MIROC5 in the western tropical Pacific are also alleviated in MIROC6. The above-mentioned PDO
improvement and the linkage between the tropics and the mid-latitude North Pacific imply a potential
for improved skills in initialized decadal climate predictions.

In the Atlantic Ocean, there is another decadal-scale variability, which is called the AMO

(Schlesinger and Ramankutty, 2004). Figure 29 shows anomalies of SST and SLP regressed onto the
AMO index, which is defined as the area average of the SST anomalies in the North Atlantic [0°–
60°N, 0°–80°W] with the global-mean SST anomalies subtracted (Trenberth and Shea, 2006). As in
the PDO, the centennial-long data of the COBE-SST2/SLP2 data in 1900–2013 are used as
observations. The observed AMO spatial pattern in its positive phase is characterized by positive SST
anomalies in the off-equator and the subarctic North Atlantic, and by negative or weakly-positive SST
anomalies in the western subtropical North Atlantic (Fig. 29a). Corresponding to negative (positive)
SLP anomalies over the subtropical (subarctic) North Atlantic, the mid-latitude westerly jet is weaker
in a positive AMO phase than in normal years. These spatial patterns in the SST and SLP are simulated
in both of MIROC6 and MIROC5. It is especially noteworthy that the positive SST anomalies in low
latitudes have larger amplitudes in MIROC6 than in MIROC5, and they extend to the South Atlantic
as in observations (Figs. 29bc). On the other hand, the positive SST anomalies in the subarctic region
are underestimated in MIROC6, which may be due to the smaller RMS amplitudes of NADW transport
in MIROC6 (see Section 3.1).

**3.3 Climate sensitivity**
Following the regression method by Gregory et al. (2004) and Gregory and Webb (2008),
we conducted abrupt $CO_2$ quadrupling experiments with MIROC6 and MIROC5 in order to evaluate
effective climate sensitivity (ECS), radiative forcing, and climate feedback. The $CO_2$ quadrupling
experiments were initiated from the pre-industrial control runs. Data from the first 150 years after the
$CO_2$ increase were used for the analysis.
ECS, $2 \times CO_2$ radiative forcing, and climate feedback for MIROC6 are estimated to be 2.6
K, 3.7 $Wm^{-2}$, and -1.4 $Wm^{-2}K^{-1}$, respectively (Fig. 30a and Table 2). The ECS, radiative forcing, and
climate feedback in MIROC6 are lower, higher, and negatively larger than those of the CMIP5 multi-
model ensemble means, although these estimates for MIROC6 are within the ensemble spread of the
multi-models (Andrews et al., 2012). The ECS of MIROC6 is almost the same as MIROC5 because
the decrease in radiative forcing is counterbalanced by the positive increase in climate feedback,
although the change in climate feedback is small and not statistically significant. The decrease in
radiative forcing of MIROC6 relative to MIROC5 is evident in the longwave and shortwave cloud
components (LCRE and SCRE in Fig. 30b and Table 3). On the other hand, the clear-sky shortwave
component (SWclr) increases in MIROC6 relative to MIROC5, which partially cancels the differences
between the two models. The positive increase in climate feedback is pronounced in the SCRE, which
is partially offset by the decrease in the clear sky longwave (LWclr) and SWclr (Fig. 30c and Table 3).
We now focus on the SCRE of the radiative forcing and climate feedback, which show the
largest differences between the two models, and compare the geographical distribution (Fig. 31). The
distribution is calculated by regressing the changes in SCRE caused by the $CO_2$ increase at each
latitude-longitude grid box against the change in the global-mean SAT. There is a large difference in
the geographical distribution between MIROC6 and MIROC5, with the former showing more
pronounced zonal contrast in the tropical Pacific than the latter. The changes in the global mean from
MIROC5 to MIROC6 (Figs. 30bc) are correlated with the changes in the western tropical Pacific,
showing more negative radiative forcing and more positive climate feedback, which are partially offset
by the changes in the central tropical Pacific with opposite signs. The radiative forcing and climate
feedback tend to show similar geographical patterns with opposite signs in each model.

**4. Summary and discussions**
The sixth version of a climate model, MIROC6, was developed by a Japanese climate
modeling community, aiming at contributing to the CMIP6 through deeper understanding of a wide
range of climate science issues and seasonal-to-decadal climate predictions and future climate
projections. The model configurations and basic performances in the pre-industrial control simulation
have been described and evaluated in the present manuscript. Major changes from MIROC5, which
was our official model for the CMIP5, to MIROC6 are mainly done in the atmospheric component.
These include implementation of a parameterization of shallow convective processes, the higher model
top and vertical resolution in the stratosphere. The ocean and land-surface components have been also
updated in terms of the horizontal grid coordinate system and higher vertical resolution in the former,
and parameterizations for sub-grid scale snow distribution and wet lands due to snow-melting water
in the latter. Overall, the model climatology and internal climate variability of MIROC6, which are
assessed in comparison with observations, are better simulated than in MIROC5.
Overestimate of low-level cloud amounts in low latitudes, which can be partly attributed
to insufficient representation of shallow convective processes, are significantly alleviated in MIROC6.
The free atmosphere becomes wetter and the precipitation over the western tropical Pacific becomes
larger in MIROC6 than in MIROC5, primarily due to vertical mixing of the humid air in the planetary
boundary layer with the dry air in the free troposphere. Shallow convections also contribute to better
propagation characteristics of intra-seasonal variability associated with MJO in MIROC6, as well as
East Asian summer monsoon variability on interannual timescales. In addition, QBO, which is absent
in MIROC5, appears in MIROC6 because of its better stratospheric resolution and non-orographic
gravity wave drag parameterization.
Climatic mean and internal climate variability in the mid-latitudes are also improved in
MIROC6. Together with enhanced activity of sub-weekly disturbances, the tropospheric westerly jets
in MIROC6 are shifted more poleward and are stronger than in MIROC5, especially in the Northern
Hemisphere. Overestimates in zonal wind speed of the polar night jet are reduced in MIROC6. These
advanced representations lead to tighter interactions between the troposphere and the stratosphere in
MIROC6. SSW events in the form of polar vortex destructions induced by upward momentum transfer
from the troposphere to the stratosphere (e.g., Matsuno, 1971), are well captured in MIROC6. On
interannual timescales, the improvement of the westerly jet results in better representations of the
spatial wind pattern of the wintertime East Asian monsoon. Associated with changes in the large-scale
atmospheric circulations, the western boundary currents in the oceans, the Kuroshio-Oyashio current
system, the Gulf Stream, and their extensions are better simulated in MIROC6. The increase in warm
water transport from the subtropical North Atlantic to the Barents Sea seems to melt the sea ice in the
Barents Sea, and to alleviate the overestimate of the wintertime sea-ice area that is seen in that region
in MIROC5. Another improvement in MIROC6 is found in the climatological snow cover fractions in
the early winter over the Northern Hemisphere continents. In the Southern Hemisphere, however, the
underestimate of mid-level clouds and the corresponding warm SAT bias, the underestimate of sea-ice
area, and the overestimate of incoming shortwave radiation in the Southern Ocean, all of which are
attributed to errors in cloud radiative and planetary boundary layer processes (e.g., Bodas-Salcedo et
al., 2012; Williams et al., 2013), remains the same as in MIROC5.
Qualitatively, the linkage representations between the tropics and the mid-latitudes
associated with ENSO in MIROC6 are mostly the same as in MIROC5. Meanwhile, oceanic
subsurface signals, which partly control ENSO characteristics, are more confined along the equatorial
thermocline in MIROC6, which is consistent with observations. Regarding the PDO, tropical influence
on the mid-latitudes is more dominant in MIROC6 than in MIROC5, suggesting improvements in
decadal-scale atmospheric teleconnections in MIROC6.
The above descriptions are mainly on the Pacific internal climate variabilities. Regarding
the Indian Ocean, the zonal dipole structures in the SST and precipitation associated with the
interannual variability, known as the IOD, are better simulated in MIROC6 than in MIROC5, which
has a bias of a false meridional precipitation pattern. In the Atlantic, the multi-decadal variability,
known as the AMO, is represented in both the models roughly consistent with observations, but their
reproducibility shows both drawbacks and advantage. Signals associated with AMO in the subarctic
(tropical) region are underestimated in MIROC6 (MIROC5).

As a metric for climate change induced by atmospheric $CO_2$ increase, ECS is also estimated.

Although the model configurations and performances are different between the models, the ECS is
almost the same (2.5 K). However, looking at geographical distributions of radiative forcing and
climate feedback, the amplitudes of shortwave cloud components are much larger in MIROC6 than in
MIROC5. Since the larger negative (positive) radiative forcing and positive (negative) climate
feedback in the western (central) tropical Pacific cancel each other, global-mean quantities in MIROC6
almost remain the same as in MIROC5. As a topic of future study, estimating radiative forcing and
climate feedback with Atmospheric Model Intercomparison Project-type experiments in order to check
robustness of the present study would be desirable. Elucidating the impact of different geographical
patterns of radiative forcing and climate feedback on the projected future climates would also be useful.

After conducting the pre-industrial control simulation and evaluating the model

reproducibility of the mean climate and the internal climate variability, ensemble historical simulations
that were initiated from the pre-industrial simulations were executed using the historical forcing data
recommended by the CMIP6 protocol. Figure 32 shows a time series of the global-mean SAT
anomalies with respect to the 1961–1990 mean. There are 30 (5) ensemble members in the MIROC6
(MIROC5) historical simulations. Note that the MIROC5 historical simulations are executed using the
forcing datasets of the CMIP5 protocol. As shown in Fig. 32, the simulated SAT variations in both of
MIROC6 and MIROC5 follow observations (HadCRUTv4.4.0; Morice et al., 2012; the data are
available at https://crudata.uea.ac.uk/cru/data/temperature/) on a centennial timescale. The
temperature rises from the nineteenth century to the early twenty-first century are about 0.72 K in
MIROC6, 0.85 K in MIROC5, and 0.82 K in observations, respectively. Focusing on the period from
the 1940s to the 1960s, the SAT variations seem to be better simulated in MIROC6 than in MIROC5,
which can be due to both of an update of the forcing datasets and the larger ensemble number in
MIROC6. On the other hand, the warming trend during the first half of the twentieth century in the
models is about half as large as in observations. Whether it can be attributed to internal climate
variability (e.g., Thompson et al., 2014; Kosaka and Xie, 2016) or to an externally forced mode (e.g.,
Meehl et al., 2003; Nozawa et al., 2005) is still being debated. The so-called recent hiatus of the global
warming (Easterling and Wehner, 2009) in the first decade of the twenty-first century is not simulated
in both of MIROC6 and MIROC5. The observed hiatus is considered to occur in association with a
negative IPO phase as internal climate variations (e.g., Meehl et al., 2011; Watanabe et al., 2014). As
external drivers of the hiatus, the increase in stratospheric water vapor and the weakening of solar
activity are given as possible candidates, for example (e.g., Solomon et al., 2010; Kaufmann et al.,
2011). Failure of simulating the hiatus in the models could be attributed to uncertainties in the
historical forcing datasets or cancellation of internal climate variations of the IPO by ensemble-mean
manipulation of the individual historical simulations.

As summarized above, the overall reproducibility of the mean climate and the internal

variability in the latest version of our climate model, MIROC6, has progressed, as well as the historical
warming trend of the climate system. During the first trial of the preindustrial simulation conducted
just after the model configuration was frozen, however, the model reproducibility was not as good as
seen in MIROC5. As described in Section 2.5, we intensively tuned the model by perturbing
parameters associated with, especially, cumulus and shallow convections, and planetary boundary
processes. In addition, before starting the historical simulations, we estimated and tuned the radiative
forcing due to aerosol-radiation and aerosol-cloud interactions by changing the parameters of cloud
microphysics in order to ensure that the estimated radiative forcing would be closer to the best-estimate
of the IPCC-AR5 (IPCC, 2013). Without this parameter tuning, the simulated warming trend after the
1960s was 70% as large as seen in observations. This dependence of radiative forcing and
reproducibility of the warming trend on cloud microphysics has also been reported in other climate
models (Golaz et al., 2013). A recent comparison of cloud mircophysical statistics between climate
models and satellite-based observations has pointed out that "tuned" model parameters that were
adjusted for adequate radiative forcing and realistic SAT changes do not necessarily ensure cloud
properties and rain/snow formations will be consistent with observations and implies the presence of
error compensations in climate models (e.g., Suzuki et al., 2013; Michibata et al., 2016). Error
compensations are found also in both of global and regional aspects. As described in Section 3.1, the
global TOA radiation imbalance in MIROC6 is about -1.1 $Wm^{-2}$, which is in the acceptable range of
observations. However, when the TOA imbalance is examined in parts, cloud radiative components in
the model contain non-negligible biases with respect to satellite-based observations. Regarding error
compensations in the oceanic processes, for example, the modeled northward transport of CDW,
which is within the uncertainty range of observations, is maintained by spurious open ocean
convections in the Southern Ocean which often appear in coarse-resolution ocean models where
oceanic mesoscale eddies and coastal bottom water formation cannot be represented (e.g., Olbers et
al., 2004; Downes and Hogg, 2013).

There remain several key foci of ongoing model development efforts. These include

process-oriented refinements of cloud microphysics and convective systems based on constraints from
satellite data and feedbacks from cloud-resolving atmospheric models (e.g., Satoh et al., 2014), higher
resolutions for representations of regional extremes, oceanic eddies and river floods, and
parameterization of tide-induced micro-scale mixing of sea water. Improvement of computational
efficiency, especially on massive parallel computing systems, is among the urgent issues for long-term
and large ensemble simulations. These improvements can contribute to deeper understanding of the
Earth's climate, reducing uncertainties in climate projections and predictions, and more precise
evaluations of human influences on carbon-nitrogen cycles when applied to Earth system models.

*Code and data availability. Please contact the corresponding author if readers may want to validate the model configurations of MIROC6 and MIROC5 and to conduct replication experiments. The source codes and required input data will be provided by the modeling community where the author belongs. The model output from the CMIP6/CMIP5 pre-industrial control and historical simulations used in the present manuscript are distributed through the Earth System Grid Federation and are freely accessible. Details on ESGF are given on the CMIP Panel website (https://www.wcrp-climate.org/wgcm-cmip).*

*Competing interests. The authors declare that they have no conflict of interest.*

**Acknowledgements**

This research is supported by the "Integrated Research Program for Advancing Climate Models (TOUGOU Program)" from the Ministry of Education, Culture, Sports, Science, and Technology (MEXT), Japan. Model simulations were performed on the Earth Simulator at JAMSTEC and NEC SX-ACE at NIES. The authors are much indebted to Dr. Teruyuki Nishimura and Mr. Hiroaki Kanai for their long-term support in areas related to model developments and server administration. The authors also wish to express thanks to our anonymous reviewers for their suggestions and careful reading of the manuscript.

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

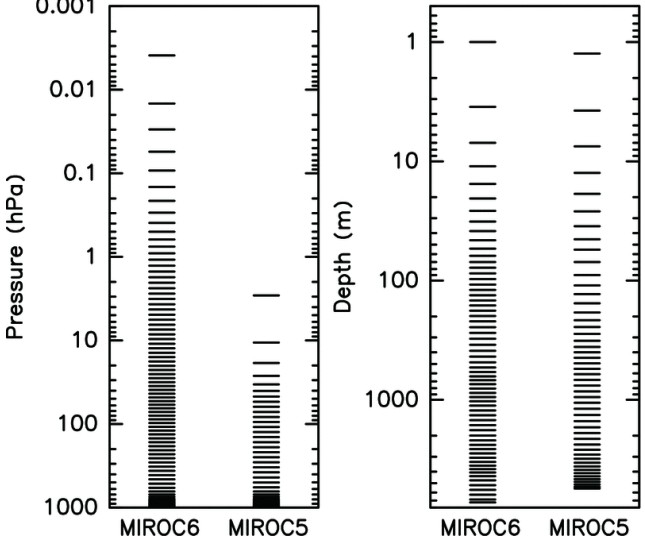


Fig. 1. Vertical half levels for the atmospheric (left panel) and the oceanic (right panel) components of
MIROC6 and MIROC5.

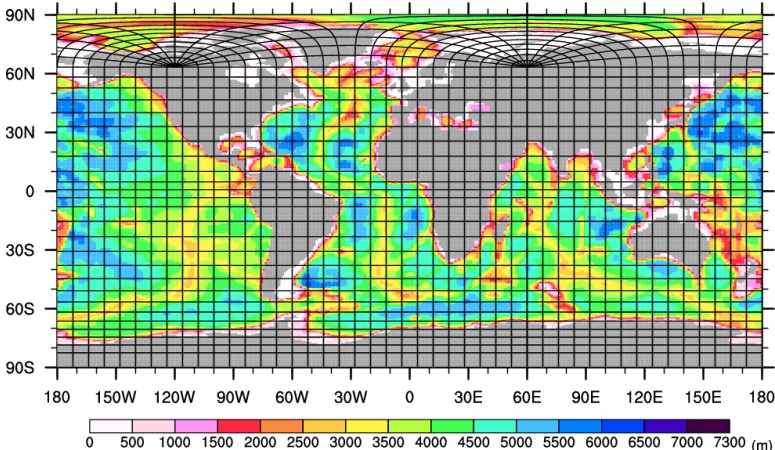


Fig. 2. Horizontal grid coordinate system and model bathymetry of the ocean component of MIROC6.



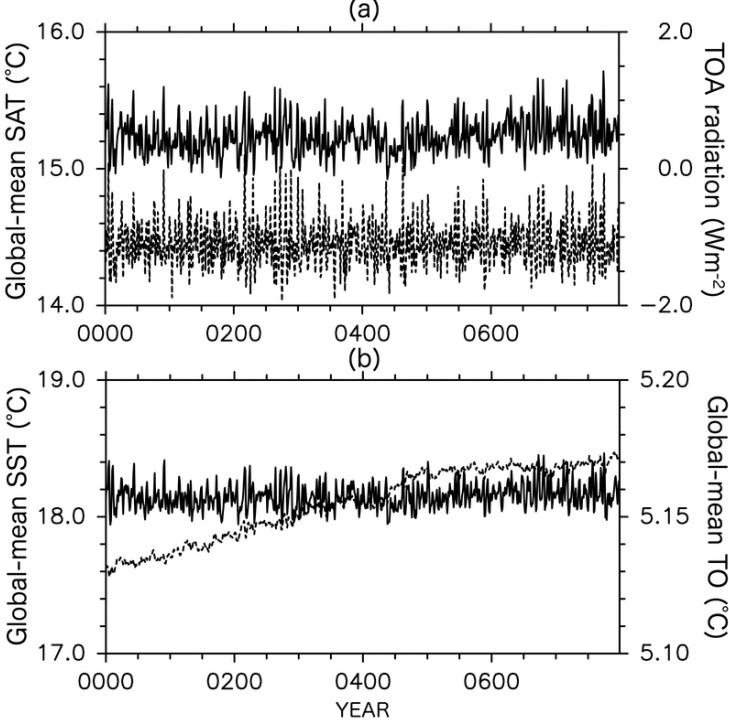


Fig. 3. (a) Time series of the global-mean SAT (solid) and the TOA radiation budget (dashed; upward
positive). (b) Same as (a), but for the global-mean SST (solid) and the ocean temperature through the
full water column (dashed).


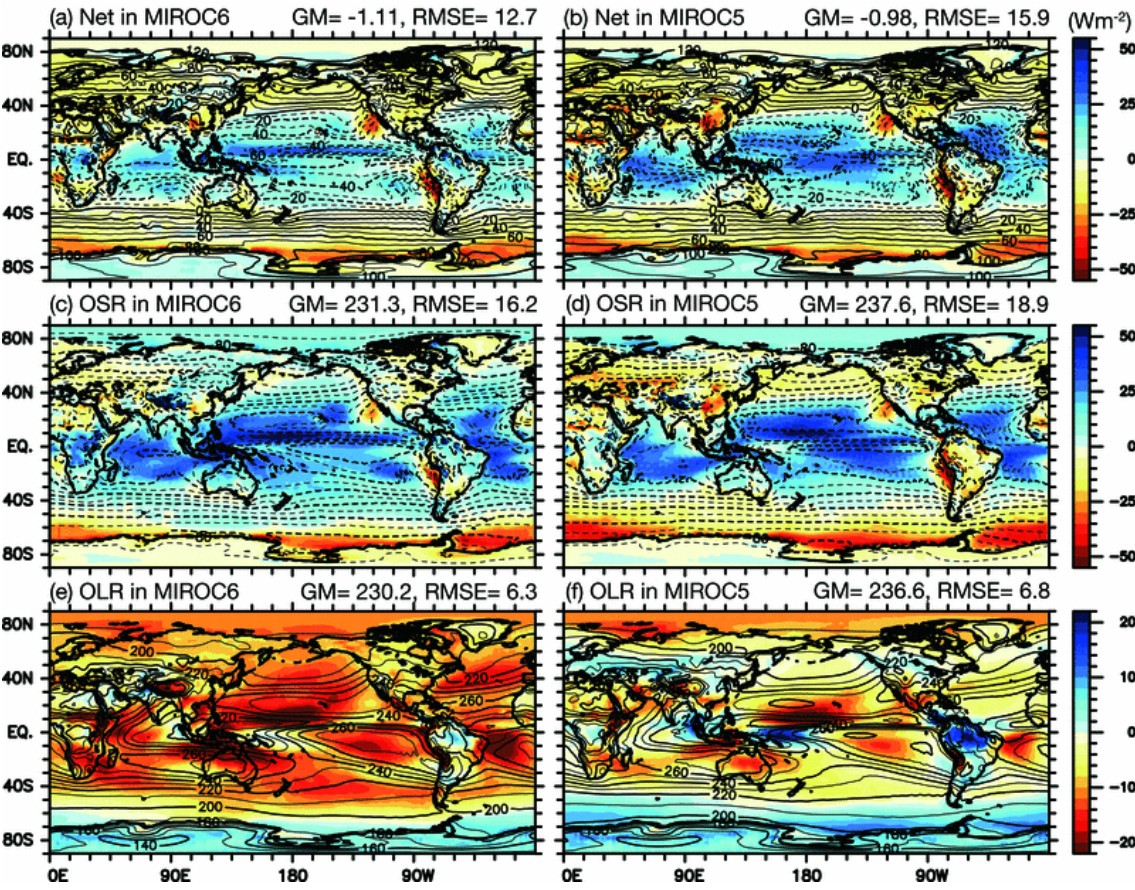


Fig. 4. Annual-mean TOA radiative fluxes in MIROC6 (left panels) and MIROC5 (right panels).
Upward is defined as positive. Net shortwave and longwave radiative fluxes, and the sum of the two
fluxes are denoted as OSR, OLR, and NET, respectively. Colors indicate errors with respect to
observations (CERES) and contours denote values in each model. Note that a different color scale is
used for the longwave radiations. The global-mean values and root-mean-squared errors are indicated
by GM and RMSE, respectively. In the present manuscript, RMSE is computed without model and
observed global-mean quantities unless otherwise noted.




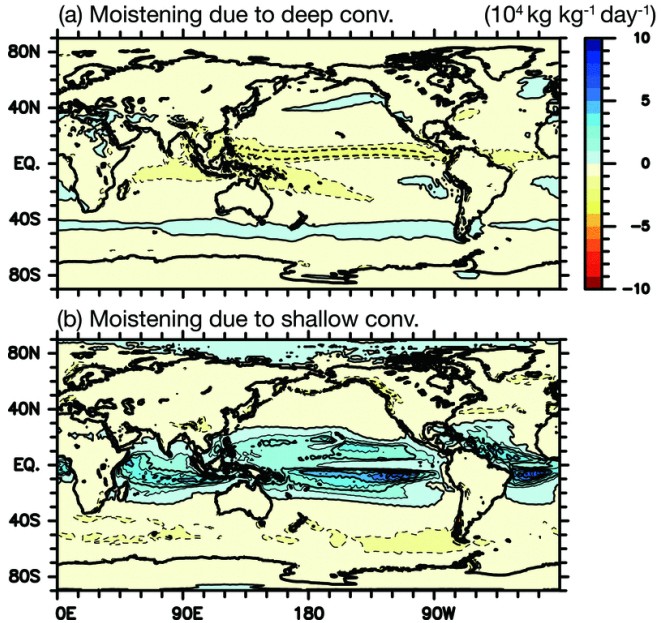


Fig. 5. Annual-mean moistening rate associated with (a) deep convections and (b) shallow convections
in MIROC6 at the 850 hPa pressure level.

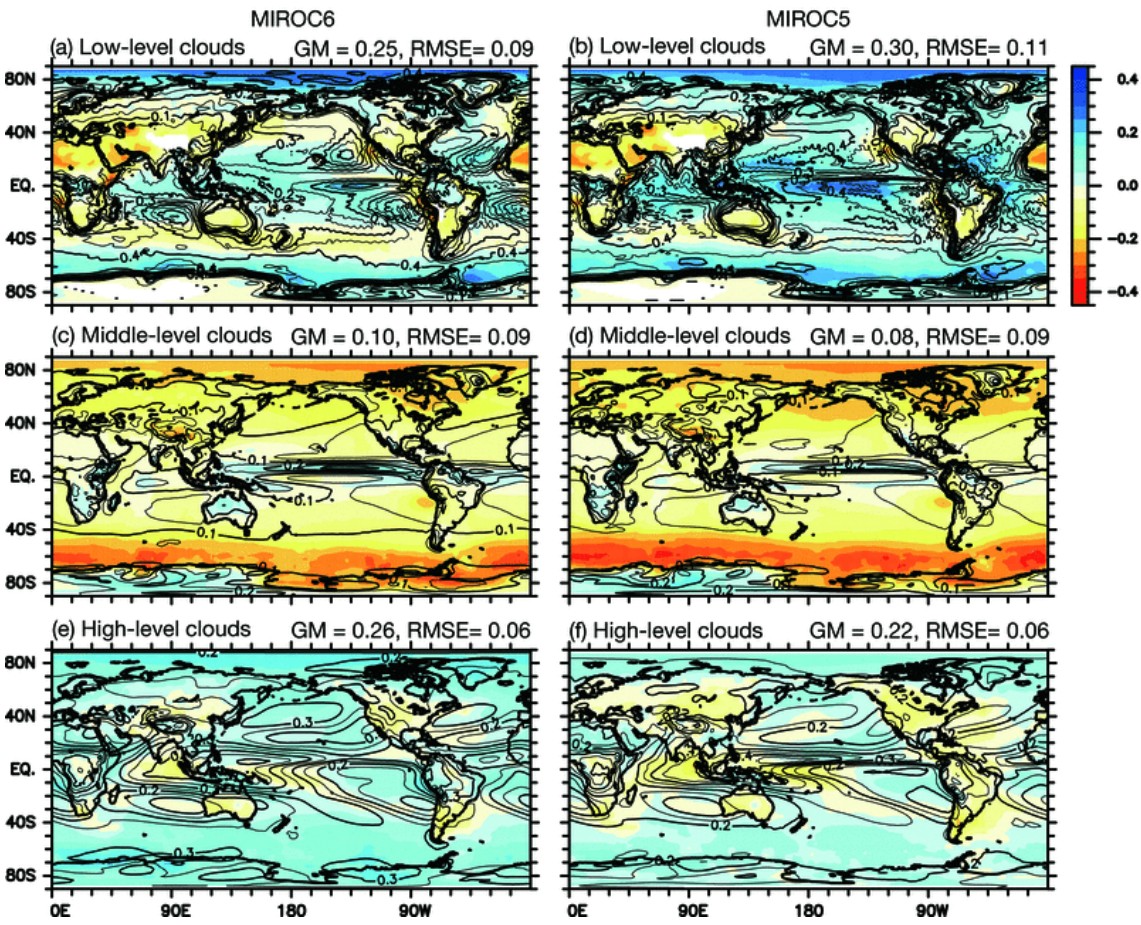

Fig. 6. Same as Fig. 4, but for cloud covers in MIROC6 (left panels) and MIROC5 (right panels).

Low-, middle-, and high-level cloud covers are aligned from the top to the bottom. The tops for low-,

middle-, and high-level clouds are defined to exist below the 680 hPa, between the 680 hPa and 440

hPa, and above the 440 hPa pressure levels, respectively. The unit is non-dimensional. ISCCP

climatology is used as observations.

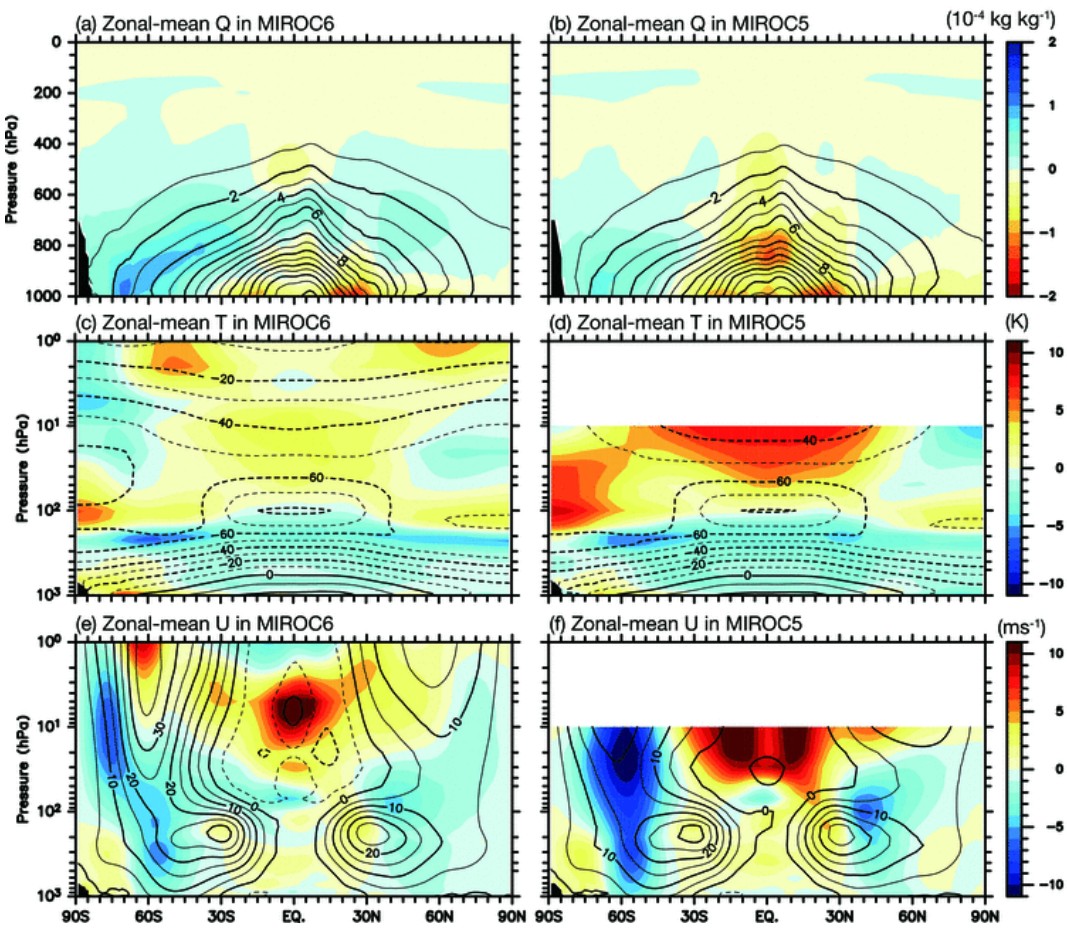


Fig. 7. Annual and zonal-mean specific humidity (top panels), temperature (middle), and zonal wind
(bottom) in MIROC6 (left) and MIROC5 (right). Colors indicate errors with respect to observations
(ERA-I) and contours denote values in each model.

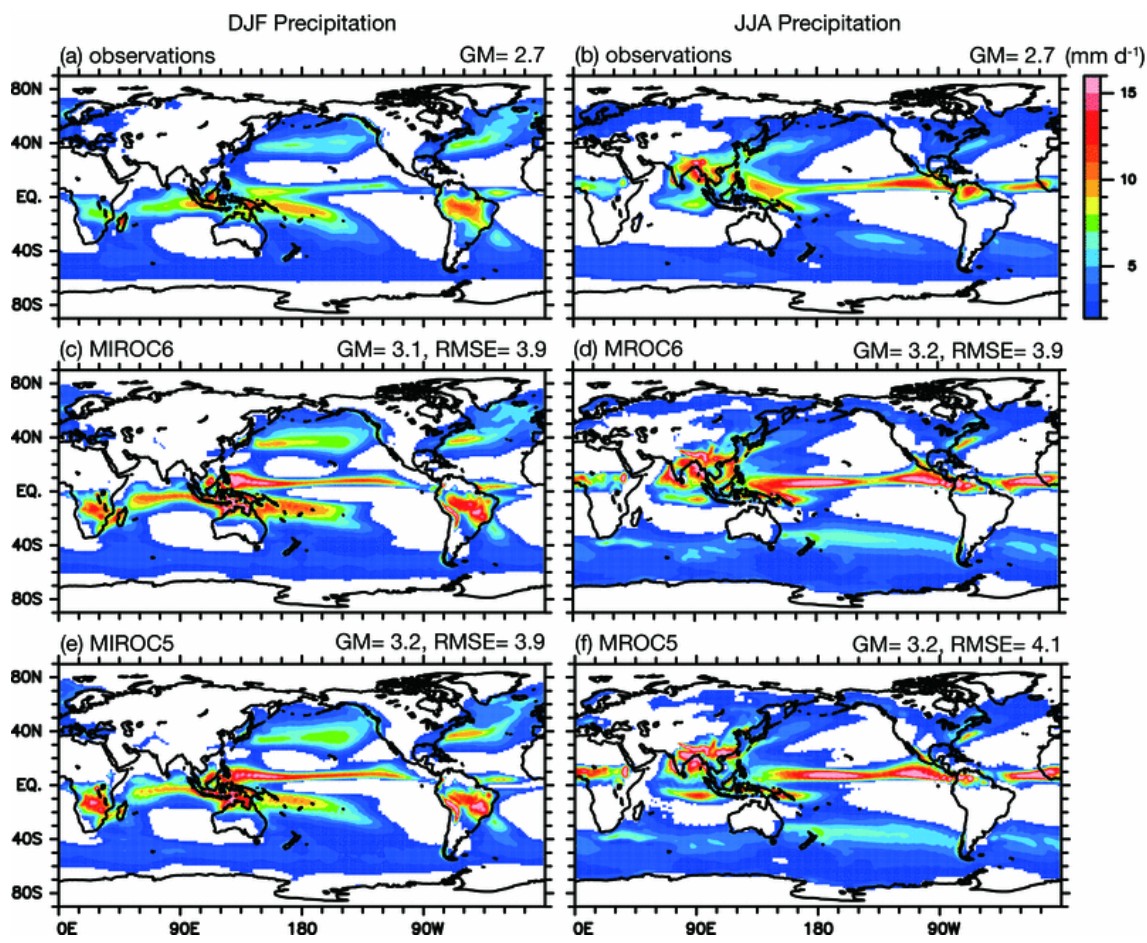


Fig. 8. Precipitation in boreal winter (December–February; left panels) and summer (June–August;
right panels) in observations (top; GPCP), MIROC6 (middle), and MIROC5 (bottom). Areas with
precipitation smaller than 3 mm d$^{-1}$ are not colored.



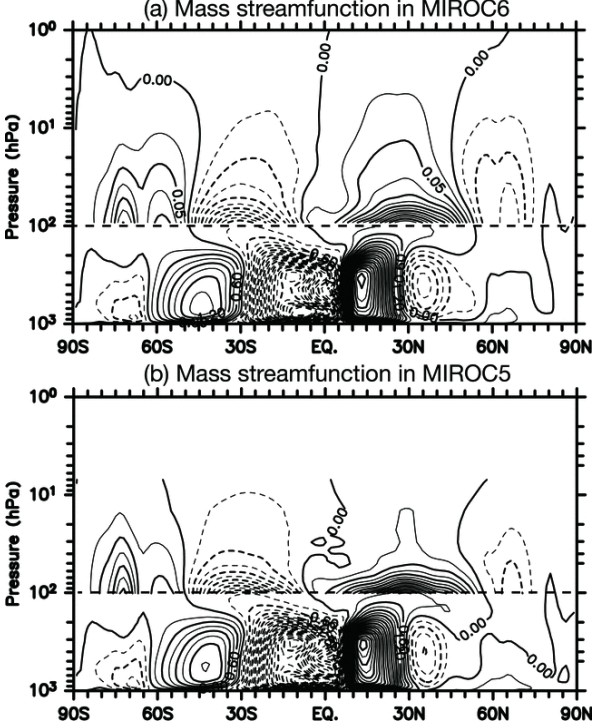


Fig. 9. Annual-mean mass stream functions in (a) MIROC6 and (b) MIROC5. Contour interval is 0.3

(0.025) $\times 10^{10}$ kg s$^{-1}$ below (above) the 100 hPa pressure level. Negative values are denoted by dashed

contours, and the horizontal dashed lines indicate the 100 hPa pressure level.




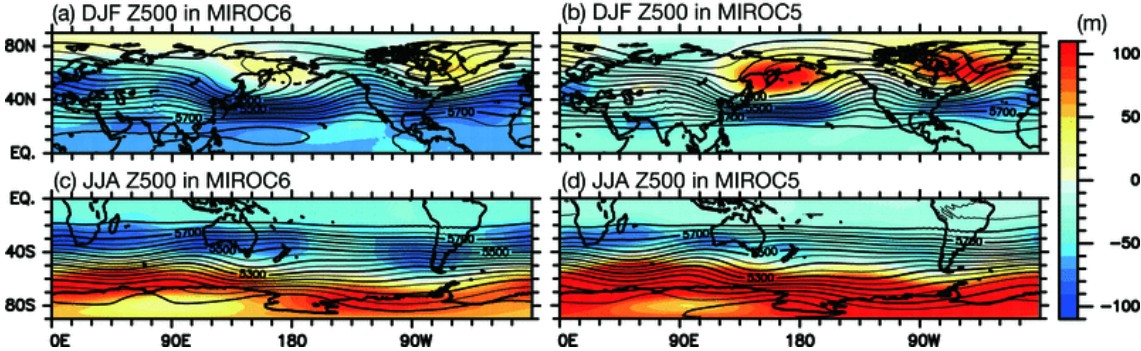

Fig. 10. Same as Fig. 4, but for the wintertime 500 hPa pressure level in MIROC6 (left panels) and
MIROC5 (right panels). Maps for boreal (austral) winter are shown in the upper (lower) panels. ERA-
I is used as observations.

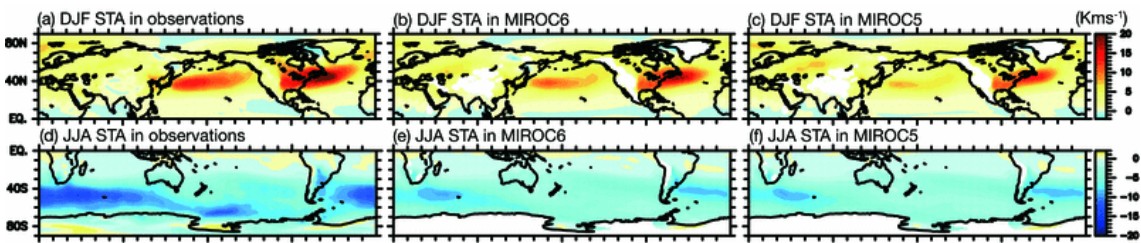

Fig. 11. Wintertime storm track activity (STA) in observations (left), MIROC6 (center), and MIROC5
(right). STA is defined as 8-day-highpass-filtered eddy meridional temperature flux at the 850 hPa
pressure level. Maps for boreal (austral) winter are shown in the upper (lower) panels. ERA-I is used
as observations.




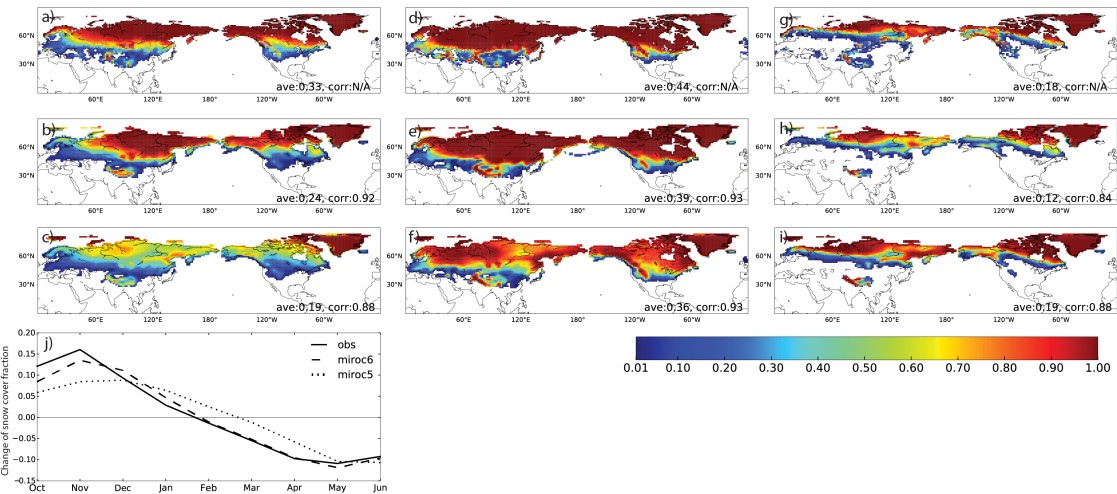


Fig. 12. Snow cover fractions for observations (top panels), MIROC6 (middle), and MIROC5 (bottom).
Maps in November, February, and May are aligned from the left to the right. The unit is non-
dimensional. Areas where snow cover fractions are less than 0.01 are masked. Ave and corr. in the
panels indicate spatial averages and correlation coefficients between observations and models over the
land surface in the Northern Hemisphere, respectively. Time series in the bottom-left panel shows
temporal rate of change of the monthly spatial averages. Snow-cover dataset of the Northern
Hemisphere EASE-Grid 2.0 is used as observations.



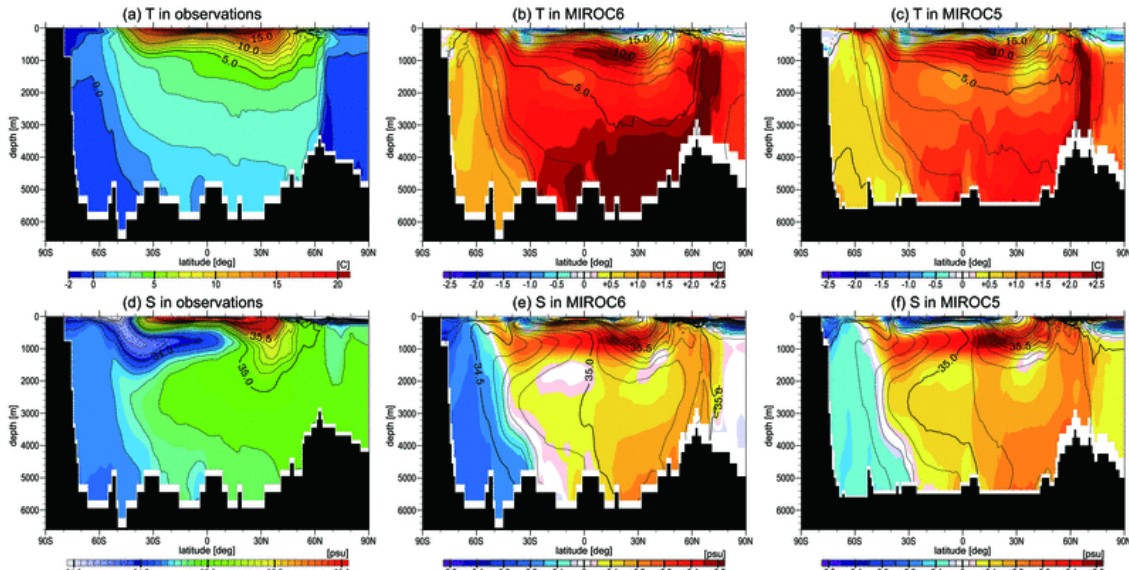


Fig. 13. Annual-mean potential temperature (upper panels; unit is °C) and salinity (lower; psu) in the
Atlantic sector for observations (left), MIROC6 (middle), and MIROC5 (right). Colors indicate errors
with respect to observations (ProjD) and contours denote model values in the middle and right panels.


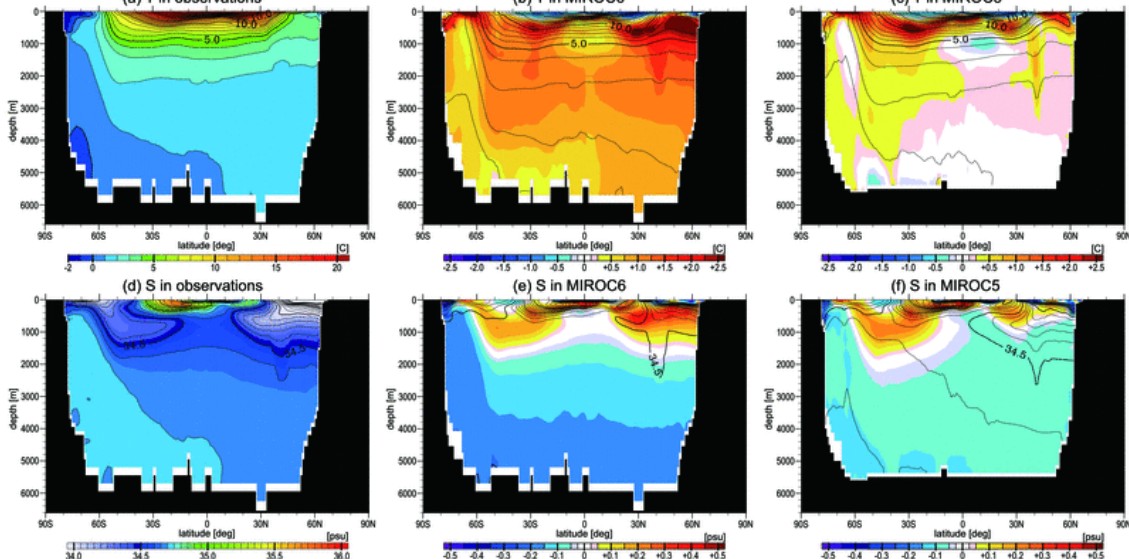


Fig. 14. Same as Fig. 13, but for the Pacific sector.



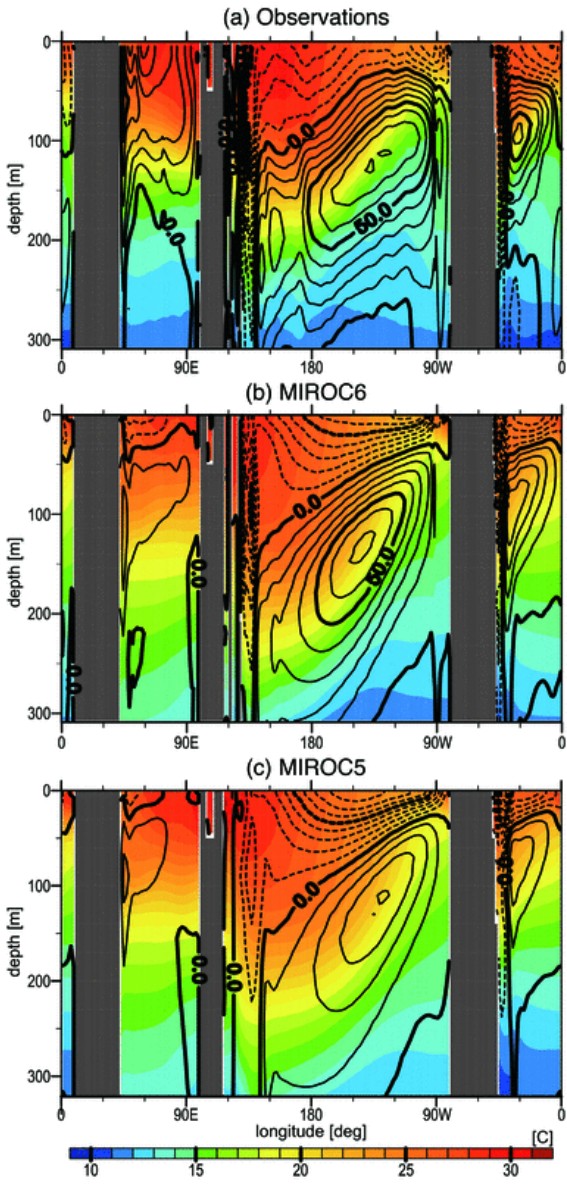


Fig. 15. Annual-mean climatology of potential temperature (°C; colors) and zonal current speed (cm
s$^{-1}$; contours) along the equator (1°S–1°N) in (a) observations (ProjD and SODA), (b) MIROC6, and
(c) MIROC5.


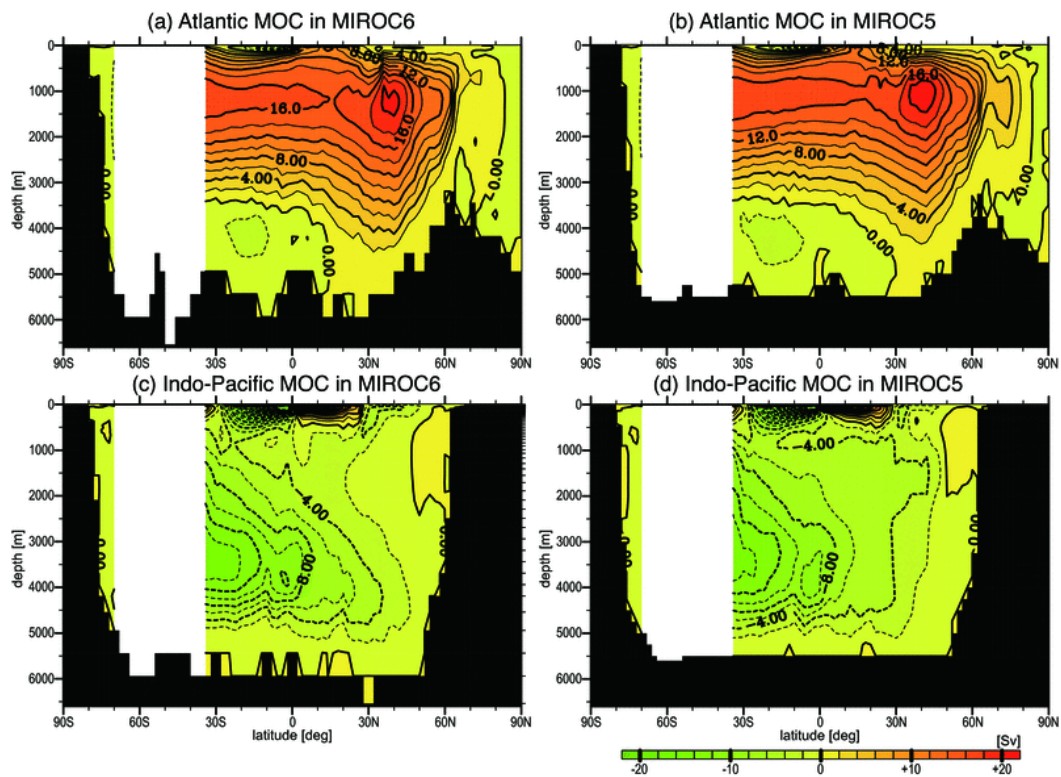


Fig. 16. Annual-mean meridional overturning circulations in the Atlantic (upper panels) and the Indo-
Pacific sectors (lower) in MIROC6 (left) and MIROC5 (right). The unit is Sv ($\equiv 10^6$ m$^3$s$^{-1}$).



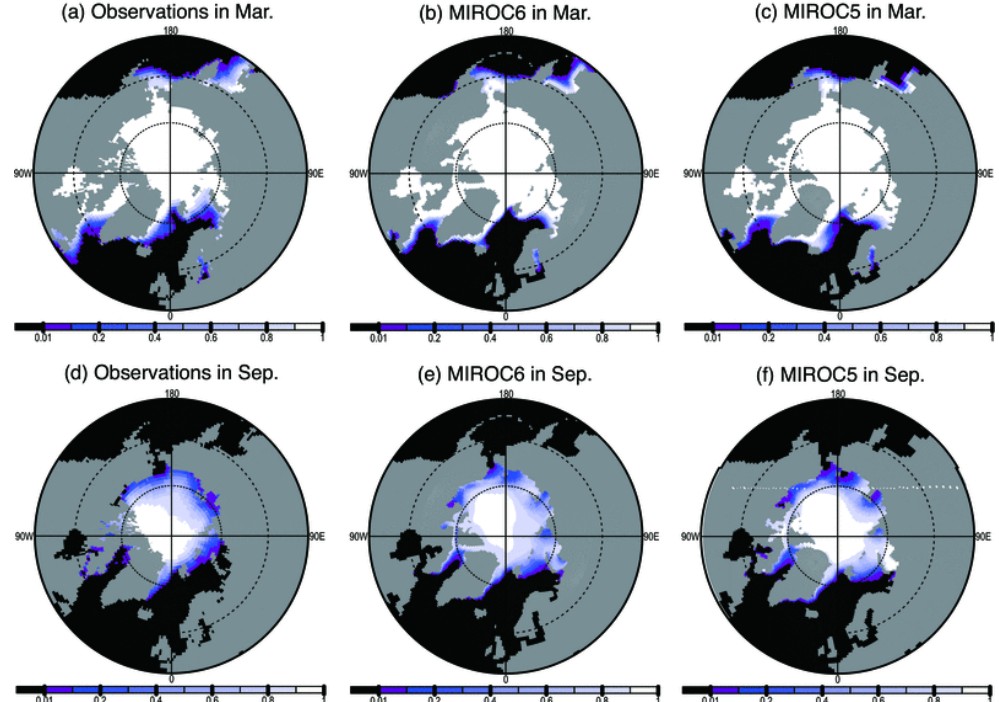


Fig. 17. Northern Hemisphere sea-ice concentrations in March (upper panels) and September (lower
panels) for observations (left), MIROC6 (middle), and MIROC5 (right). The unit is non-dimensional.
Satellite-based sea-ice concentration data of the SSM/I are used as observations.



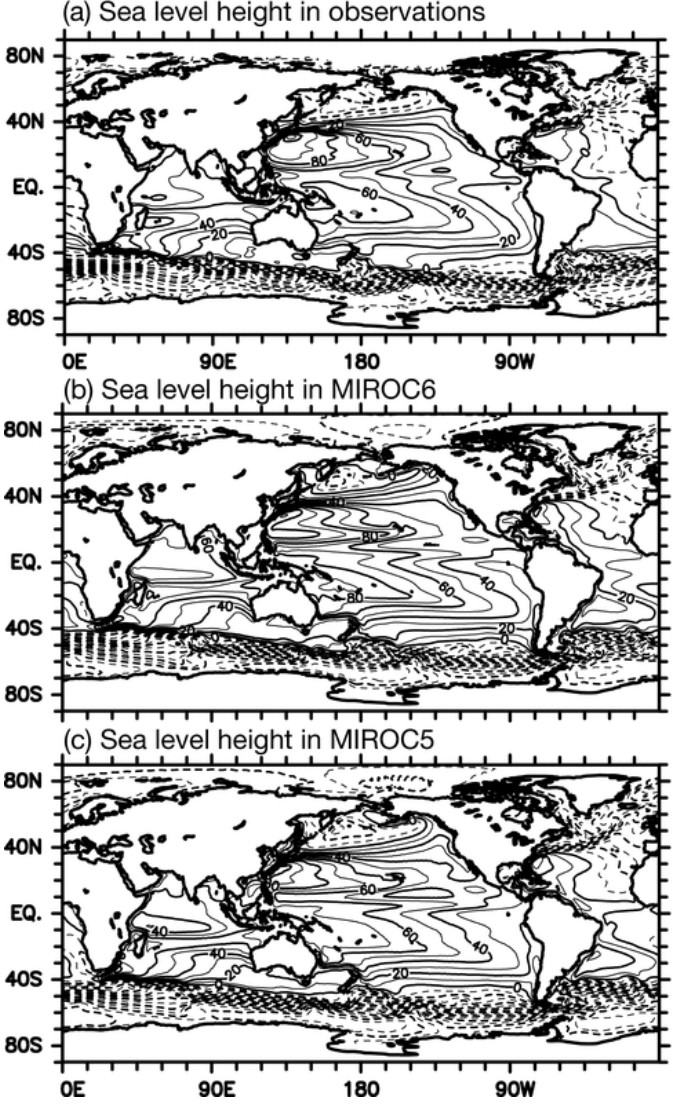


Fig. 18. Annual-mean sea level height relative to the geoid in (a) observations, (b) MIROC6, and (c)
MIROC5. Contour interval is 20 cm. Negative values are denoted by dashed lines. Note that loading
due to sea-ice and accumulated snow on sea-ice are removed from the model sea level height and that
the global-mean value is eliminated.


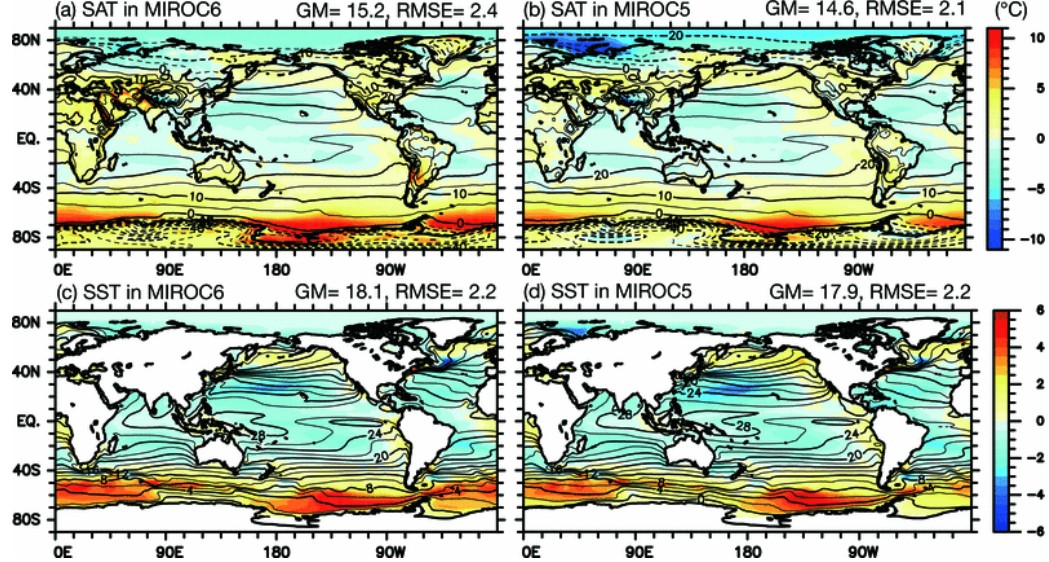


Fig. 19. Same as Fig. 4, but for annual-mean SAT (upper panels) and SST (lower panels). ERA-I for
the SAT and the ProjD for the SST are used as observations.



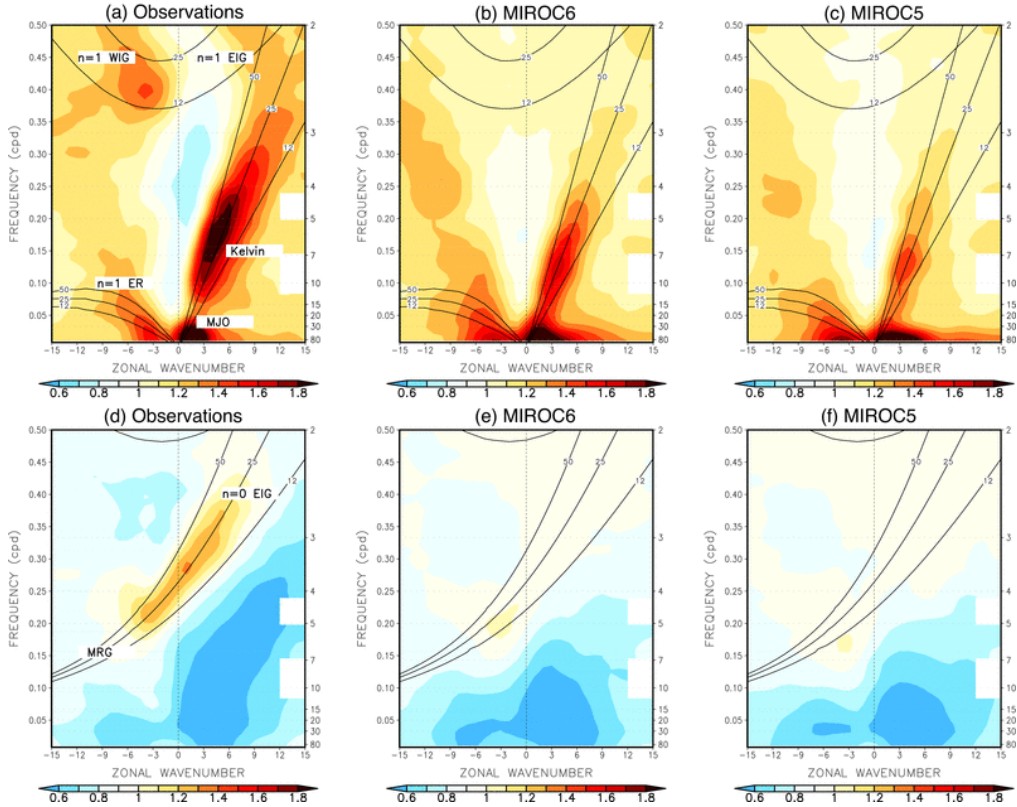


Fig. 20. Zonal wavenumber–frequency power spectra of the (a-c) symmetric and (d-f) antisymmetric
component of OLR divided by background power in (a, d) observations (NOAA OLR), (b, e) MIROC6,
and (c, f) MIROC5. Dispersion curves of equatorial waves for the three equivalent depths of 12, 25,
and 50 m are indicated by black lines. Signals corresponding to the westward and eastward inertia-
gravity (WIG and EIG) waves, the equatorial Rossby (ER) waves, equatorial Kelvin waves, the mix-
ed Rossby-gravity waves (MRG), and Madden-Julian oscillation (MJO) are labeled in (a). The unit of
the vertical axes is cycle per day (cpd).


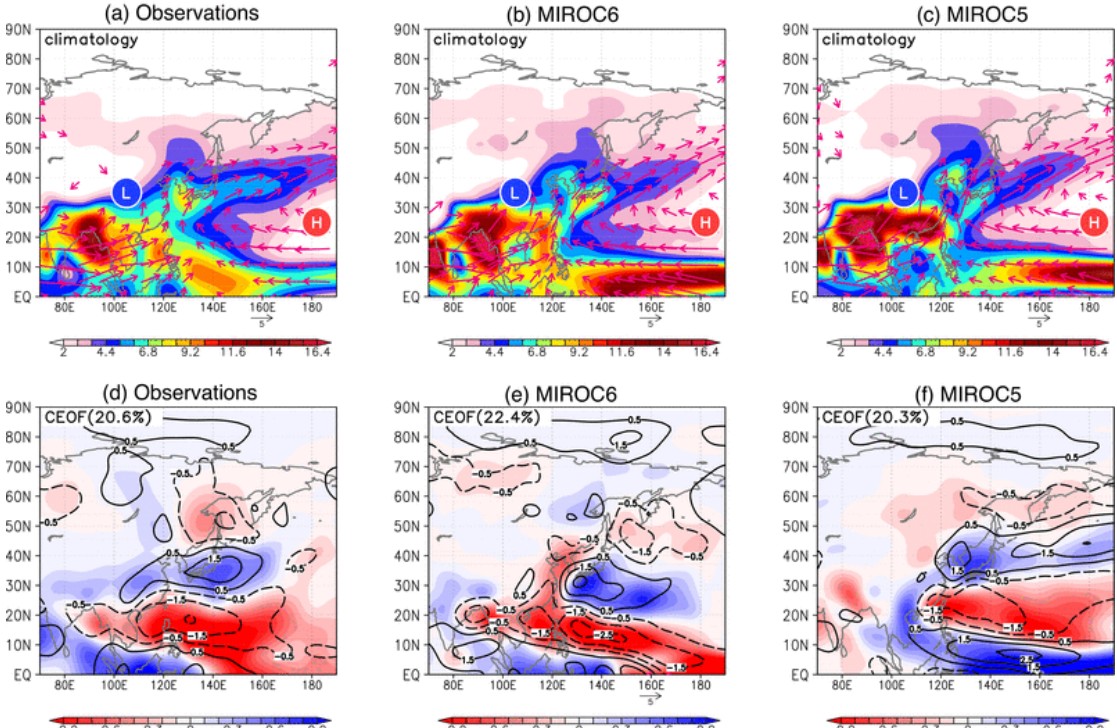


Fig. 21. (a-c) Summertime (JJA) climatology of precipitation (colors, mm day$^{-1}$) and the 850 hPa
horizontal wind (vector; m s$^{-1}$) for (a) observations (ERA-I), (b) MIROC6, and (c) MIROC5. (d-f)
Anomalies of summertime precipitation (shading; mm day$^{-1}$) and the 850 hPa vorticity (contour; 10$^{-6}$
s$^{-1}$) regressed to the time series of EOF1 of the 850 hPa vorticity over [100°E–150°E, 0°N–60°N] for
(d) observations, (e) MIROC6, and (f) MIROC5.

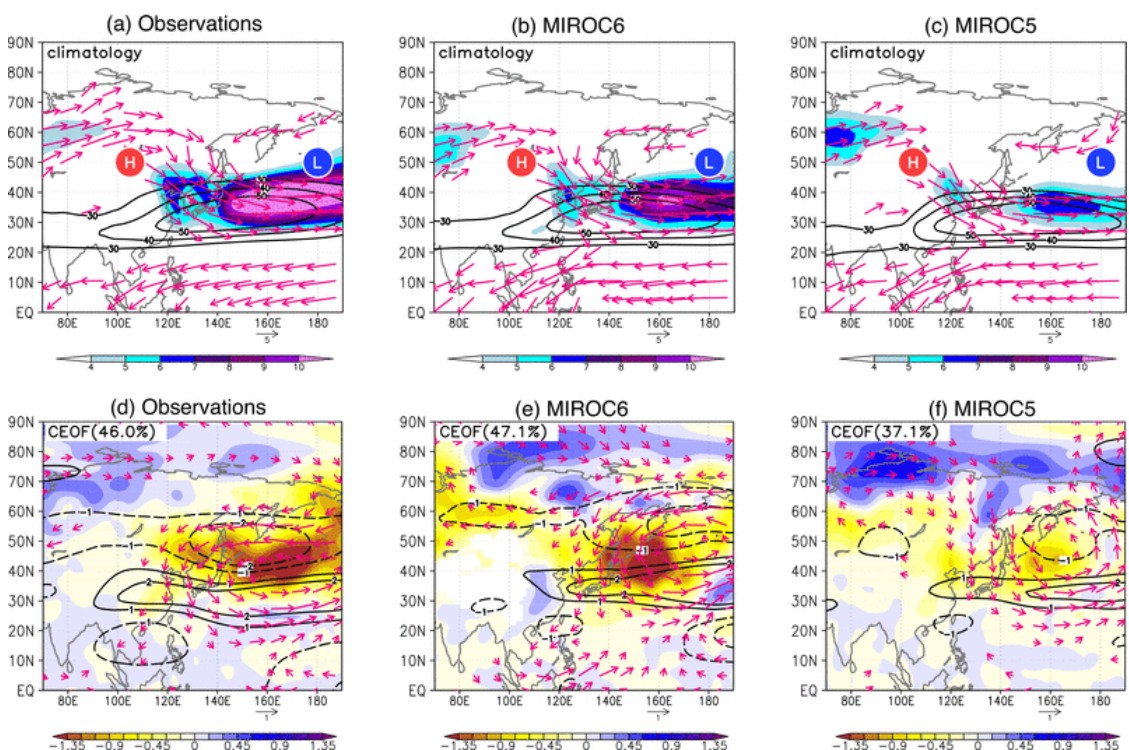


Fig. 22. (a-c) Wintertime (DJF) climatology of STA (colors; K m s$^{-1}$), the 300 hPa zonal wind (contour;
m s$^{-1}$), and the 300 hPa horizontal wind (vector; m s$^{-1}$) for (a) observations (ERA-I), (b) MIROC6, and
(c) MIROC5. (d-f) As in (a-c), but for anomalies regressed onto the time series of the EOF1 of the 850
hPa meridional wind over [120°E–150°E, 30°N–60°N].



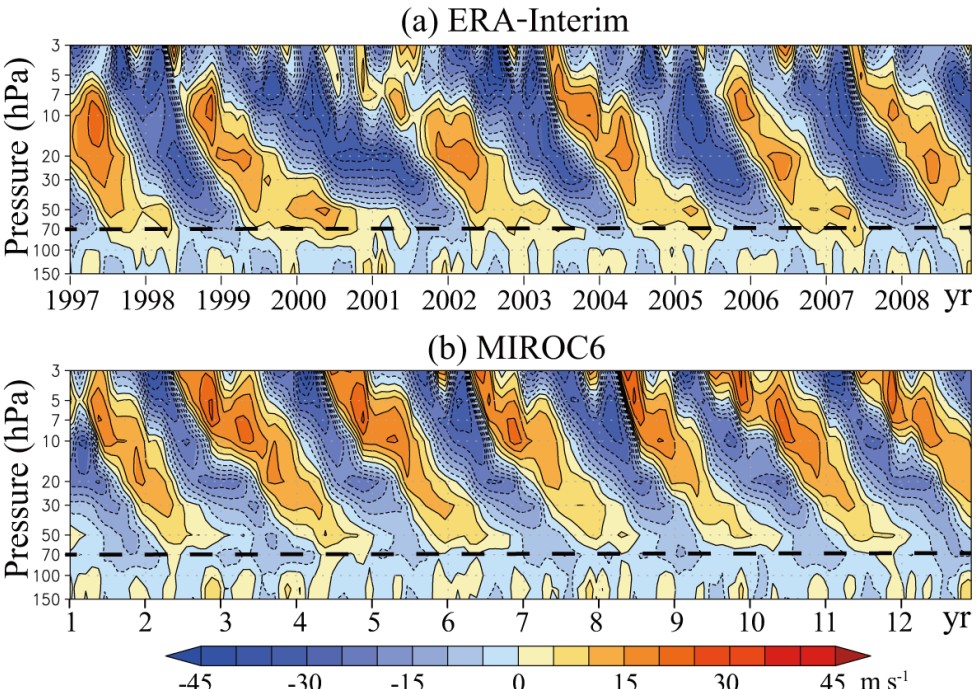


Fig. 23. Time-height cross section of the monthly mean, zonal mean zonal wind over the equator for
(a) observations (ERA-I) and (b) MIROC6. The contour intervals are 5 m s$^{-1}$. Dashed lines correspond
to the altitude of the 70 hPa pressure level. The red and blue colors correspond to westerlies and
easterlies, respectively.


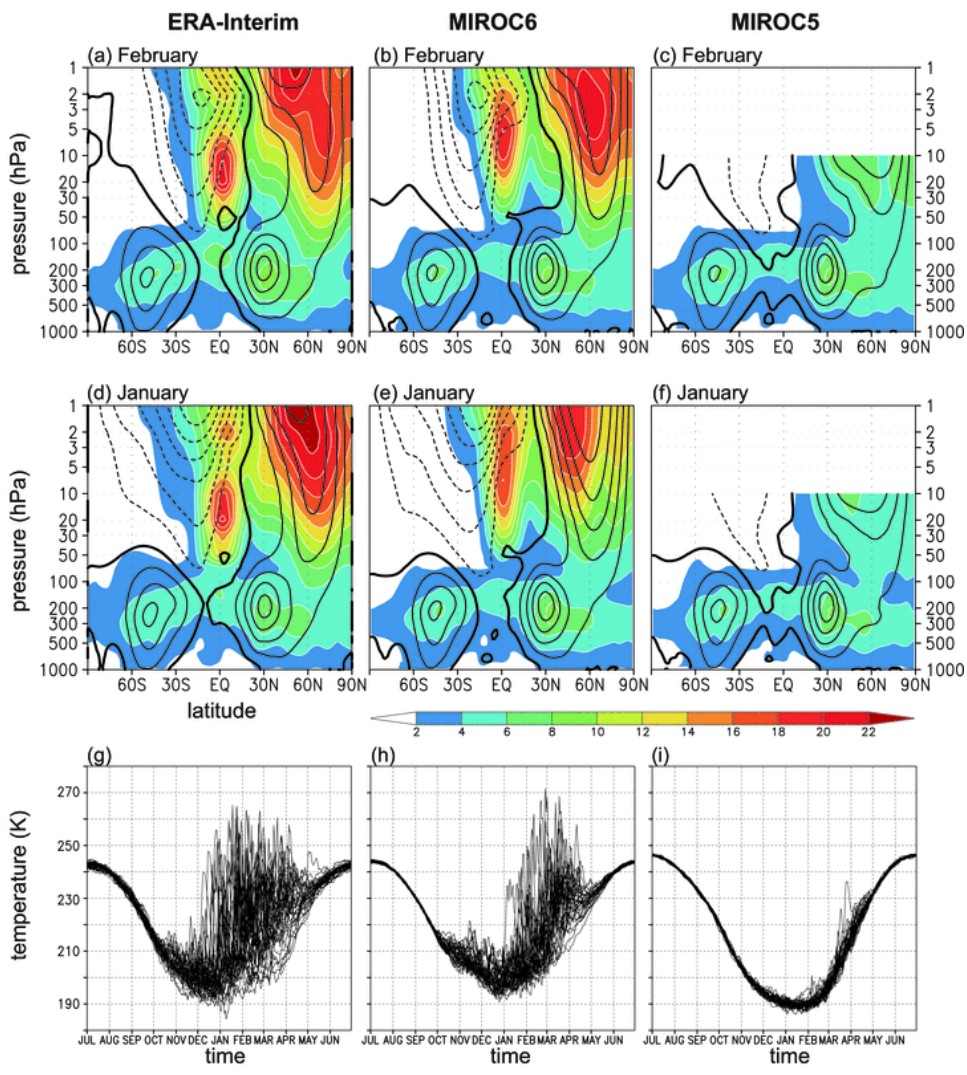


Fig. 24. Standard deviation of monthly and zonal-mean zonal wind (colors; unit is m s$^{-1}$) superimposed

on monthly climatology of zonal-mean zonal wind (black contours; unit is m s$^{-1}$) in (a-c) February and

(d-f) January for observations (ERA-I in 1979-2014; left panels), MIROC6 (middle) and MIROC5

(right) during 60-year period. In panels (g-i), the daily-mean temperatures at the 10 hPa pressure level

on the North Pole are plotted.




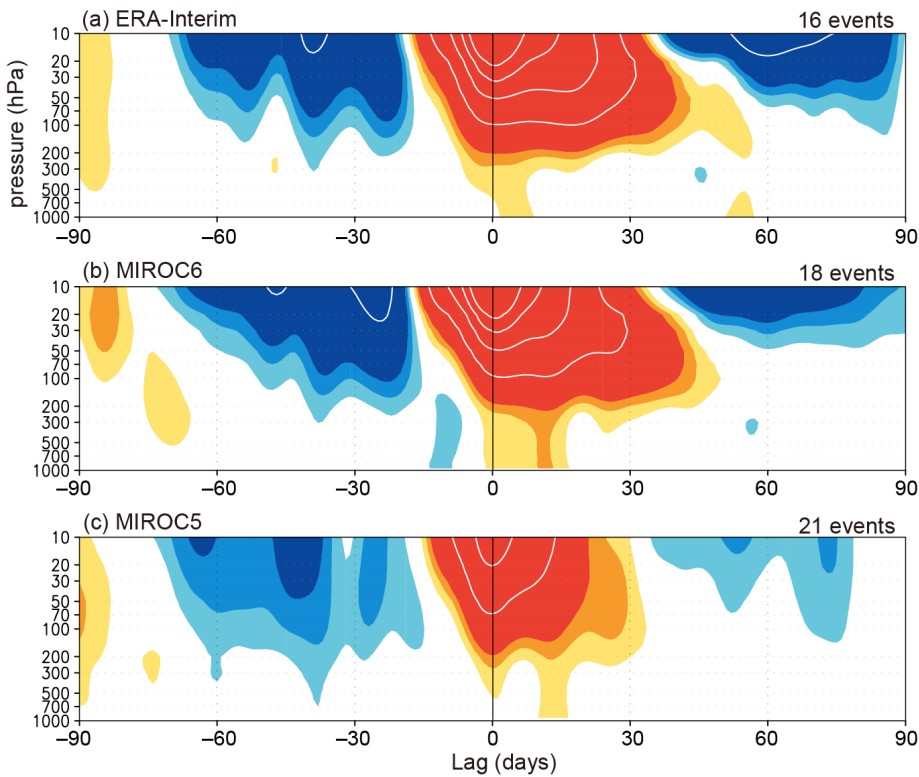


Fig. 25. Composites of time development of the zonal-mean NAM index for stratospheric weak polar
vortex events in (a) observations (ERA-I), (b) MIROC6, and (c) MIROC5. The indices having
dimension of geopotential height (m), and red colors denote negative values. Interval of colors
(contours) is 50 (400) m. The number of events included in the composite are indicated above each
panel.


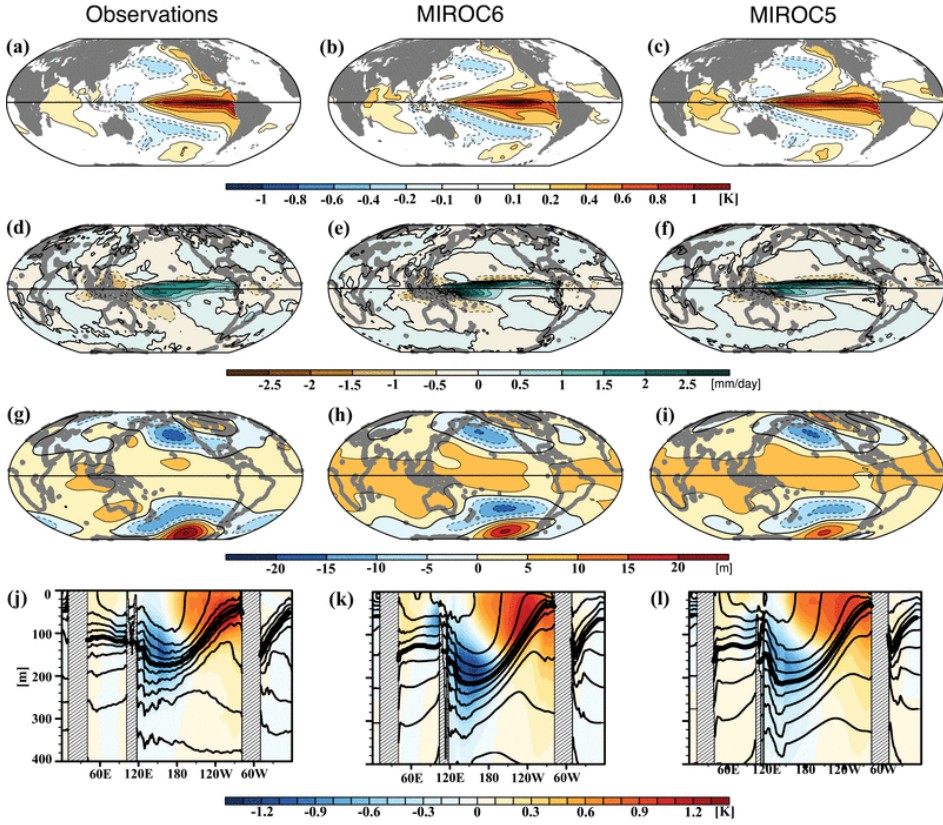


Figure 26. Anomalies of SST (K), precipitation (mm day$^{-1}$), the 500 hPa pressure height (m), and the
equatorial ocean temperature averaged in 5°S–5°N (K) which are regressed onto the Niño3 index.
Monthly anomalies with respect to monthly climatology are used here. From the left to the right, the
anomalies in observations (ProjD and ERA-I), MIROC6, and MIROC5 are aligned. In the bottom
panels, contours denote annual-mean climatological temperature with the 20°C isotherms thickened
and the contour interval is 2°C.


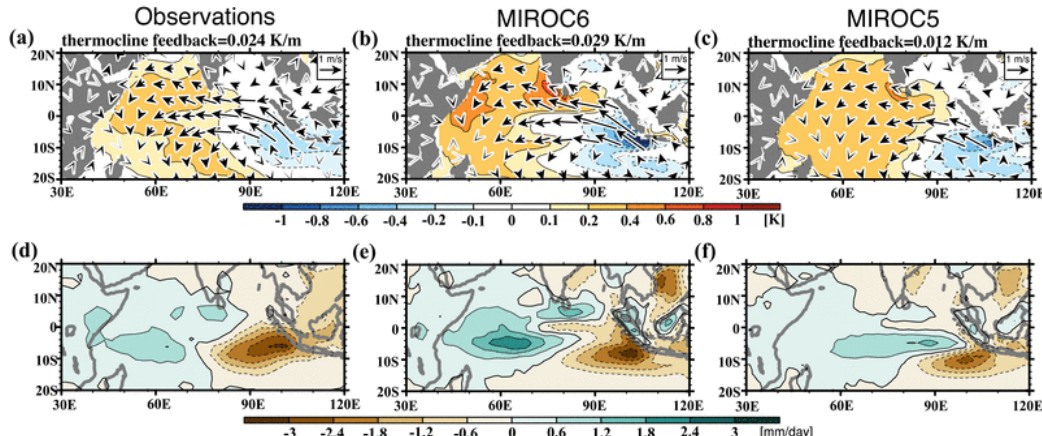


Figure 27. Same as Fig. 26, but for anomalies of SST (colors), 10 m wind vectors (upper panels) and
precipitation (lower panels) regressed onto the autumn DMI. The values of the regression slope
between anomalies of the 20°C isotherm depth and the SST over the eastern IOD region, which
indicates the thermocline feedback, are displayed on the top of the upper panels.

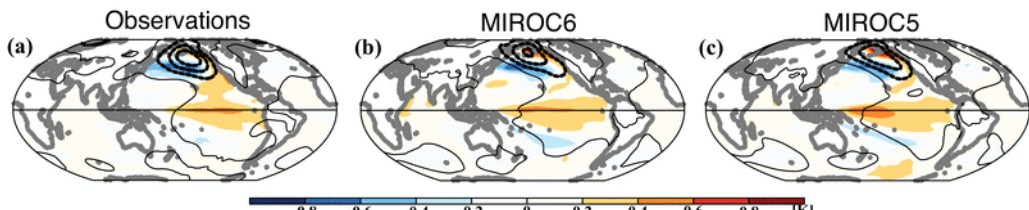


Figure 28. Same as Fig. 26, but for anomalies of monthly SST and wintertime SLP regressed onto the
PDO index (see the text). COBE-SST2/SLP2 data in 1900–2013 are used as observations.

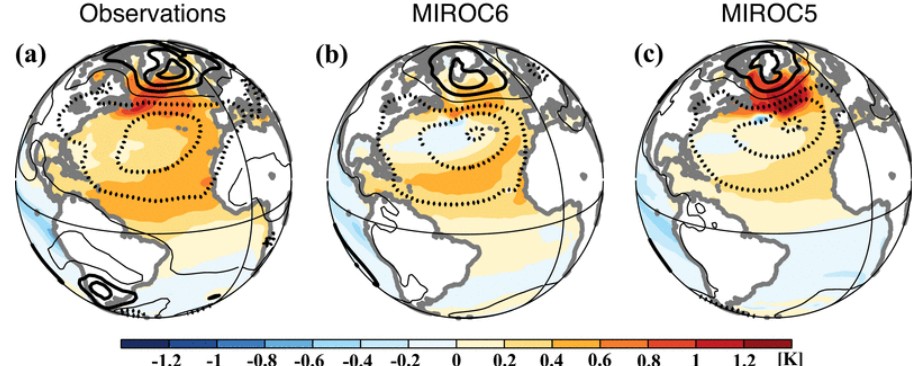

 Figure 29. Same as Fig. 26, but for anomalies of SST (colors) and SLP (contours; 0.2 hPa) regressed

 onto the AMO index (see the text). Negative values are denoted by dashed contours.

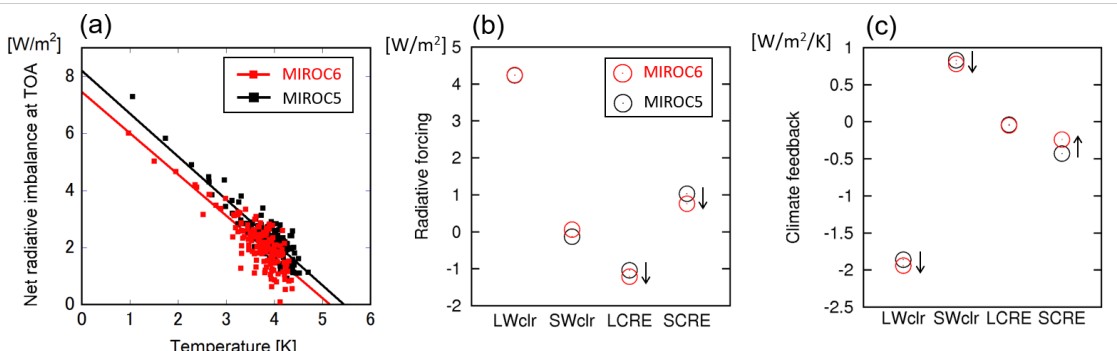

 Fig. 30. (a) Global mean net radiative imbalance at the TOA plotted against the global mean SAT

 increase. Data from the first 150 years after the abrupt $CO_2$ quadrupling are used. (b) $2 \times CO_2$ radiative

 forcing estimated by regressing four components of TOA radiation against the global-mean SAT,

 following Gregory and Webb (2008). (c) Same as (b) but for climate feedback. In Figs. 30bc, LWclr

 (SWclr) and LCRE (SCRE) denote a clear-sky longwave (shortwave) component and a longwave

 (shortwave) cloud component, respectively. The arrows in (b) and (c) indicate that the results of

 MIROC6 are different from MIROC5 at the 5% level.


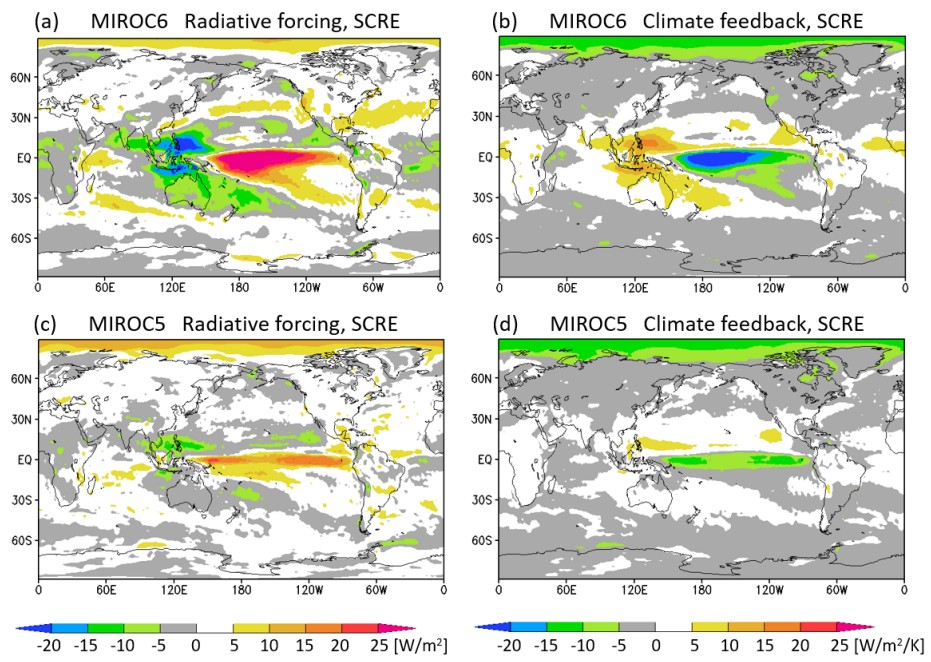


Figure 31. Shortwave cloud component of (a, c) 2 × $CO_2$ radiative forcing and (b, d) climate feedback
in MIROC6 (upper panels) and MIROC5 (lower panels).

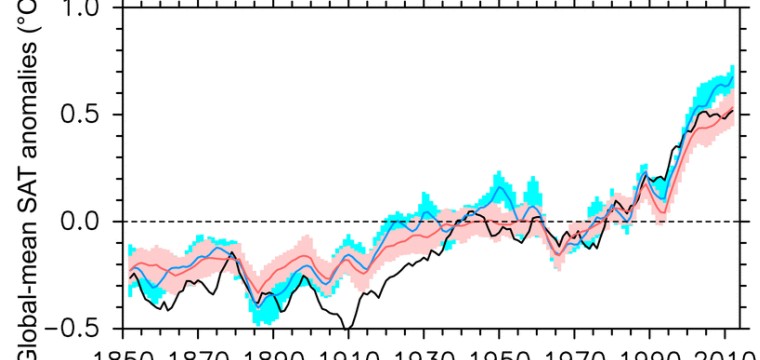


Figure 32. Time series of the global-mean SAT anomalies for observations (black), MIROC6 (red),
and MIROC5 (blue). A 5-yr running-mean filter is applied to the anomalies with respect to the 1961–
1990 mean. Colors indicate spreads of ensemble experiments for each model (1 standard deviation).



| Dataset | Used data period (year) | Reference |
|---|---|---|
| CERES (edition 2.8) | 2001–2013 | Loeb et al. (2009) |
| ISCCP | Climatology | Zhang et al. (2004) |
| ERA-Interim | 1980–2009 | Dee et al. (2011) |
| GPCPv2 | 1980–2009 | Adler et al. (2003) |
| EASE-Grid 2.0 | 1980–2009 | Brodzik and Armstrong (2013) |
| ProjD | 1980–2009 | Ishii et al. (2013) |
| SODA | 1980–2009 | Carton and Giese (2008) |
| SSM/I | 1980–2009 | Cavarieli et al. (1991) |
| NOAA OLR | 1974–2013 | Liebmann and Smith (1996) |
| COBE-SST2/SLP2 | 1900–2013 | Hirahara et al. (2014) |
| HadCRUT | 1850–2015 | Morice et al. (2012) |

Table 1. Summary of observation and reanalysis datasets used as the references in the present
manuscript.


| Model | ECS [K] | Radiative forcing [$W/m^2$] | Climate feedback [$W/m^2/K$] |
|---|---|---|---|
| MIROC6 | 2.6 | 3.72* | -1.44 |
| MIROC5 | 2.7 | 4.10 | -1.50 |

Table 2. Effective climate sensitivity (ECS), radiative forcing of $CO_2$ doubling, and climate feedback
for MIROC6 and MIROC5. The result of MIROC6 with '*' is different from MIROC5 at the 5% level.




| Model | Radiative forcing [W/m$^2$] | | | | Climate feedback [W/m$^2$/K] | | | |
|-------|-------|-------|-------|-------|-------|-------|-------|-------|
| | LWclr | SWclr | LCRE | SCRE | LWclr | SWclr | LCRE | SCRE |
| MIROC6 | 4.24 | -0.06 | -1.21* | 0.76* | -1.94* | 0.78* | -0.05 | -0.24* |
| MIROC5 | 4.23 | -0.13 | -1.04 | 1.03 | -1.86 | 0.83 | -0.04 | -0.43 |

Table 3. Radiative forcing of $CO_2$ doubling and climate feedback for MIROC6 and MIROC5,
evaluated with different components of TOA radiation as longwave clear sky (LWclr), shortwave clear
sky (SWclr), longwave cloud radiative effect (LCRE), and shortwave cloud radiative effect (SCRE).
The results of MIROC6 with '*' are different from MIROC5 at the 5% level.

**Appendix**

| | | MIROC5 (Watanabe et al., 2010) | MIROC6 (this issue) |
|---|---|---|---|
| Atmosphere | Core | CCSR-NIES AGCM (Numaguti et al., 1997) | Same as MIROC5 |
| | Resolution | T85 (150 km), 40 levels up to 3 hPa | T85 (150 km), 81 levels up to 0.004 hPa |
| | Cumulus | An entrainment plume model with multiple cloud-types (Chikira and Sugiyama, 2010) | Same as MIROC5 |
| | Shallow conv. | N/A | A mass flux-based single plume model based on Park and Bretherton (2009) |
| | Aerosol | SPRINTARS (Takemura et al., 2000, 2005, 2009) | Same as MIROC5, but with prognostic precursor gases of organic matters and diagnostic oceanic primary and secondary organic matters. |
| | Radiation | k-distribution scheme (Sekiguchi and Nakajima, 2008) | Same as MIROC5, but with a hexagonal solid column as ice particle habit and extended mode radius of cloud particles. |
| | Gravity waves | An orographic gravity wave parameterization (McFarlane, 1987) | Same as MIROC5, but with a non-orographic gravity wave parameterization (Hines, 1997) |
| Land | Core | MATSIRO (Takata et al., 2003) | Same as MIROC5, but with parameterizations for subgrid snow distribution (Linston et al. 2004; Nitta et al., 2014) and a snow-fed wetland (Nitta et al., 2017) |
| | Resolution | T85 (150 km), 3 snow layers and 6 soil layers down to 14 m depth | Same as MIROC5 |
| Ocean/sea-ice | Core | COCO4.9 (Hasumi, 2004) | Same as MIROC5 |
| | Resolution | Nominal 1.4° (bipolar grid system), 49 levels down to 5500 m | Nominal 1° (tripolar grid system), 63 levels down to 6300 m |
| | Turbulence | 1.5 level turbulent closure model (Noh and Kim, 1999) | Same as MIROC5, but modified turbulent kinetic energy input and smaller background vertical diffusivity under sea-ice (Komuro 2014) |

Table A. Summary of the updated configurations from MIROC5 to MIROC6
