# Peer review of "Description and basic evaluation of simulated mean state, internal variability, and climate sensitivity"

_Geoscientific Model Development, 2018_

## Referee Comment (RC1) · Anonymous Referee #1 · 19 Oct 2018

General comments

This paper describes MIROC6, a new climate model aiming at participating in CMIP6, by developing the previous climate model MIROC5 that participated in CMIP5. Following the description of the model formulation focusing on the changes from MIROC5 together with the model's tuning procedure, the model's mean climate and variability in the preindustrial experiment are presented. Furthermore, climate sensitivity of the model and reproducibility of the past climate change are also evaluated. Although the manuscript is comprehensive, it is well-constructed and well-documented. Climate variabilities of the model has also been widely evaluated, which brings many useful

scientific knowledges for future studies using this model. In addition, model tuning procedure is also described in detail, which contains very useful information to be helpful for climate model developers. It is recommended that it will be published after minor revisions.

Specific comments

L.355: The main parameters . . . in which the uncertainty of the climate sensitivity . . .

Does this mean that the model is tuned for a climate sensitivity as a result? If so, it is desirable to describe what is the target climate sensitivity (2.5 K?) for the tuning.

L.374: interactions between anthropogenic aerosol emissions and . . .

"emissions" do not interact with cloud-radiation processes. Do you mean "aerosol-cloud interaction"? Rephrase it.

L.380: a present-day run

Is the run a fixed SST? Since the value of –0.9 Wm–2 by IPCC (2013) is for ERF, it should be evaluated by radiation change under the condition that SST does not change. Please explain.

L.397: the global-mean ocean temperature shows a larger trend of . . .

On average there is 1.1 Wm–2 heating. Are these trends consistent with the radiation budget?

L.477: consistent with the observed value of –0.81 Wm–2.

The observed value is –0.8 Wm–2 because the system is warming in the present-day conditions. Ideally it should be 0 W m–2 in the preindustrial conditions. The radiation imbalance of –1.1 Wm–2 is in the marginally acceptable range.

L.542: increase in precipitation (Figs. 8ce)

Increase in precipitation is found only in the North Pacific.

Fig. 13 and 14

It is easy to understand if the biases are indicated by color shadings.

L.595: the Pacific sector (Figs. 13a-c)

→ "the Atlantic sector (Figs. 13a-c)" or "the Pacific sector (Figs. 14a-c)"

L.622: better representation of cloud physics

How does cloud physics relate to trade wind? It seems to me that they are incoherent.

L.648: present-day conditions

Specify the years of the observation. (1980-2009)?

Figure 18

Adding a plot for the observed sea surface height will be helpful.

L.687: strengthening of the Aleutian low lead to increase in southward transport . . .

I could not understand why the strengthening of the Aleutian low lead to increase in southward transport along the west coast.

L.919: first 20 years

By the CMIP6 protocol, 150 year-long simulations are requested. ECS may change according to the length of analysis period. Describe why you made analysis for the first 20 years.

L.939: are consistent with . . .

→ "are correlated with"

L.998: subarctic (tropical) region are underestimated (overestimated) in MIROC6 (MIROC5)

"subarctic (tropical) region are underestimated in MIROC6 (MIROC5)" or "subarctic

region are underestimated (overestimated) in MIROC6 (MIROC5)"

L.1053: which is consistent with . . . in observations.

→ which is in the acceptable range.

Technical corrections

L.185: is insufficient

→ delete.

L.229: in order to to

→ in order to

L.433: , which has a shallow . . .

It is unnecessary as it already described in section 2.1.

L.481: 2.9 (3.1) Wm-2 in MIROC5.

→ 2.9 (–3.1) Wm–2 in MIROC5.

L.490: better simulated in MIROC5

→ better simulated in MIROC6

L.922: -1.5 Wm–2

→ –1.5 Wm–2K–1

L.987: , qualitatively

→ delete.

---

## Referee Comment (RC2) · Anonymous Referee #2 · 19 Feb 2019

The authors describe in this manuscript version 6 of the climate model MIROC and its performance. Model description papers are useful to provide information that may be needed in future more science-oriented publications based on simulations with the respective model. Selecting the material for a model description paper is however a difficult task because it is clear that the information will always be insufficient to recreate the model from the description. I think that the authors present in general an appropriate selection of material. They only mention details of the model and its parameterizations where there are differences from the predecessor MIROC5, and they provide more or less typical evaluations of the simulated mean climatologies and variability based mostly on a pre-industrial control simulation. I appreciate that the authors

describe the tuning procedure applied for arriving at the final model configuration. The presentation is in general clear and the use of language appropriate. I would recommend the editors to check in particular the use of articles, however. In general I would recommend publication of the article after introduction of minor revisions which I will list in the following.

L34 It's not clear to me what "directly resolved stratosphere" means. Could one resolve it indirectly?

Introduction: I would recommend to shorten the introduction. This is a paper for specialists who know about global warming, IPCC, and the purposes of climate models. It would be good to report which specific goals the MIROC6 development had but I would cut the general introduction.

L42ff ". . . has been already observed that . . . will drastically change" Rephrase if you want to keep this sentence.

L46 Not clear what "will increase" means e.g. for tropical cyclones. In size? Its number? Its strength?

L56ff Why are only the two most recent ARs mentioned (if one wants to mention them at all)?

L64 Is there such a consensus that "sophisticating . . . parameterizations . . . are necessary" "to reduce uncertainties . . . in climate projections"?

L82ff What means "K-1 model developers"

L116ff As before: How may improvement of parameterizations "may result in reducing uncertainty"?

L126ff The sentence on the "signal-to-noise ratio" is difficult to understand.

L139/140 I'd try to avoid terms like high-resolution" or "medium resolution" The notion of what is high, medium o low is very different among climate modelers and cer-

tainly changes over time. L151 I suggest to add a sentence on how MIROC relates to MIROC-ESM (which is referred to in Section 3.2.2). In my understanding MIROC6 is a climate model that concentrates on the physical part of the Earth system and it would be useful to mention that because many of the models used in CMIPs these days include some component cycles.

Section 2: I'm missing some technical information in the model description. I guess that in particular time steps (atmosphere, ocean, coupling, exceptions for specific parameterizations) had to be changed in comparison to MIROC5.

Fig. 1: Do the marks indicate half levels or full levels or what else?

L164 Why is there a Table A1? I'd suggest to use a simple numbering for all tables.

L174 I would speak of "model top", not "TOA".

L184 Not clear what is meant with "dry air . . . is insufficient".

L224 Remove "a".

L225 Not clear why there is reference to "future versions". Does the current model version use the described features or not?

L233 Tuning of gravity wave parameterizations. Often, the Hines parameterization is used with very simplified and globally homogeneous characteristics of the gravity waves at the launching levels? This may make it, however, difficult to tune as well the QBO and high-latitude circulations. In particular as this is a feature new to MIROC it would be useful to elaborate a bit more on the tuning of GW parameters.

L244ff Example of the SSNOWD parameterization: It is useful to mention that the new parameterization is "physically" based. But in general I would like to read what the motivation for introducing changes with respect to MIROC5, what the expectations were, and, later in the evaluation, if the expectations were met. This is done well for some of the changes (e.g. L271: "increased vertical layers have been adopted in order

to .."), but less for others (e.g. the tripolar ocean grid mentioned at line 262). I would like to ask the authors to do this more consistently for the changes because it may help other modeling groups to judge if specific changes may be worthwhile to apply in other models or not.

L309 Only extinction coefficients would not allow to compute the radiative effects of aerosols.

L322 "These" instead of "This".

L328 I guess averages over these time periods are used?

L344 "land surface components are determined" sounds odd. I guess they are interpolated from some dataset. Please specify.

L350 "Reproducibility" of what?

L362 Here, the first tuning step for the coupled model is described. But there needs to be some procedure to specify tuning parameters in the component models, or not? Furthermore, I think it would be very useful for other modellers to be more specific about which parameters have been tuned to which effect. In some places this is described in acceptable detail, but in particular the first two sentences of this paragraph are very vague.

L376 Which "cooling effects" are meant, here? Aerosol-cloud effects as mentioned in the sentence before? Or total aerosol effects? I'd also prefer to speak of "radiative forcing" instead of cooling effects.

L402 Apparently global mean SAT is not a tuning goal. It might be useful to mention this, and also why not. Additionally I'd like to read a comment on the imbalance of about 1 W/m2 that seems to exist in equilibrium. It seems like there is some artificial energy source in the model. Is there any knowledge where this originates from? Atmosphere, ocean, dynamical core, specific parameterizations? Is it known if this changes with the model state?

Fig. 4 and corresponding text: Names for the TOA fluxes are confusing. What is called NET is actually the total net flux, while what is called OSR is the net SW flux.

L426 The sentence on the consideration of "global-mean values" for RMSE calculations is difficult to understand. Maybe provide a formula or clearer description on how the RMSE are calculated? Is that true for all RMSE in this manuscript? It would be good to mention in every caption of Figures where RMSE are presented how these values are calculated, i.e. in particular if a global or some other mean have been subtracted before calculation of the error. The OLR in Fig. 4 looks particularly confusing without such information because while the RMSE is smaller in MIROC6 than in MIROC5 one would guess otherwise from the color shading because of the dominance of red in the case of MIROC6.

L476 I accept that for many climate variables it may not be essential if the evaluation is done for a pre-industrial o present-day simulation. But for some it is crucial. The energy balance is such a case, because the total net TOA flux should be zero in equilibrium. One can't say that the imbalance in the models is consistent with some observed imbalance, because the latter is related to the system not being in balance currently. It is also necessary to provide a reference for the observed value.

Fig. 4 and others. Parts of this and other figures are very blurred. This should be improved for the final publication.

L512ff The region of the western tropical Pacific is singled out as a region of improvement in MIROC6. It should be mentioned that it seems that in other regions there is a clear worsening.

L518ff For the discussion of the upper stratospheric warm bias it also matters that present-day and pre-industrial are compared due to the known stratospheric cooling with increased GHGs and reduced ozone.

L519: Again, I would prefer to speak of model top or lid and not of TOA.

[Figure]

L522 It would be good to say that this is the stream function of zonal mean meridional winds and not of residual winds, I guess.

L524 Please rephrase this sentence.

L530 I guess the absorption of LW radiation plays a minor role compared to SW radiation.

L561 "extend", not "extends"

L561f Not clear what "more active troposphere-stratosphere" interactions are supposed to mean (radiative, wave coupling, trace gas exchange?) and why the stream functions would indicate that.

L566ff There is no "remarkable improvement" in May. Furthermore, I suggest to avoid subjective terms like "remarkable". Please check all the text.

L586 "into which cold an dense water forms" Please rephrase.

L609ff This is no sentence.

L612 The caption of Fig. 15 says only "temperature" while here you speak of "potential temperature".

L614 Please rephrase "is risen".

L623 "Better representation of cloud physics" would be "required" for what?

L638 Remove "to".

L648 It would be useful to check if the historical simulation shows more realistic numbers in the comparison to [resent-day sea ice.

L649ff Is there any idea why Antarctic sea ice is strongly underestimated?

L655 Please specify what "sea level height" is presented in Fig. 18 and to which data it. Is compared in the text.

L676 "land surface variables" Actually I identify only one: snow cover.

L678 Again, global mean SST and SAT are variables for which the comparison of preindustrial simulations and present-day observations is misleading.

L695ff A prominent feature of SST and SAT biases is the strong warm bias close to Antarctica. This should be mentioned when discussing these variables.

L718ff It might be good to also mention the apparently missing features WIG and possibly EIG in the models. And what about anti-symmetric waves?

L727 Remove "and".

L743 Remove "MIROC6"

L760 "become" Rephrase.

L778 MIROC-ESM should be introduced in the beginning (or here)

L778 "whic"

L787 Remove "that"

L787 What means "correlations . . . are not clear"? Insignificant? Small?

L793f SSWs are only a typical feature of the NH stratosphere.

Fig. 24 and its discussion. I'd find it helpful to add Figures for January to make clear also the deficiencies of the model.

L807ff One can't evaluate the polar night jet in Fig. 7e because it shows annual means. It might actually be an option to add some seasonal wind fields in Fig. 7. The paper has many figures anyhow, so I wouldn't mind adding a few more. Additionally there is a problem of chicken and egg with the wave-mean flow interaction mentioned here.

L846 Rephrase "existence depths".

L885 "SLP anomalies are larger and better represented in MIROC6" The maximum is

deeper, but otherwise I find it hard to judge which of the two models is better.

L889 I'd avoid words like "excessively" which are subjective statements.

L918ff In IPCC AR5 and also the paper by Andrews et al. (2012) which is cited, here, climate sensitivity and forcing are calculated from 150 years of the 4xCO2 simulation, not 20 years. I would suggest to follow this 150-year standard to ensure comparability. Some models show clear non-linearities during this period. It seems like the effect is relatively small in MIROC5, but this would need to be confirmed for MIROC6.

Tables 2 and 3: It would be convenient for the reader to combine the tables.

L966,979 Again, please avoid "remarkably".

L999 Why would ECS quantify uncertainty?

L1028f It's true that the hiatus is sometimes associated with the IPO, but there are plenty of other attempts to explain it and even arguments that the real reason maybe unidentifiable. So I'd suggest to not only mention the IPO.

L1031ff I don't understand this sentence. I agree that the simulated hiatus could be spurious, but the argument of the ensemble mean wouldn't support this.

L1055 Should a new paragraph start here?

L1068ff I have no idea why the final sentence suddenly makes a statement concerning component cycles which were not at all mentioned anywhere else in the text.

L1074 I don't know what the policy of GMD is concerning the availability of primary data (which I think should be the code of the model and all input data needed to redo the experiments), but I find it problematic that the code is only available under the condition of "collaborative research". As mentioned in my initial statement, a model description is necessarily incomplete. It can only be completed by the model code.

---

## Author Comment (AC1) · 15 May 2019

We would like to thank the reviewer for taking the time to carefully read our manuscript, for several very valuable suggestions and English grammatical corrections. We have much revised our manuscript and answered all the comments given by the reviewer. In the separate reply letter uploaded as a supplement, point-by-point responses to the reviewer's comments and how we revised the manuscript are described, referring to the revised manuscript and the manuscript with revision history which were also uploaded as supplements.

[Figure]

Please also note the supplement to this comment:
https://www.geosci-model-dev-discuss.net/gmd-2018-155/gmd-2018-155-AC1-supplement.zip

---

## Author Response (AR1)

Dear the reviewer for the manuscript entitled "Description and basic evaluation of simulated mean state, internal variability, and climate sensitivity in MIROC6" by Tatebe et al.

We would like to thank the reviewer for taking the time to carefully read our manuscript, for several very valuable suggestions and English grammatical corrections. We have much revised our manuscript and answered all the comments given by the reviewer. In the separate reply letter uploaded as a supplement, point-by-point responses to the reviewer's comments and how we revised the manuscript are described, referring to the revised manuscript and the manuscript with revision history which were also uploaded as supplements.

**Hiroaki Tatebe**

Research Center for Environmental Modeling and Application,

Japan Agency for Marine-Earth Science and Technology

3173-25 Showamachi, Kanazawaku, Yokohama, Kanagawa 236-0001, Japan

Response to reviewers' comments on " Description and basic evaluation of simulated mean state, internal variability, and climate sensitivity in MIROC6" by Tatebe et al.

**Reply to the reviewer #1**

**General comments**

This paper describes MIROC6, a new climate model aiming at participating in CMIP6, by developing the previous climate model MIROC5 that participated in CMIP5. Following the description of the model formulation focusing on the changes from MIROC5 together with the model's tuning procedure, the model's mean climate and variability in the preindustrial experiment are presented. Furthermore, climate sensitivity of the model and reproducibility of the past climate change are also evaluated. Although the manuscript is comprehensive, it is well-constructed and well-documented. Climate variabilities of the model has also been widely evaluated, which brings many useful scientific knowledges for future studies using this model. In addition, model tuning procedure is also described in detail, which contains very useful information to be helpful for climate model developers. It is recommended that it will be published after minor revisions.

We would like to thank the reviewer for taking the time to carefully read our manuscript, for several very valuable suggestions, and English grammatical corrections. We would like to answer the questions given by the reviewer and to describe how we have revised our manuscript point by point. Please note that our replies are written in red letters in this reply letter.

**Reply to specific comments**

L.355: The main parameters...in which the uncertainty of the climate sensitivity...

Does this mean that the model is tuned for a climate sensitivity as a result? If so, it is desirable to describe what is the target climate sensitivity (2.5 K?) for the tuning.

Here, the authors just wanted to mention that parameters listed in Shiogama et al. (2012) are mainly used for a tuning procedure. Climate sensitivity was not a tuning target. In the revised manuscript, we have rephrased the sentence as "The main parameters used in our tuning procedures are chosen referring to a perturbed parameter ensemble set made by Shiogama et al. (2012) in which parameter sensitivity to cloud-radiative processes is examined". Please see the lines 374 -377 in the revised manuscript.

L.374: interactions between anthropogenic aerosol emissions and...

"emissions" do not interact with cloud-radiation processes. Do you mean "aerosolcloud interaction"? Rephrase it.

In the revised manuscript, the words are replaced by "aerosol-cloud interaction". Please see the lines 402-403. Thank you for your suggestion.

**L.380: a present-day run**

Is the run a fixed SST? Since the value of  $-0.9 \text{ Wm}^{-2}$  by IPCC (2013) is for ERF, it should be evaluated by radiation change under the condition that SST does not change. Please explain.

The tuning was done under a coupled mode, namely, SST is not fixed. In the revised manuscript, we added the sentence "Note that MIROC6 in a coupled mode is used in this tuning procedure, and thus the sea surface temperature (SST) is not fixed. The estimated cooling effects here are not strictly the same as the effective radiative forcing estimated in IPCC (2013). However, by the present tuning procedure, the global-mean surface air temperature (SAT) change after the mid-19th century is well reproduced in the historical runs by MIROC6 (details are discussed in Section 4).". Please read the lines 411-416.

L.397: the global-mean ocean temperature shows a larger trend of...

On average there is 1.1 Wm-2 heating. Are these trends consistent with the radiation budget?

Discussions on the relationship between the warming trend of the ocean temperature and the TOA radiation budget/ocean heat uptake have been added in the revised manuscript. And we also added the explanation on the heat energy inconsistency between the TOA radiation budget and the ocean heat uptake in association with the model imperfection. We have rephrased the last paragraph of Section 2.5 as "The trend of the global-mean ocean temperature in the later period suggests slight but continuous warming of the deep ocean. The radiation budget at the TOA is 1.1 Wm-2 downward on average (linear trend of  $9.5 \times 10^{-3}$  K/100 yr) and the net heat input at the sea surface is 0.32 Wm-2. The deep ocean warming is explained by the net heat input. Note that there is about 0.78 Wm-2 inconsistency between the TOA radiation budget and the ocean heat uptake. This heat energy inconsistency is due to that internal energy associated with precipitation, water vapor and river runoff is not taken account in the atmospheric and land surface component in MIROC6 and that these waters with no temperature information implicitly set their temperature to the SST when they flow or fall into the ocean. Perpetual melting of the prescribed Antarctic region (details will be discussed in Section 3.1.3), is also a cause of the heat energy inconsistency".

L.477: consistent with the observed value of -0.81 Wm-2.

The observed value is  $-0.8 \text{ Wm}^{-2}$  because the system is warming in the present-day conditions. Ideally it should be  $0 \text{ Wm}^{-2}$  in the preindustrial conditions. The radiation imbalance of  $-1.1 \text{ Wm}^{-2}$  is in the marginally acceptable range.

Thank you for your comment and the authors agree with the reviewer. In the revised manuscript, we added the sentence "However, the observed value is estimated in the present-day condition. Ideally, the model value in the preindustrial condition should be 0  $Wm^{-2}$  and is in the marginally acceptable range". Please read the lines 521-523.

L.542: increase in precipitation (Figs. 8ce)

Increase in precipitation is found only in the North Pacific.

In the revised manuscript, the corresponding sentence is rewritten as "is accompanied by an associated increase in precipitation, especially in the North Pacific (Figs. 8ce).". Please read the lines 594-595.

Fig. 13 and 14: It is easy to understand if the biases are indicated by color shadings.

Following the comments, we have redrawn Figs. 13 and 14, and corresponding descriptions on ocean climatological hydrography have been rephrased partly in the revised manuscript. Please read the 1st and 2nd paragraphs of Section 3.1.2. Also, the revised manuscript with revision history is useful for checking the revision.

L.595: the Pacific sector (Figs. 13a-c)  $\rightarrow$  "the Atlantic sector (Figs. 13a-c)" or "the Pacific sector (Figs. 14a-c)"

Carefully checking zonal-mean ocean temperature and salinity in the Pacific sector, the authors considered that representation of the northward intrusion of Antarctic Intermediate Water in the Southern Hemisphere in MIROC6 is not better than in MIROC5. In the revised manuscript, we deleted the sentence "Meanwhile, the northward intrusion of Antarctic Intermediate Water in the Southern Hemisphere around the 1000 m depth is better simulated in MIROC6 than in MIROC5, especially in the Pacific sector (Figs. 14a-c)".

L.622: better representation of cloud physics

How does cloud physics relate to trade wind? It seems to me that they are incoherent.

In the revised manuscript, we have descried the details about the relationship between the stronger trade

wind and cumulus processes referring to the stand-alone AGCM experiments as "However, the thermocline depths in the western tropical Pacific are still larger in the models than in observations and are attributed to the stronger trade winds in the models. When both of MIROC6 and MIROC5 are executed as stand-alone AGCMs with the prescribed SST obtained from observations, the overestimate of the equatorial trade winds also appears due to overestimate of the upward winds over the maritime continent associated with deep cumulus convection and the resultant strengthening of the Walker circulation over the equatorial Pacific. Better parameterizing deep cumulus convection in the models could be required". Please read the lines 677-684.

L.648: present-day conditions. Specify the years of the observation. (1980-2009)?

In the revised manuscript, the years are specified as "while observations are taken in present-day conditions of 1980–2009...". Please read the lines 7070-711.

Figure 18: Adding a plot for the observed sea surface height will be helpful.

A figure of observed sea level height has been added as Fig. 18a and the reference for the observation data has been written in References (please see Rio et al. 2014).

L.687: strengthening of the Aleutian low lead to increase in southward transport...

I could not understand why the strengthening of the Aleutian low lead to increase in southward transport along the west coast.

We have rephrased the corresponding sentence as "Warm SAT and SST biases along the west coast of the North America are smaller in MIROC6 than in MIROC5. The reason is that an increase of southeastward Ekman transport in the eastern subarctic North Pacific due to the strengthening of the mid-latitude westerly jet (Fig. 10) and the Aleutian low tend to cancel out the relatively warm water supply from the subtropics to the subarctic region by the surface geostrophic current". Please read the lines 7555-759.

**L.919: first 20 years**

By the CMIP6 protocol, 150 year-long simulations are requested. ECS may change according to the length of analysis period. Describe why you made analysis for the first 20 years.

The authors agree that analysis for the first 20 years is not consistent with the CMIP6 protocol. Following the comment, we repeated the analysis using the first 150-yr-long data, and confirmed that the results were similar to the ones based on the first 20-yr-long data. The manuscript is updated based on the present

analysis. Please read Section 3.3 of the revised manuscript. Also, Figures 30, 31 and Tables 2 and 3 have been replaced by the revised ones.

L.939: are consistent with... $\rightarrow$  "are correlated with"

In the revised manuscript, "are consistent with.." was replaced by "are correlated with". Please read the lines 1015-1019.

L.998: subarctic (tropical) region are underestimated (overestimated) in MIROC6 (MIROC5)

"subarctic (tropical) region are underestimated in MIROC6 (MIROC5)" or "subarctic region are underestimated (overestimated) in MIROC6 (MIROC5)"

Following the reviewer's comment, we have rewritten the sentence as "Signals associated with AMO in the subarctic (tropical) region are underestimated in MIROC6 (MIROC5)". Please read the lines 1073-1074. Thank you very much.

L.1053: which is consistent with...in observations  $\rightarrow$  which is in the acceptable range.

In the revised manuscript, the corresponding sentence is rewritten as "the global TOA radiation imbalance in MIROC6 is about  $-1.1 \text{ Wm}^{-2}$ , which is in the acceptable range of observations". Please see the lines 1128-1130.

**Technical corrections**

- L.185: is insufficient  $\rightarrow$  delete : Deleted.
- L.229: in order to to  $\rightarrow$  in order to : Fixed.
- L.433: , which has a shallow...: It is unnecessary as it already described in section 2.1.

The corresponding sentence was deleted in the revised manuscript.

L.481: 2.9 (3.1) Wm-2 in MIROC5  $\rightarrow$  2.9 (-3.1) Wm-2 in MIROC5: We have added "-" in front of "3.1".

L.490: better simulated in MIROC5  $\rightarrow$  better simulated in MIROC6 : Fixed.

L.922: -1.5  $\text{Wm}^{-2} \rightarrow -1.5 \text{ Wm}^{-2}\text{K}^{-1}$  :Fixed. Thank you for your comment.

L.987: , qualitatively  $\rightarrow$  delete. : Deleted.

Response to reviewers' comments on " Description and basic evaluation of simulated mean state, internal variability, and climate sensitivity in MIROC6" by Tatebe et al.

**Reply to the reviewer #2**

**General comments**

The authors describe in this manuscript version 6 of the climate model MIROC and its performance. Model description papers are useful to provide information that may be needed in future more science-oriented publications based on simulations with the respective model. Selecting the material for a model description paper is however a difficult task because it is clear that the information will always be insufficient to recreate the model from the description. I think that the authors present in general an appropriate selection of material. They only mention details of the model and its parameterizations where there are differences from the predecessor MIROC5, and they provide more or less typical evaluations of the simulated mean climatologies and variability based mostly on a pre-industrial control simulation. I appreciate that the authors describe the tuning procedure applied for arriving at the final model configuration. The presentation is in general clear and the use of language appropriate. I would recommend the editors to check in particular the use of articles, however. In general I would recommend publication of the article after introduction of minor revisions which I will list in the following.

We would like to thank the reviewer for taking the time to carefully read our manuscript, for several very valuable suggestions, and English grammatical corrections. We would like to answer the questions given by the reviewer and to describe how we have revised our manuscript point by point. Please note that our replies are written in red letters in this reply letter.

**Reply to specific comments**

L34 It's not clear to me what "directly resolved stratosphere" means. Could one resolve it indirectly? In the revised manuscript, we rephrased it as "to the inclusion of the stratosphere". Please read the lines 34-35.

Introduction: I would recommend to shorten the introduction. This is a paper for specialists who know about global warming, IPCC, and the purposes of climate models. It would be good to report which specific goals the MIROC6 development had but I would cut the general introduction.

Thank you for your suggestions. Recently, the publication policy of GMD requires the short summary of a manuscript which is for non-specialists as well as specialist because simulation models are related scientific issues are of great concern of various fields of climate sciences and socio-economic sciences. Also, as stated in the WCRP conference in 2011, the decrease of the number of climate modelers is recognized as important problem to be solved for healthy progress of climate sciences and its application to mitigation and adaptation of human society to the changing climate (please see the slides 21-25; https://www.wcrp-climate.org/conference2011/orals/A4/Jakob A4.pdf), suggesting that the importance of recruitment of students and younger scientists to climate model developments. As well as in climate centers in the world, we, Japanese climate modelling community, have faced the same problem for promoting climate modelers. In order to make some contributions to this issue, the authors wrote comprehensive description on motivations, purposes, and history of our model development to students and young researchers who may be interested in climate sciences as well as socio-economic scientist who will use our simulation data. We would be grateful if you could understand why we wrote rather long introduction in the present manuscript. On the other hand, the authors agree with your comment. So, we have revised, shortened, and reconstructed as possible as we could. Please read the introduction in the revised manuscript. Thank you very much for your suggestion again.

L42 "...has been already observed that..will drastically change": Rephrase if you want to keep this sentence. We have rephrased the sentence as "As the global warming due to increasing emissions of the anthropogenic greenhouse gases progresses, global and regional patterns of atmospheric circulations and precipitation as well as temperature are projected to be drastically changed at the end of the twentieth-first century". Please read the 1st sentence of Section 1.

L46 Not clear what "will increase" means e.g. for tropical cyclones. In size? Its number? Its strength? We have rewritten the sentence as "occurrence frequency of extreme weather events such as heatwaves, droughts will be increased and extratropical cyclones will be stronger than in the present" in the revised manuscript. Please see the lines 46-48.

L56 Why are only the two most recent ARs mentioned (if one wants to mention them at all)? In revised manuscript, the citation of IPCC as (IPCC 2007; 2013) is deleted and IPCC (2007) is removed from the reference list. Please read the lines 57-59. L64 Is there such a consensus that "sophisticating...parameterizations...are necessary" "to reduce uncertainties...in climate projections"?

In my opinion and as described in Chapter 9 of IPCC-AR5 WG1, climate model development towards resolving various processes or representing unresolved sub-grid scale phenomena based on process-oriented understanding of physical processes could contribute to more reliable climate projections. But, in the revised manuscript, we have avoided using affirmative expressions and have rewritten the corresponding sentence as "To reduce the uncertainties and errors in climate projections and predictions, utilizing observations, extracting essences of physical processes in the real climate, and investigating the response of the climate system to various external forcings based on a set of climate model simulations are necessary. In particular, a state-of-the-art climate model which can represent various processes in the Earth's climate system is a powerful tool for deeper understanding the Earth's climate system." Please read the lines 65-70.

L82 What means "K-1 model developers"

K-1 model developers (2004) is a technical report which was published by the Center for Climate System Research, the Univ. of Tokyo. The report was edited by H. Hasumi and S. Emori, but the first author is not specified. Although "K-1 model developers" doesn't look like a reference, a manuscript, which was published in GMD in 2011 (Watanabe et al. 2011, vol. 4, 845-874) cited this report in a same manner as in the present manuscript. So, "K-1 model developers" remains unchanged in the revised manuscript.

L116 As before: How may improvement of parameterizations "may result in reducing uncertainty"? We have deleted "and may result in reducing uncertainty range of climate projections". Please see the line 116.

L126 The sentence on the "signal-to-noise ratio" is difficult to understand.

We have deleted " because the signal-to-noise ratio is smaller in the mid-latitude atmosphere than in the tropics". Please see the lines 126-128.

L139/140 I'd try to avoid terms like high-resolution" or "medium resolution" The notion of what is high, medium or low is very different among climate modelers and certainly changes over time.

I agree with the reviewer's comment. We have rephrased the sentence as "Considering that the computational costs of large ensemble predictions based on climate models with horizontal resolutions of,

for example, 50 km atmosphere and eddy-resolving ocean are still huge on recent computer systems, the use of relatively low resolution models such as MIROC6...". Please read the lines 136-140. Also, we have tried to avoid the use of terms like "medium" or "high" resolution through the text.

L151 I suggest to add a sentence on how MIROC relates to MIROC-ESM (which is referred to in Section 3.2.2). In my understanding MIROC6 is a climate model that concentrates on the physical part of the Earth system and it would be useful to mention that because many of the models used in CMIPs these days include some component cycles.

Following the comments, we have added a sentence on our earth system model and relationship between the earth system model and MIROC. Please read the lines 148-150.

Section 2: I'm missing some technical information in the model description. I guess that in particular time steps (atmosphere, ocean, coupling, exceptions for specific parameterizations) had to be changed in comparison to MIROC5.

Following the comment, we have written the timesteps used in the sub-models, coupling interval and specific parameterizations in MIROC6 and MIROC5. Please see the lines 178-181, 247-248, 287-288, and 364-366 in the revised manuscript.

Fig. 1: Do the marks indicate half levels or full levels or what else? The marks indicate model half levels. We have revised the caption of Figure 1.

L164 Why is there a Table A1? I'd suggest to use a simple numbering for all tables.

The corresponding table is placed in Appendix. So, we numbered as Table A. But, we had typo in the previous manuscript. In the revised manuscript, "Table A1" is replaced by "Table A".

L174 I would speak of "model top", not "TOA".

"TOA" is replaced by "model top". If this replacement is adequate, other TOA in the manuscript is also replaced by "model top".

L184 Not clear what is meant with "dry air : : : is insufficient".

The sentence was not grammatically correct. We have corrected the sentence as " These biases appear to be the result of insufficient vertical mixing of the humid air in the planetary boundary layer and the dry air in the free troposphere". Please read the lines 186-187.

L224 Remove "a" .: "a" is removed. Thank you for your comment.

L225 Not clear why there is reference to "future versions". Does the current model version use the described features or not?

The current model (MIROC6) use the described features. The authors just wanted to express "extended capability may be effective in future climate modeling study. To avoid confusion, the corresponding sentence has been removed in the revised manuscript.

L233 Tuning of gravity wave parameterizations. Often, the Hines parameterization is used with very simplified and globally homogeneous characteristics of the gravity waves at the launching levels? This may make it, however, difficult to tune as well the QBO and high-latitude circulations. In particular as this is a feature new to MIROC it would be useful to elaborate a bit more on the tuning of GW parameters.

We have revised the text to include some more explanations on the non-orographic gravity wave parameterization. In the revised manuscript, "Following Watanabe (2008), a present-day climatological source of non-orographic gravity waves, which is estimated using results of a gravity wave-resolving version of MIROC-AGCM (Watanabe et al., 2008), is launched at the 70 hPa level in the extratropics, while an isotropic source of non-orographic gravity waves is launched at the 650 hPa level in the tropics" has been added. Please read the lines 231-235. The corresponding references (Watanabe 2008; Watanabe et al. 2008) have been added to the reference list.

L244 Example of the SSNOWD parameterization: It is useful to mention that the new parameterization is "physically" based. But in general I would like to read what the motivation for introducing changes with respect to MIROC5, what the expectations were, and, later in the evaluation, if the expectations were met. This is done well for some of the changes (e.g. L271: "increased vertical layers have been adopted in order to .."), but less for others (e.g. the tripolar ocean grid mentioned at line 262). I would like to ask the authors to do this more consistently for the changes because it may help other modeling groups to judge if specific changes may be worthwhile to apply in other models or not.

Thank you for the comment. Regarding the SSNOWD, we have added the descriptions about the reason we implemented the scheme as "in order to improve seasonal cycle of snow cover". Please read the lines 250-252. For the ocean component, we have added the sentences "By introducing the horizontal tripolar

coordinate system, it is expected that theoretical westward propagation of the oceanic baroclinic Rossby can be represented with less numerical dispersions because of agreement of the coordinate system and the geographical coordinate system and that the horizontal resolutions in the Arctic Ocean where the Rossby radius of deformation is relatively small are higher than in the case where the bipolar warped coordinate system in MIROC5 is adopted". Please read the lines 274-279.

L309 Only extinction coefficients would not allow to compute the radiative effects of aerosols.

In the revised manuscript, the corresponding sentence has been rewritten as "Radiative forcing of stratospheric aerosols due to volcanic eruptions are computed by vertically integrating extinction coefficients for each radiation band, which are provided by Thomason et al. (2016), in the model layers above the tropopause". Please read the lines 321-323.

L322 "These" instead of "This".: Done.

L328 I guess averages over these time periods are used?

Yes. In the revised manuscript, we added "averaged" in front of the periods.

L344 "land surface components are determined" sounds odd. I guess they are interpolated from some dataset. Please specify.

Following the comment, we have added the information on how to make coastline and topography of the atmospheric and land surface models as well as those in the ocean component. The corresponding sentences have been rewritten or added as "Ocean model coastline geometry and bottom bathymetry are specified based on horizontal interpolation of the land and sea-floor dataset of ETOPO5 (National Geophysical Data Center, 1993).", "the land-sea distribution and land-sea area ratios on the atmospheric and land surface model grids are determined according the coastline geometry of the ocean component", and "Surface topography in the atmospheric and land surface component are also made using the ETOPO5 dataset. Note that horizontal grid arrangement of the land surface model is exactly same as the atmospheric component". Please read the lines 349-351, 359-360, 361-365. The reference for ETOPO5 dataset has been added to the reference list.

**L350 "Reproducibility" of what?**

In the revised manuscript, we have rephrased as "reproducibility of climatic-mean state and internal climate

**variations". Please see the lines 369-370.**

L362 Here, the first tuning step for the coupled model is described. But there needs to be some procedure to specify tuning parameters in the component models, or not? Furthermore, I think it would be very useful for other modellers to be more specific about which parameters have been tuned to which effect. In some places this is described in acceptable detail, but in particular the first two sentences of this paragraph are very vague.

As pointed out by the reviewer, before coupling component models, parameter tuning was done in stand-alone component model. However, the tuning procedures were complex and depend on the component model group in our modeling community. So, it is better to describe only the parameter tuning procedure for coupled system, we think. Thank you for your suggestion.

Following the 2nd comment by the reviewer, we have described details on tuning parameters for tropical climate system as "Specifically, parameters of reference height for cumulus precipitation, efficiency of the cumulus entrainment of surrounding environment and maximum cumulus updraft velocity at the cumulus base are used to tune strength of the equatorial trade wind, climatological position and intensity of the Inter-Tropical Convergence Zone (ITCZ) and South Pacific Convergence Zone (SPCZ), and interannual variability of El-Niño/Southern Oscillation (ENSO). In particular, the parameter for the cumulus entrainment is known as a controlling factor of ENSO in MIROC5 (Watanabe et al., 2011). Summertime precipitation in the western tropical Pacific and characteristic of tropical intraseasonal oscillations are tuned by using the parameter for shallow convection describing the partitioning of turbulent kinetic energy between horizontal and vertical motions at the sub-cloud layer inversion". Please read the lines 385-393.

L376 Which "cooling effects" are meant, here? Aerosol-cloud effects as mentioned in the sentence before? Or total aerosol effects? I'd also prefer to speak of "radiative forcing" instead of cooling effects.

We tuned the total radiative forcing associated with aerosol-radiation interaction and aerosol-cloud interaction. In the revised manuscript, we have specified this and also have used "radiative forcing" instead of "cooling effects". Please read the lines 402-407.

L402 Apparently global mean SAT is not a tuning goal. It might be useful to mention this, and also why not. Additionally, I'd like to read a comment on the imbalance of about 1 Wm-2 that seems to exist in equilibrium. It seems like there is some artificial energy source in the model. Is there any knowledge where this originates from? Atmosphere, ocean, dynamical core, specific parameterizations? Is it known if this changes with the model state?

Following the comment, we have inserted the sentences "As above-mentioned, reproducibility of the global-mean SAT is not a tuning goal but is a typical metric which reflects results of the parameter tunings for individual processes of convections, dynamics, and radiative forcing" in the lines 416-418 in the revised manuscript. And following the 2nd comment of the reviewer, we have added the explanation on the heat energy inconsistency between the TOA radiation budget and the ocean heat uptake in association with the model imperfection. We have rephrased the last paragraph of Section 2.5 as "The trend of the global-mean ocean temperature in the later period suggests slight but continuous warming of the deep ocean. The radiation budget at the TOA is 1.1 Wm-2 downward on average (linear trend of  $9.5 \times 10^{-3}$  K/100 yr) and the net heat input at the sea surface is  $0.32 \text{ Wm}^{-2}$ . The deep ocean warming is explained by the net heat input. Note that there is about 0.78  $\text{Wm}^{-2}$  inconsistency between the TOA radiation budget and the ocean heat uptake. This heat energy inconsistency is due to that internal energy associated with precipitation, water vapor and river runoff is not taken account in the atmospheric and land surface component in MIROC6 and that these waters with no temperature information implicitly set their temperature to the SST when they flow or fall into the ocean. Perpetual melting of the prescribed Antarctic ice-sheet with invariant ice thickness, which is occurred due to the warm SAT bias in the Antarctic region (details will be discussed in Section 3.1.3), is also a cause of the heat energy inconsistency".

Fig. 4 and corresponding text: Names for the TOA fluxes are confusing. What is called NET is actually the total net flux, while what is called OSR is the net SW flux.

Thank you for the comment. To specify what NET, OSR, and OLR denote in the present manuscript, we have added the sentence "Hereafter, net shortwave, longwave, and the sum of them are denoted as OSR, OLR and NET, respectively, for simplicity" to the text and the caption of Figure 4. Please see the lines 473-474.

L426 The sentence on the consideration of "global-mean values" for RMSE calculations is difficult to understand. Maybe provide a formula or clearer description on how the RMSE are calculated? Is that true for all RMSE in this manuscript? It would be good to mention in every caption of Figures where RMSE are presented how these values are calculated, i.e. in particular if a global or some other mean have been subtracted before calculation of the error. The OLR in Fig. 4 looks particularly confusing without such information because while the RMSE is smaller in MIROC6 than in MIROC5 one would guess otherwise

from the color shading because of the dominance of red in the case of MIROC6.

In the present manuscript, all of RMSE were calculated without global-mean values. We have described this clearly in the revised manuscript as "In the present manuscript, RMSE is computed without model and observed global-mean quantities unless otherwise noted". Please read the lines 469-470 and the caption of Fig. 4. The above is not described in every figure where RMSE are presented in order to avoid lengthy caption, and a formula for RMSE calculation is not added because we use the most conventional formula. Thank you for your suggestion.

L476 I accept that for many climate variables it may not be essential if the evaluation is done for a pre-industrial or present-day simulation. But for some it is crucial. The energy balance is such a case, because the total net TOA flux should be zero in equilibrium. One can't say that the imbalance in the models is consistent with some observed imbalance, because the latter is related to the system not being in balance currently. It is also necessary to provide a reference for the observed value.

Following the comment, we have added the sentence "However, the observed value is estimated in the present-day condition. Ideally, the model value in the preindustrial condition should be 0 Wm-2 and is in the marginally acceptable range.". The reference of the observed values has been added as "(CERES; Loeb et all, 2009)". Please read the lines 521-522, and 520. Also, we have described possible cause of the non-zero TOA flux in our climate model. Please read our reply to the reviewer's comment to L. 402.

Fig. 4 and others. Parts of this and other figures are very blurred. This should be improved for the final publication.

Although the figures in the automatically-generated PDF manuscript look blurred, all the figures in the present manuscript was originally prepared in the EPS format. In final publication, higher resolution figures based on the EPS figure files can be used.

L512 The region of the western tropical Pacific is singled out as a region of improvement in MIROC6. It should be mentioned that it seems that in other regions there is a clear worsening.

Following the comment, we have added the sentences "On the other hand, model representation of the precipitation in MIROC6 is not necessarily alleviated other than the western tropical Pacific. For example, the overestimate of wintertime precipitation over the Indian Ocean and the mid-latitude North Pacific is worse in MIROC6 than in MIROC5" in the lines 561-564.

L518 For the discussion of the upper stratospheric warm bias it also matters that present-day and pre-industrial are compared due to the known stratospheric cooling with increased GHGs and reduced ozone.

As shown in the figure just below and as suggested by the reviewer, the zonal-mean climatology of the stratospheric (tropospheric) temperature in 1980-2009 of a historical simulation is colder (warmer) than in the preindustrial simulation. Thus, the temperature bias shown in Fig. 7c can be smaller when the modeled temperature in the present-day simulation is compared with observations. In the revised manuscript, we have added the sentences "Note that the zonal-mean temperature bias in Fig. 7c is smaller when the climatological-mean temperature from 1980 to 2009 in a historical simulation are evaluated against observations because of the known stratospheric cooling with increased greenhouse gases and reduced  $O_3$  concentrations." in the lines 579 -582.

Zonal-mean climatological temperature difference between a historical simulation (1980-2009) and the preindustrial simulation (shading). Contours denote values in a historical simulation.

L519: Again, I would prefer to speak of model top or lid and not of TOA. In the revised manuscript, "model top" is used instead of TOA.

L522 It would be good to say that this is the stream function of zonal mean meridional winds and not of residual winds, I guess.

Following the comment, we have specified. Please read the line 571. Thank you for the comment.

**L524 Please rephrase this sentence.**

In the revised manuscript, we have rephrased the sentence as "It is considered that an increased upward advection of the temperature minimum around the tropopause in 30°S–30°N may lead to reduction of warm temperature bias in the stratosphere which is significant in MIROC5." Please read the lines 572-574.

L530 I guess the absorption of LW radiation plays a minor role compared to SW radiation.

We just had a wrong description. In the revised manuscript, "longwave" has been replaced by "shortwave". Please see the line 579. Thank you for your suggestion.

L561 "extend", not "extends"

Done.

L561 Not clear what "more active troposphere-stratosphere" interactions are supposed to mean (radiative, wave coupling, trace gas exchange?) and why the stream functions would indicate that.

As shown in Fig. 25 and its explanation described in Section 3.2.2, the stratosphere-troposphere interactions associated with the Northern Annular Mode is better simulated in MIROC6 than in MIROC5. Thus, we have specified and rephrased as " more active troposphere-stratosphere interactions associated with wave-coupling exist in MIROC6". Please read the lines 614-615.

L566 There is no "remarkable improvement" in May. Furthermore, I suggest to avoid subjective terms like "remarkable". Please check all the text.

In the revised manuscript, the corresponding sentence has been changed to "it can be seen that the former parameterization brings about significant improvement in the Northern Hemisphere snow cover fractions from the early to the late winter" and the sentences " Note that no clear improvement is found in May" has been added. Please see the lines 619-62 and 625. Following the comments, we have deleted or rephrased "remarkable" through the text. Thank you for your suggestion.

L586 "into which cold and dense water forms" Please rephrase.

The sentences have been rephrased and the associated sentences have been also rewritten for clear descriptions. Please read the lines 638-641.

L609 This is no sentence.

"and" at the last of the sentence has been removed in the revised manuscript.

L612 The caption of Fig. 15 says only "temperature" while here you speak of "potential temperature". "potential" have been added in the caption of Fig. 15. Thank you for your comment. L614 Please rephrase "is risen": "risen" is replaced by "upwelled". Please see the line 671...

L623 "Better representation of cloud physics" would be "required" for what?

We have specified for "what" and have revised the corresponding sentences as " Better parameterizing deep cumulus convection in the models could be required for better representation of the equatorial trade winds and thus oceanic states." Please read the lines 682-684 and the lines just before.

L638 Remove "to". : Done.

L648 It would be useful to check if the historical simulation shows more realistic numbers in the comparison to present-day sea ice.

As shown in the figure just below, the model does not capture the realistic number and amplitude of drastic decrease of the sea-ice area. However, the September sea-ice area in 1980–2009 of historical simulations in MIROC6 is smaller than the observations, indicating that decreasing trend of the sea-ice area in the twenties century in MIROC6 is slightly larger than in observations. In the revised manuscript, the corresponding descriptions have been added. Please read the lines 711-715.

Time series of the Northern Hemisphere sea-ice area in historical simulations of MIROC6 (blue) and observations (black). Because data reliability before 1979 is not high in observations, the observed values are plotted only after 1979. Note that each blue line indicates the result from each ensemble historical simulation.

L649 Is there any idea why Antarctic sea ice is strongly underestimated?

The possible reason is described in the 2nd paragraph of Section 3.1.3 together with another prominent biases and theirs causes.

L655 Please specify what "sea level height" is presented in Fig. 18 and to which data it. Is compared in the text.

We have rewritten the corresponding sentence and the caption of Fig. 18 as "sea level height relative to the geoid" (L. 720). Also, a figure of observed sea level height has been added to the revised manuscript as Fig. 18a.

L676 "land surface variables" Actually I identify only one: snow cover.

"land surface variables" has been replaced by "the snow cover fractions in the Northern Hemisphere". Please see the line 744.

L678 Again, global mean SST and SAT are variables for which the comparison of preindustrial simulations and present-day observations is misleading.

Following the comment, we have added notes as " However, since the observed (model) value is estimated in the present-day (preindustrial) condition, the model global-mean SATs and SSTs are overestimated". Please read the lines 749-750.

L695 A prominent feature of SST and SAT biases is the strong warm bias close to Antarctica. This should be mentioned when discussing these variables.

I agree with the reviewer's comment. For example, you can find descriptions on the warm biases in the last sentence of the 3rd paragraph of Section 4 (Summary and Discussions) as "In the Southern Hemisphere, however, the underestimate of mid-level clouds and the corresponding warm SAT bias, the underestimate of sea-ice area, and the overestimate of incoming shortwave radiation in the Southern Ocean, all of which are attributed to errors in cloud radiative and planetary boundary layer processes (e.g., Bodas-Salcedo et al., 2012; Williams et al., 2013), remains the same as in MIROC5".

L718 It might be good to also mention the apparently missing features WIG and possibly EIG in the models. And what about anti-symmetric waves?

Thank you for your suggestion. In revised manuscript, we have mentioned the missing features of WIG and EIG waves. Also, we have added the figures for the zonal wavenumber–frequency power spectra of antisymmetric waves to Fig. 20 and descriptions on the antisymmetric waves. Please read the 1st paragraph of Section 3.2.1.

L727 Remove "and". : Done.

L743 Remove "MIROC6" : Done.

L760 "become" Rephrase.

In the revised manuscript, "become closer to observations" has been changed to "is consistent with observations". Please see the line 836.

L778 MIROC-ESM should be introduced in the beginning (or here)

Following the comments, we have added a sentence on our earth system model and relationship between the earth system model and MIROC. Please read the last sentence of Section 1.

L778 "whic": "whic" is replaced by "which" in the revised manuscript. Thank you very much.

L787 Remove "that" : Done.

L787 What means "correlations...are not clear"? Insignificant? Small? In the revised manuscript, "not clear" has been rephrased as "insignificant". Please see the line 864.

L793 SSWs are only a typical feature of the NH stratosphere.

In the revised manuscript, "a typical intraseasonal variability in the mid-latitude stratosphere" has been rephrased as "a typical intraseasonal variability of the mid-latitude stratosphere in the Northern Hemisphere". Please see the lines 869-870.

Fig. 24 and its discussion. I'd find it helpful to add Figures for January to make clear also the deficiencies of the model.

Following the comment, we have added January maps to Figure 14(d-f), and the figure caption and the text have been rewritten consistently with the new Fig. 14. Please read the paragraph starting from the line 869.

L807 One can't evaluate the polar night jet in Fig. 7e because it shows annual means. It might actually be an option to add some seasonal wind fields in Fig. 7. The paper has many figures anyhow, so I wouldn't mind adding a few more. Additionally there is a problem of chicken and egg with the wave-mean flow

interaction mentioned here.

Following the comment, we have added January maps to Figure 14 as written in the reply to the comment just above. And we have rephrased the corresponding sentence as " It is conjectured that the less frequent SSW in December–January could be attributed to less frequent stationary wave breakings due to overestimate of climatological zonal wind speed of the polar night jet in MIROC6 (Figs. 24d and e)". Please read the lines 884-887.

L846 Rephrase "existence depths".

In the revised manuscript, we have replaced "However, the existence depths of the subsurface signals are larger in MIROC6 than in observations" with "However, the subsurface signals in MIROC6 reside deeper than in observations." Please read the line 924.

L885 "SLP anomalies are larger and better represented in MIROC6" The maximum is deeper, but otherwise I find it hard to judge which of the two models is better.

In the revised manuscript, the corresponding sentence is rephrased as "it can be seen that the amplitudes of the SLP anomalies in MIROC6 are larger than in MIROC5, which is closer to the observation". Please read the lines 962-963.

L889 I'd avoid words like "excessively" which are subjective statements.: "excessively" is deleted.

L918 In IPCC AR5 and also the paper by Andrews et al. (2012) which is cited, here, climate sensitivity and forcing are calculated from 150 years of the 4xCO2 simulation, not 20 years. I would suggest to follow this 150-year standard to ensure comparability. Some models show clear non-linearities during this period. It seems like the effect is relatively small in MIROC5, but this would need to be confirmed for MIROC6.

The authors agree that analysis for the first 20 years is not consistent with the CMIP6 protocol. Following the comment, we repeated the analysis using the first 150-yr-long data, and confirmed that the results were similar to the ones based on the first 20-yr-long data. The manuscript is updated based on the present analysis. Please read Section 3.3 of the revised manuscript. Also, Figures 30, 31 and Tables 2 and 3 have been replaced by the revised ones.

Tables 2 and 3: It would be convenient for the reader to combine the tables. We consider the table may be more complex and the caption would be very long if the Tatebe 2 & 3 are combined. So, Table 2 & 3 are not combined in the revised manuscript. Thank you for your comment.

L966, 979 Again, please avoid "remarkably".: "remarkably" has been deleted.

**L999 Why would ECS quantify uncertainty?**

The sentence was not appropriate because ECS quantifies climate change itself, not uncertainty of climate change. We therefore rephrased the sentence "As a metric for climate change induced by atmospheric CO2 increase, ECS is also estimated". Please see the line 1075.

L1028 It's true that the hiatus is sometimes associated with the IPO, but there are plenty of other attempts to explain it and even arguments that the real reason maybe unidentifiable. So I'd suggest to not only mention the IPO.

We have added other candidate for the hiatus and we have rewritten the corresponding sentences as "The observed hiatus is considered to occur in association with a negative IPO phase as internal climate variations (e.g., Meehl et al., 2011; Watanabe et al., 2014). As external drivers of the hiatus, the weakening of solar activity and increase in stratospheric aerosols are given as possible candidates, for example (e.g., Solomon et al., 2010; Kaufmann et al., 2011)". Please read the lines 1103-1107 in the revised manuscript.

L1031 I don't understand this sentence. I agree that the simulated hiatus could be spurious, but the argument of the ensemble mean wouldn't support this.

After submitting the manuscript, we increased the number of ensemble historical simulations by MIROC6 up to 30 members. When we redrew the time series of the global-mean SAT anomalies using 30 members, the hiatus-like temperature change in the early 21th century is vanished and continuous temperature rise is appeared. We have replaced Fig. 32 by the new one with 30 ensemble members and we have rewritten the descriptions on the model hiatus as " The so-called recent hiatus of the global warming (Easterling and Wehner, 2009) in the first decade of the twenty-first century is not simulated in both of MIROC6 and MIROC5" and as " Failure of simulating the hiatus in the models could be attributed to uncertainties in the historical forcing datasets or cancellation of internal climate variations of the IPO by ensemble-mean manipulation of the individual historical simulations". Please read the lines 1101-1103 and 1107-1109 in the revised manuscript.

L1055 Should a new paragraph start here?

Because we would like to give an example of error compensation in oceanic processes, we did not start a new paragraph here. In the revised manuscript, we have rewritten the sentence describing the oceanic error compensation and have not started a new paragraph. Please read the line 1130-1135.

L1068 I have no idea why the final sentence suddenly makes a statement concerning component cycles which were not at all mentioned anywhere else in the text.

In the revised manuscript, the corresponding sentence has been deleted.

L1074 I don't know what the policy of GMD is concerning the availability of primary data (which I think should be the code of the model and all input data needed to redo the experiments), but I find it problematic that the code is only available under the condition of "collaborative research". As mentioned in my initial statement, a model description is necessarily incomplete. It can only be completed by the model code.

Following the reviewer's suggestions and that the simulation data used in the present manuscript have been distributed from December 2018 and the data are freely accessible, we have rewritten the code and data availability part as "Please contact the corresponding author if readers may want to validate the model configurations of MIROC6 and MIROC5 and to conduct replication experiments. The source codes and required input data will be provided by the modeling community where the author belongs. The model output from the CMIP6/CMIP5 pre-industrial control and historical simulations used in the present manuscript are distributed through the Earth System Grid Federation and are freely accessible. Details on ESGF are given on the CMIP Panel website (https://www.wcrp-climate.org/wgcm-cmip)."

[revised manuscript text omitted]

**42 **1 Introduction**

As the global warming due to increasing emissions of the anthropogenic greenhouse gases progresses, it is anticipated, or has been already observed that global and regional patterns of elimatie mean atmospheric temperature, circulations, and precipitation as well as temperature are projected to be will drastically changed until the end of the twentieth-first century (e.g., Neelin et al., 2006; Zhang et al., 2007; Bengtsson et al., 2009; Andrews et al., 2010; Scaife et al., 2012) and that occurrence frequency of

49 extreme weather events such as heatwaves, droughts will be increased, and extratropical cyclones will 50 be stronger than in the present increase (e.g., Mizuta et al., 2012; Sillmann et al., 2013; Zappa et al., 51 2013). Corresponding to the atmospheric changes under the global warming, the sea levels will rise 52 due to the thermal expansion of sea water and ice-sheet melting in the polar continental regions (e.g., 53 Church and White, 2011; Bamber and Aspinall, 2013). Additionally, ocean acidification due to 54 absorption of atmospheric carbon dioxide (CO2) and changes in carbon-nitrogen cycles are expected 55 to lead to the loss of Earth biodiversity (e.g., Riebesell et al., 2009; Rockström, et al. 2009; Taucher 56 and Oschlies, 2011; Watanabe et al., 2017). Societal demands for information on the global and 57 regional climate changes have increased significantly worldwide in order to meet information 58 requirements for political decision making related to mitigation and adaptation to the global warming. 59 The Intergovernmental Panel on Climate Change (IPCC) has continuously published the 60 assessment reports (ARs) in which a comprehensive view of past, present, and future climate changes 61 on various timescales, including the centennial global warming, are synthesized (IPCC 2007; 2013). 62 Together with observations, climate models have been contributing to the IPCC-ARs through a broad 63 range of numerical simulations, especially, future climate projections after the twenty-first century. 64 However, there are many uncertainties in future climate projections and the range of uncertainties has 65 not been narrowed by an update of the IPCC reports. The uncertainties are arising from imperfections

66 of climate models in representing micro- to global-scale physical and dynamical processes in sub-67 systems of the Earth's climate and their interactions. To reduce the uncertainties and errors in climate 68 projections and predictions, utilizing observations, extracting essences of physical processes in the 69 real climate, and investigating the response of the climate system to various external forcings based 70on a set of climate model simulations sophisticating physical parameterizations of elimate models, 71 which represent unresolved sub-grid scale phenomena, are necessary. In particular, aA state-of-the-art 72 climate model which can represent various processes in the Earth's climate system is a powerful tool 73 for deeper understanding the Earth's climate system.

74 One of Japanese climate models, which is called MIROC (Model for Interdisciplinary 75 Research on Climate), has been cooperatively developed at the Center for Climate System Research 76 (CCSR; the precursor of a part of the Atmosphere and Ocean Research Institute), the University of 77 Tokyo, the Japan Agency for Marine-Earth Science and Technology (JAMSTEC), and the National 78 Institute for Environmental Studies (NIES). Utilizing MIROC, our Japanese climate modelling group 79 has been tackling a wide range of climate science issues and seasonal-to-decadal climate predictions 80 and future climate projections. At the same time, by providing simulation data, we have been 81 participating to the third and fifth phases of the Coupled Model Intercomparison Projects (CMIP3 and 82 CMIP5; Meehl et al. 2007; Taylor et al. 2011) which have been contributing to the IPCC-ARs by 83 synthesizing multi-model ensemble datasets.

In the years up to the IPCC fifth assessment report (IPCC-AR5; IPCC 2013), we have developed four versions of MIROC, three of which (MIROC3m, MIROC3h, and MIROC4h) have almost the same dynamical and physical packages, but different resolutions. MIROC3m (K-1 model developers, 2004) is composed of a medium-resolution model consisting of T42L20 atmosphere and 1.4°L43 ocean-components. Resolutions of MIROC3h (K-1 model developers, 2004) are higher than MIROC3m and are T106L56 for the atmosphere and eddy-permitting for the ocean ( $1/4^{\circ} \times 1/6^{\circ}$ ). Only 90 the horizontal resolution of the atmosphere of MIROC3h is changed to T213 in MIROC4h (Sakamoto 91 et al., 2012). MIROC5 is a medium-resolution model composed nsisting of T85L40 atmosphere and 92 1.4°L50 ocean components, but with considerably updated physical and dynamical packages 93 (Watanabe et al., 2010). These models have been used to study various scientific issues such as the 94 detection of natural influences on climate changes (e.g., Nozawa et al., 2005; Mori et al, 2014; 95 Watanabe et al., 2014), uncertainty quantification of climate sensitivity (e.g., Shiogama et al., 2012; 96 Kamae et al., 2016), future projections of regional sea-level rises (e.g., Suzuki et al., 2005; Suzuki and 97 Ishii, 2011), and mechanism studies on tropical decadal variability (e.g., Tatebe et al., 2013; Mochizuki 98 et al., 2016).

99 During the last decade, our efforts have been preferentially devoted to providing science-100 oriented risk information on climate changes that is beneficial to international, domestic, and 101 municipal communities. For example, so-called event attribution (EA) studies with large ensemble 102 simulations initiated from slightly different conditions have been conducted in order to statistically 103 evaluate influences of the global warming on the occurrence frequencies of observed individual 104 extremes (e.g., Imada et al., 2013; Watanabe et al., 2013; Shiogama et al., 2014). Seasonal-to-decadal 105 climate predictions are also of significant concerns. By initializing prognostic variables in our climate 106 models using observation-based data (Tatebe et al., 2012), significant prediction skills in several 107 specific phenomena, such as the El Niño/Southern Oscillation (ENSO) and the Arctic sea-ice extent 108 on seasonal timescales, the Pacific Decadal Oscillations (PDO; Mantua et al., 1997), the Atlantic 109 Multi-decadal Oscillations (AMO; Schlesinger and Ramankutty, 2004), and the tropical trans-basin 110 interactions between the Pacific and the Atlantic on decadal timescales, are detected (e.g., Mochizuki 111 et al., 2010; Chikamoto et al. 2015; Imada et al., 2015; Ono et al., 2018).

112 However, while the applicability of MIROC has been extended to a wide range of climate 113 science issues, almost all of the above-mentioned approaches were based on our medium-resolution

114 versions of MIROC (MIROC3m and MIROC5), and it is well known that higher-resolution models 115 are capable of better representing the model mean climate and internal climate variability, such as 116 regional extremes, orographic winds, and oceanic western boundary currents/eddies than lower-117 resolution models (e.g., Shaffrey et al., 2009; Roberts et al., 2009; Sakamoto et al., 2012). Nevertheless, 118 even in high-resolution models, there remain persistent biases associated with, for example, cloud-119 aerosol-radiative feedback and turbulent vertical mixing of the air in the planetary boundary layer (e.g., 120 Bony and Dufresne, 2005; Bodas-Salcedo et al., 2012; Williams et al., 2013), which are tightly linked 121 with dominant uncertainties in climate projections. Therefore, improvement of physical 122 parameterizations for sub-grid scale processes is essential for better representing observed climatic-123 mean states and internal climate variability-and may result in reducing uncertainty range of climate 124 projections. As well as physical parameterizations, enhanced vertical resolution in both of atmosphere 125 and ocean components, along with a highly accurate tracer advection scheme, have been suggested to 126 have impacts on reproducibility of model-climate and internal climate variations (e.g., Tatebe and 127 Hasumi, 2010; Ineson and Scaife, 2009; Scaife et al., 2012).

128 Recently, we have developed the sixth version of MIROC, called MIROC6. This newly 129 developed climate model has updated physical parameterizations in all sub-modules. In order to 130 suppress an increase of computational cost, the horizontal resolutions of MIROC6 are not significantly 131 higher than those of MIROC5. The reason is that a larger number of ensemble members are required 132 to realize significant seasonal predictions of, for example, the wintertime Eurasian climate (Murphy 133 et al., 1990; Scaife et al., 2014) because the signal-to-noise ratio is smaller in the mid-latitude 134 atmosphere than in the tropies. Indeed, climate predictions by the older versions of MIROC having at 135 most 10 ensemble members are skillful only in the tropical climate andor the mid-latitude ocean not 136 in the smid-latitude atmosphere. - In addition, when evaluating the contributions of internal variations, 137 which will be done in preparation for use in the global stocktake, namely, a five-yearly review of each

138 countries' provisions to climate changes, established by the Paris Agreement in 2015, ILarge ensemble 139 predictions are may also be 
[revised manuscript text omitted]
 tropopauseare taken into 337 account as extinction coefficients for each radiation band. Three-dimensional atmospheric 338 concentrations of historical ozone (O3) are produced by the Chemi